# Quantum Algorithms for Finite-horizon Markov Decision Processes

Bin Luo [1]   Yuwen Huang [1]   Jonathan Allcock [2]   Xiaojun Lin [1]   Shengyu Zhang [2]   John C.S. Lui [1]

## Abstract

In this work, we design quantum algorithms that are more efficient than classical algorithms to solve time-dependent and finite-horizon Markov Decision Processes (MDPs) in two distinct settings: (1) In the exact dynamics setting, where the agent has full knowledge of the environment's dynamics (i.e., transition probabilities), we prove that our **Quantum Value Iteration (QVI)** algorithm **QVI-1** achieves a quadratic speedup in the size of the action space ($A$) compared with the classical value iteration algorithm for computing the optimal policy ($\pi^*$) and the optimal V-value function ($V_0^*$). Furthermore, our algorithm **QVI-2** provides an additional speedup in the size of the state space ($S$) when obtaining near-optimal policies and V-value functions. Both **QVI-1** and **QVI-2** achieve quantum query complexities that provably improve upon classical lower bounds, particularly in their dependences on $S$ and $A$. (2) In the generative model setting, where samples from the environment are accessible in quantum superposition, we prove that our algorithms **QVI-3** and **QVI-4** achieve improvements in sample complexity over the state-of-the-art (SOTA) classical algorithm in terms of $A$, estimation error ($\epsilon$), and time horizon ($H$). More importantly, we prove quantum lower bounds to show that **QVI-3** and **QVI-4** are asymptotically optimal, up to logarithmic factors, assuming a constant time horizon.

## 1. Introduction

Markov Decision Processes (MDPs) provide a mathematical framework for modeling decision-making problems in uncertain environments. They are an important framework to model discrete-time stochastic control and reinforcement learning (RL) (Puterman, 2014; Agarwal et al., 2019).

MDPs have been applied in fields such as networks, robotics, and operations research (Alsheikh et al., 2015; Matignon et al., 2012). Despite their wide applicability, MDPs often face significant computational challenges in practice. A key issue arises when the number of possible states or actions in the system becomes very large. In particular, this "curse of dimensionality" makes solving MDPs computationally infeasible in many practical scenarios (Powell, 2007).

Quantum computing is a new computing paradigm that harnesses the laws of quantum mechanics. For certain classes of problems, such as unstructured search (Grover, 1996), prime number factoring (Shor, 1994), optimization (Sidford & Zhang, 2023; Jordan, 2005; Liu et al., 2024) and online learning (He et al., 2024; 2022; Wan et al., 2023), quantum computing demonstrates significant speedups over classical computing. Recent advancements in quantum hardware (Arute et al., 2019; AI et al., 2024) indicate that practical quantum computers can be a reality in the near future.

Given the importance of MDPs and the advancement in quantum computing, researchers have explored various quantum algorithms to reduce the time complexity of solving MDPs. In the stochastic control domain, (Naguleswaran & White, 2005) suggested two quantum techniques that can potentially be used to accelerate classical algorithms for finite-horizon MDPs (Puterman, 2014). However, this work only focused on problem formulation and did not provide a concrete quantum algorithm for finite-horizon MDPs with performance guarantee. (Naguleswaran et al., 2006) applied quantum walk (Magniez et al., 2007) to efficiently solve a specific class of MDPs, namely deterministic shortest path problems. However, the quantum algorithm and analysis there cannot be applied to general finite-horizon MDPs. For RL, researchers proposed to replace subroutines of existing RL frameworks by quantum algorithms. For example, (Wiedemann et al., 2022) proposed to use a quantum Monte Carlo (MC) (Montanaro, 2015) to replace the classical MC method on policy evaluation. However, their algorithm is inefficient as its quantum sample complexity is exponential with respect to $S$ in both time-dependent and time-independent settings for obtaining near-optimal policies. Improved sample-complexity results that do not increase exponentially in $S$ have been obtained for infinite-horizon MDPs. For example, (Cherrat et al., 2023) utilized a quantum linear system solver (Chakraborty et al., 2019) to

[1]The Chinese University of Hong Kong, Hong Kong, China [2]Tencent Quantum Laboratory, Hong Kong, China. Correspondence to: Yuwen Huang <yuwen.huang@link.cuhk.edu.hk>.

*Proceedings of the $42^{nd}$ International Conference on Machine Learning*, Vancouver, Canada. PMLR 267, 2025. Copyright 2025 by the author(s).

approximate Q-values during the policy evaluation. (Wang et al., 2021) proposed nearly minimax optimal quantum algorithms for infinite-horizon MDPs by leveraging quantum mean estimation (Montanaro, 2015) and quantum maximum searching (Durr & Hoyer, 1999). Besides, (Cornelissen, 2018) applied quantum gradient estimation (Gilyén et al., 2019) in policy improvement. However, these algorithms are only tailored to infinite-horizon problems with a time-invariant value function, thus *preventing their use in finite-horizon and time-dependent scenarios* where the value functions depend on time.

Thus, one open question is, *can one design quantum algorithms that are more efficient than classical algorithms in obtaining the optimal or $\epsilon$-optimal policy, V-value and Q-value functions for "finite-horizon" and "time-dependent" MDPs?* We address this open question in both the exact dynamics setting and the generative model setting. Our contributions are as follows:

- In the exact dynamics setting (Section 3), we propose a **Quantum Value Iteration (QVI)** algorithm **QVI-1**, that computes the optimal policy and V-value function with a quadratic speedup in $A$ compared with the classical value iteration algorithm. Additionally, **QVI-2** achieves a further speedup in $S$ for obtaining near-optimal policies and V-value functions, enabled by our novel quantum subroutine, quantum mean estimation with binary oracles (**QMEBO**), for mean estimation of arbitrary bounded functions. Besides, we also derive new classical lower bounds for computing near-optimal policies and V-value functions. A summary of these results is provided in Table 1.

- In the generative model setting (Section 4), we propose two quantum algorithms, **QVI-3** and **QVI-4**, to efficiently compute $\epsilon$-optimal policies and value functions. Compared with SOTA classical algorithms for time-dependent and finite-horizon MDPs, both **QVI-3** and **QVI-4** achieves speedups in $H$, and $\epsilon$, with **QVI-3** additionally achieving a quadratic speedups in $A$.

- Assuming access to a quantum generative oracle for time-dependent and finite-horizon MDPs, we establish quantum lower bounds for obtaining near-optimal policies, V-value functions, and Q-value functions. Our results demonstrate that **QVI-3** and **QVI-4** are asymptotically optimal, up to log factors, provided that $H$ is a constant. Further, our results also lead to a new lower bound for obtaining Q-values in the classical setting. A summary of the upper and lower bounds in the generative model setting is provided in Table 2.

## 2. Preliminaries

**Define notations:** For an arbitrary positive integer $n$, we define $[n]$ as the set $\{0, ..., n-1\}$. For any finite set $X$

and any vector $f \in Y^X$, we denote the element of $f$ at entry $x$ by $f(x)$. For any $f \in \mathbb{R}^X$, the operations $\sqrt{f}$, $|f|$, and $f^2$ are applied component-wise. Given two vectors $f_1, f_2 \in \mathbb{R}^X$, we define $\max\{f_1, f_2\}$ as their element-wise maximum, and write $f_1 \leq f_2$ to indicate component-wise inequality. The bold symbols $\mathbf{0}$ and $\mathbf{1}$ represent vectors of all zeros and ones, respectively, and a scalar $x$ in an equation with vectors should be interpreted as $x \cdot \mathbf{1}$. We usually identify a function $f : X \to Y$ as a vector $f \in Y^X$.

**MDP Preliminaries:** We study time-dependent and finite-horizon MDPs in two settings: (a) the exact dynamics setting (Section 3) and (b) the generative model setting (Section 4). In both settings, the MDP has a finite and discrete state space $\mathcal{S}$ and action space $\mathcal{A}$. In each time step $h \in [H]$, an agent need to decide which action $a \in \mathcal{A}$ to take for each state $s \in \mathcal{S}$. After taking the action $a$ at the state $s$ in the time step $h \in [H]$, the agent obtains a reward $r_h(s, a) \in [0, 1]$ and transitions to the next state $s' \in \mathcal{S}$ with probability $P_h(s'|s, a)$. We define a finite-horizon and time-dependent MDP as a 5-tuple $\mathcal{M} = (\mathcal{S}, \mathcal{A}, \{P_h\}_{h=0}^{H-1}, \{r_h\}_{h=0}^{H-1}, H)$. We define $S := |\mathcal{S}|$ and $A := |\mathcal{A}|$, which are the cardinalities of $\mathcal{S}$ and $\mathcal{A}$ respectively. A policy $\pi$ is a mapping from $\mathcal{S} \times [H]$ to $\mathcal{A}$, where $\pi(s, h)$ specifies the action that the agent should take in the state $s$ at the time step $h$. The policy space is defined as $\Pi := \mathcal{A}^{\mathcal{S} \times [H]}$. In MDPs, the objective of the agent is to find a policy $\pi$ that maximizes the expected cumulative reward over $H$ time horizon. This can be written as maximizing the V-value function, $V_h^\pi : \mathcal{S} \to \mathbb{R}$, at each time step $h$. Specifically, the V-value function at time $h$ for an initial state $s$ under a policy $\pi$ is defined as $V_h^\pi(s) := \mathbb{E}\left[\sum_{t=h}^{H-1} r_t(s_t, a_t) | \pi, s_h = s\right]$, where $a_t = \pi(s_t, t)$. Similarly, the Q-value function $Q_h^\pi : \mathcal{S} \times \mathcal{A} \to \mathbb{R}$ is defined as $Q_h^\pi(s, a) := \mathbb{E}\left[\sum_{t=h}^{H-1} r_t(s_t, a_t) | \pi, s_h = s, a_h = a\right]$.

For a policy $\pi$, we define $P_h^\pi \in \mathbb{R}^{\mathcal{S}\mathcal{A} \times \mathcal{S}\mathcal{A}}$ as the matrix with entries $P_h^\pi((s, a), (s', a')) = P_h(s'|s, a)$ if $a' = \pi(s')$ and 0 otherwise. For any $Q \in \mathbb{R}^{\mathcal{S} \times \mathcal{A}}$, we define $P_h^\pi Q \in \mathbb{R}^{\mathcal{S} \times \mathcal{A}}$ as $(P_h^\pi Q)(s, a) = \sum_{s' \in \mathcal{S}} P_h(s'|s, a) Q(s', \pi(s'))$. With a slight abuse of notation, we define $P_h \in \mathbb{R}^{\mathcal{S}\mathcal{A} \times \mathcal{S}}$ as the matrix satisfying $P_h((s, a), s') = P_h(s'|s, a)$ for any $h \in [H]$. For any fixed $s \in \mathcal{S}, a \in \mathcal{A}$ and $h \in [H]$, we define $P_{h|s,a} \in \mathbb{R}^{\mathcal{S}}$ as the vector satisfying $P_{h|s,a}(s') = P_h(s'|s, a)$. Therefore, we can express $\mathbb{E}[f(s')|s' \sim P_{h|s,a}] = P_{h|s,a}^{\mathsf{T}} f$ for any $f \in \mathbb{R}^{\mathcal{S}}$.

For any vector $v \in \mathbb{R}^{\mathcal{S}}$, we define $\sigma_h^2(v) \in \mathbb{R}^{\mathcal{S} \times \mathcal{A}}$ as a vector satisfying $[\sigma_h^2(v)](s, a) := \mathrm{Var}[v(s')|s' \sim P_h(\cdot|s, a)]$ for any $h \in [H]$. In the vector notation, it can be written as $\sigma_h^2(v) = P_h v^2 - (P_h v)^2$. We also define $\sigma_h(v) = \sqrt{\sigma_h^2(v)}$.

We define $V(Q) \in \mathbb{R}^{\mathcal{S}}$ as $[V(Q)]_s = \max_{a \in \mathcal{A}}\{Q(s, a)\}$ and $\pi(Q) \in \mathcal{A}^{\mathcal{S}}$ as $[\pi(Q)]_s = \arg\max_{a \in \mathcal{A}}\{Q(s, a)\}$ for any vector $Q \in \mathbb{R}^{\mathcal{S} \times \mathcal{A}}$.

| Goal: | Classical query complexity | | Quantum query complexity |
| --- | --- | --- | --- |
| | Upper bound | Lower bound | Upper bound |
| Optimal $\pi^*$, $V_0^*$ | $S^2AH$ | $S^2A$ [Theorem 3.2] | $S^2\sqrt{A}H$ [Theorem 3.6] |
| $\epsilon$-accurate estimate of $\pi^*$ and $\{V_h^*\}_{h=0}^{H-1}$ | $S^2AH$ | $S^2A$ [Theorem 3.2] | $\frac{S^{1.5}\sqrt{A}H^3}{\epsilon}$ [Theorem 3.9] |

*Table 1.* Classical and quantum query complexities for solving time-dependent and finite-horizon MDPs in the exact dynamics setting. All quantum upper bounds are $\tilde{O}(\cdot)$, assuming a constant failure probability $\delta$. The range of error term $\epsilon$ is $(0, H]$. The classical upper bounds are $O(\cdot)$, derived from the value iteration algorithm in (Puterman, 2014). The classical lower bounds are $\Omega(\cdot)$, which holds for $\epsilon \in O(H)$.

| Goal: obtain an $\epsilon$-accurate estimate of | Classical sample complexity | | Quantum sample complexity | |
| --- | --- | --- | --- | --- |
| | Upper bound | Lower bound | Upper bound | Lower bound |
| $\{Q_h^*\}_{h=0}^{H-1}$ | $\frac{SAH^4}{\epsilon^2}$ | $\frac{SAH^3}{\epsilon^2}$ [Theorem 4.7] | $\frac{SAH^{2.5}}{\epsilon}$ [Theorem 4.6] | $\frac{SAH^{1.5}}{\epsilon}$ [Theorem 4.7] |
| $\pi^*$, $\{V_h^*\}_{h=0}^{H-1}$ | $\frac{SAH^4}{\epsilon^2}$ | $\frac{SAH^3}{\epsilon^2}$ [Theorem 4.7] | $\frac{SAH^{2.5}}{\epsilon}$ [Theorem 4.6] $\frac{S\sqrt{A}H^3}{\epsilon}$ [Theorem 4.4] | $\frac{S\sqrt{A}H^{1.5}}{\epsilon}$ [Theorem 4.7] |

*Table 2.* Classical and quantum sample complexities for solving time-dependent and finite-horizon MDPs in the generative model setting. All bounds assume a constant maximum failure probability $\delta$. All upper bounds are $\tilde{O}(\cdot)$, which requires $\epsilon \in O(1/\sqrt{H})$ for [Theorem 4.6] and $\epsilon \in (0, H]$ for [Theorem 4.4]. All lower bounds are $\tilde{\Omega}(\cdot)$, which holds for $\epsilon \in (0, 1/2)$. The classical upper bounds for all goals were shown in (Li et al., 2020). The classical lower bound for $\pi^*$ and $\{V_h^*\}_{h=0}^{H-1}$ was shown in (Sidford et al., 2018).

Below, we provide formal definitions for some important concepts in the finite-horizon MDP $\mathcal{M}$.

**Definition 2.1** (Value operator associated with a policy). For any policy $\pi \in \Pi$, let $\mathcal{T}_\pi^h(\cdot)$ be the value operator associated with $\pi$ such that, for all $u \in \mathbb{R}^\mathcal{S}$, $h \in [H]$ and $s \in \mathcal{S}$, $[\mathcal{T}_\pi^h(u)]_s := r(s, \pi(s, h)) + P_{h|s,\pi(s,h)}^{\mathrm{T}} u$. We let $\{V_h^\pi\}_{h=0}^{H-1}$ denote the V-value functions of policy $\pi$, which satisfies $\mathcal{T}_\pi^h(V_{h+1}^\pi) = V_h^\pi$ for all $h \in [H]$.

**Definition 2.2** (Optimal value and policy). Define the optimal value of an initial state $s \in \mathcal{S}$ at each time step $h \in [H]$ of the finite-horizon MDP $\mathcal{M}$ as $V_h^*(s) := \max_{\pi \in \Pi} V_h^\pi(s)$. A policy $\pi$ is said to be an optimal policy $\pi^*$ if $V_0^\pi = V_0^*$. Similarly, we can also define the optimal value of an initial pair of $(s, a) \in \mathcal{S} \times \mathcal{A}$ at each time step $h \in [H]$ as $Q_h^*(s, a) := \max_{\pi \in \Pi} Q_h^\pi(s, a)$.

**Definition 2.3** ($\epsilon$-optimal value function and policy). We say that V-value functions $\{V_h\}_{h=0}^{H-1}$ are $\epsilon$-optimal if $\|V_h^* - V_h\|_\infty \leq \epsilon$ for all $h \in [H]$ and a policy $\pi \in \Pi$ is $\epsilon$-optimal if $\|V_h^* - V_h^\pi\|_\infty \leq \epsilon$ for all $h \in [H]$, which implies the V-value functions of $\pi$ are $\epsilon$-optimal. Similarly, we say that Q-value functions $\{Q_h\}_{h=0}^{H-1}$ are $\epsilon$-optimal if $\|Q_h^* - Q_h\|_\infty \leq \epsilon$ for all $h \in [H]$.

**Quantum Preliminaries:** Before introducing our quantum algorithms, a brief overview of Dirac notation (Nielsen & Chuang, 2010) is given to ensure clarity. In Dirac notation, vectors $v$ in a complex vector space $\mathbb{C}^n$ are represented

as $|v\rangle$. The symbol $|i\rangle$, where $i \in [n]$, denotes the $i + 1$-th standard basis vector, with $|0\rangle$ typically reserved for the first standard basis vector. In this paper, real numbers are encoded in the computational basis using a fixed-point binary representation with precision $2^{-p}$. Specifically, a real number $k$ is encoded as $|\mathrm{Bi}[k]\rangle = |k_1 \ldots k_q\rangle \in \mathbb{C}^{2^q}$, where $k_1 \ldots k_q = k_1 \ldots k_{q-p}.k_{q-p+1} \ldots k_q$ is the binary string of $k$. We assume that $q$ and $p$ are sufficiently large so that there is no overflow in storing real numbers.

We now define a quantum oracle for arbitrary functions and vectors, which is often referred to as *binary oracle*.

**Definition 2.4** (Quantum oracle for functions and vectors). Let $\Omega$ be a finite set of size $N$ and $f \in \mathbb{R}^\Omega$. A quantum oracle encoding $f$ is a unitary operator $B_f : \mathbb{C}^N \otimes \mathbb{C}^{2^q} \to \mathbb{C}^N \otimes \mathbb{C}^{2^q}$ such that $B_f : |i\rangle |0\rangle \mapsto |i\rangle |\mathrm{Bi}[f(i)]\rangle$ for all $i \in [N]$, where $\mathrm{Bi}[f(i)]$ is the binary representation of $f(i)$ with precision $2^{-p}$.

## 3. Exact Dynamics Setting

In this setting, it is assumed that the environment's dynamics are fully known, i.e., the transition probability matrix $P_h$ at each time step $h$ is explicitly provided for the entire state-action space. To formalize this assumption, we introduce the classical oracle for finite-horizon MDPs $O_\mathcal{M}$ in Definition 3.1. Given this classical oracle, the classical value iteration algorithm (Algorithm 6) can obtain an optimal policy $\pi^*$

and optimal value $V_0^*(s)$ for any initial state $s \in \mathcal{S}$ with $O(S^2AH)$ queries to the oracle $O_{\mathcal{M}}$ (Bellman, 1958).

**Definition 3.1** (Classical oracle of an MDP). We define a classical oracle $O_{\mathcal{M}} : \mathcal{S} \times \mathcal{A} \times [H] \times \mathcal{S} \to [0,1] \times [0,1]$ for a time-dependent and finite-horizon MDP $\mathcal{M}$ satisfying $O_{\mathcal{M}} : (s, a, h, s') \mapsto (r_h(s,a), P_{h|s,a}(s'))$.

To understand the limits of classical algorithms under this setting, we establish a lower bound on query complexity for computing near-optimal policies and V-value functions. This result adapts the techniques developed for infinite-horizon MDPs in (Chen & Wang, 2017) to the finite-horizon case. The rigorous proof of Theorem 3.2 is presented in Appendix A.2.

**Theorem 3.2** (Classical lower bounds). *Let $\mathcal{S}$ and $\mathcal{A}$ be finite sets of states and actions. Let $H \geq 2$ be a positive integer and $\epsilon \in (0, \frac{H-1}{4})$ be an error parameter. We consider the following time-dependent and finite-horizon MDP $\mathcal{M} = (\mathcal{S}, \mathcal{A}, \{P_h\}_{h=0}^{H-1}, \{r_h\}_{h=0}^{H-1}, H)$, where $r_h \in [0,1]^{\mathcal{S} \times \mathcal{A}}$ for all $h \in [H]$. Given access to the classical oracle $O_{\mathcal{M}}$, any algorithm $\mathcal{K}$, which takes $\mathcal{M}$ as an input and outputs $\epsilon$-approximations of $\{V_h^*\}_{h=0}^{H-1}$ or $\pi^*$ with probability at least 0.9, must require at least $\Omega(S^2A)$ queries to $O_{\mathcal{M}}$ on the worst case of input $\mathcal{M}$.*

### 3.1. Speedup on $A$

Having established the classical baseline, we now turn to investigating whether quantum algorithms can offer improvements, particularly in the dependence on the action space size $A$. To have a fair comparison on the time complexity between a classical algorithm and a quantum algorithm, we first define the quantum analog of the classical oracle $O_{\mathcal{M}}$.

**Definition 3.3** (Quantum oracle of an MDP). A quantum oracle of an MDP $\mathcal{M}$ is a unitary operator $O_{\mathcal{QM}} : \mathbb{C}^S \otimes \mathbb{C}^A \otimes \mathbb{C}^H \otimes \mathbb{C}^S \otimes \mathbb{C}^{2^q} \otimes \mathbb{C}^{2^q} \to \mathbb{C}^S \otimes \mathbb{C}^A \otimes \mathbb{C}^H \otimes \mathbb{C}^S \otimes \mathbb{C}^{2^q} \otimes \mathbb{C}^{2^q}$ such that

$$
\begin{aligned}
O_{\mathcal{QM}} : &|s\rangle \, |a\rangle \, |h\rangle \, |s'\rangle \, |0\rangle \, |0\rangle \\
&\mapsto |s\rangle \, |a\rangle \, |h\rangle \, |s'\rangle \, |\mathrm{Bi}[r_h(s,a)]\rangle \, |\mathrm{Bi}[P_{h|s,a}(s')]\rangle ,
\end{aligned}
\tag{1}
$$

for all $(s, a, h, s') \in \mathcal{S} \times \mathcal{A} \times [H] \times \mathcal{S}$, where $\mathrm{Bi}[r_h(s,a)]$ and $\mathrm{Bi}[P_{h|s,a}(s')]$ denote the binary representation of $r_h(s,a)$ and $P_{h|s,a}(s')$ with precision $2^{-p}$.

We define the number of queries made to the quantum oracle $O_{\mathcal{QM}}$ or classical oracle $O_{\mathcal{M}}$ as the quantum or classical query complexity, respectively. Comparing quantum and classical time complexities can be achieved by examining their respective query complexities, because implementing $O_{\mathcal{QM}}$ has comparable overhead as $O_{\mathcal{M}}$. Specifically, given a Boolean circuit of $O_{\mathcal{M}}$ with $N$ logic gates, it can be converted into a quantum circuit of $O_{\mathcal{QM}}$ with $O(N)$ quantum gates. This conversion can be efficiently achieved by simple conversion rules at the logic gate level (Nielsen & Chuang, 2010). Therefore, $O_{\mathcal{QM}}$ and $O_{\mathcal{M}}$ have comparable costs at the elementary gate level. Then, if the classical oracle $O_{\mathcal{M}}$

---

**Algorithm 1** Quantum Value Iteration **QVI-1**$(\mathcal{M}, \delta)$

---

1: **Require:** MDP $\mathcal{M}$, quantum oracle $O_{\mathcal{QM}}$, maximum failure probability $\delta \in (0,1)$.
2: **Initialize:** $\zeta \leftarrow \delta/(SH)$, $\hat{V}_H \leftarrow \mathbf{0}$.
3: **for** $h := H - 1, \ldots, 0$ **do**
4:      create a quantum oracle $B_{\hat{V}_{h+1}}$ for vector $\hat{V}_{h+1} \in \mathbb{R}^{\mathcal{S}}$
5:      $\forall s \in \mathcal{S}$: create a quantum oracle $B_{\hat{Q}_{h,s}}$ encoding vector $\hat{Q}_{h,s} \in \mathbb{R}^{\mathcal{A}}$ with $O_{\mathcal{QM}}$ and $B_{\hat{V}_{h+1}}$ satisfying
$$\hat{Q}_{h,s}(a) \leftarrow r_h(s,a) + P_{h|s,a}^{\mathrm{T}} \hat{V}_{h+1}$$
6:      $\forall s \in \mathcal{S}$: $\hat{\pi}(s,h) \leftarrow \mathbf{QMS}_\zeta\{\hat{Q}_{h,s}(a) : a \in \mathcal{A}\}$
7:      $\forall s \in \mathcal{S}$: $\hat{V}_h(s) \leftarrow \hat{Q}_{h,s}(\hat{\pi}(s,h))$
8: **end for**
9: **Return:** $\hat{\pi}, \hat{V}_0$

---

can be called in constant time, the quantum oracle $O_{\mathcal{QM}}$ can be called in constant time as well. Under this assumption, query complexity directly reflects the time complexity for both the classical and quantum algorithms.

With the quantum oracle $O_{\mathcal{QM}}$, our objective is to design quantum algorithms that can compute $\pi^*$ and $V_0^*(s)$ for all $s \in \mathcal{S}$ with probability at least $1 - \delta$, while minimizing the total number of queries to $O_{\mathcal{QM}}$.

We first introduce an existing quantum subroutine, quantum maximum searching algorithm (Durr & Hoyer, 1999), which can efficiently find the maximum of a list of unsorted $N \in \mathbb{Z}^+$ numbers using only $O(\sqrt{N})$ queries to that list. In contrast, the best-possible classical algorithm must examine all $N$ elements in the worst case to find the maximum.

**Theorem 3.4** (Quantum maximum searching (Durr & Hoyer, 1999)). *Let $B_f$ be a quantum oracle encoding a vector $f \in \mathbb{R}^N$, $N \in \mathbb{Z}^+$. There exists a quantum maximum searching algorithm, $\mathbf{QMS}$, which, for any $\delta > 0$, can identify an index $i$ such that $f(i)$ is the maximum value in $f$, with a success probability of at least $1 - \delta$. The algorithm requires at most $\tilde{c}\sqrt{N} \log(1/\delta)$ queries to $B_f$, where $\tilde{c} > 0$ is a constant.*

We use $\mathbf{QMS}_\delta\{f(i) : i \in [N]\}$ to denote the process of finding the index of the maximum value of a vector $f$ using $\mathbf{QMS}$, with a success probability at least $1 - \delta$. Note that the classical value iteration algorithm needs to take the maximum over the whole action space in the Bellman recursion to obtain the estimates of optimal V-value function $V_h^*$ and optimal action $\pi^*(s,h)$ for state $s$ at time stage $h$. We incorporate $\mathbf{QMS}$ in this step to reduce the query complexity from $O(A)$ to $O(\sqrt{A})$. Now, we propose our quantum value iteration algorithm **QVI-1** in Algorithm 1. In order to use $\mathbf{QMS}$ correctly, one needs to suitably encode the vector $\hat{V}_{h+1}$ and $\hat{Q}_{h,s}$ with the binary oracles. In summary, **QVI-1** returns an optimal policy and optimal values (Theorem 3.5) but only requires $\tilde{O}(S^2\sqrt{A}H)$ queries to the quantum oracle $O_{\mathcal{QM}}$ (Theorem 3.6). The proof of Theorems 3.5 and

3.6 can be found in Appendix A.3, where we also analyze the cost of the qubit resources of **QVI-1**.

**Theorem 3.5** (Correctness of **QVI-1**). *The outputs $\hat{\pi}$ and $\hat{V}_0$ satisfy that $\hat{\pi} = \pi^*$ and $\hat{V}_0 = V_0^*$ with a success probability at least $1 - \delta$.*

**Theorem 3.6** (Complexity of **QVI-1**). *The quantum query complexity of **QVI-1** in terms of the quantum oracle $O_{\mathcal{QM}}$ is $O(S^2\sqrt{A}H\log(SH/\delta))$.*

### 3.2. Speedup on $S$

Since **QVI-1** achieves a speedup in the action space size $A$, it is advantageous for problems with a large action space, such as natural language processing, where each text in a large dictionary corresponds to a distinct action (Feng et al., 2024). However, in problems modeled by numerous variables, such as Chess or Go, where each position in a vast board is represented as a state, the state space can be much larger than the action space and time horizon (Bellman, 1962). In such scenarios, **QVI-1** may not be suitable due to its complexity of $O(S^2)$. This complexity arises for two reasons: (1) one needs to update $O(S)$ Q-value functions at each time step; (2) computing the "*precise mean*" of the V-value function from the last time step needs $O(S)$ queries to the oracle $O_{\mathcal{QM}}$ when updating each Q-value function. Note that for obtaining an "*$\epsilon$-estimation of the mean*" of $n$ Boolean variables, quantum algorithms only need $\Theta(\min\{\epsilon^{-1}, n\})$ queries to a binary oracle (Nayak & Wu, 1999; Beals et al., 2001). This suggests that a quantum speedup in $S$ may be achievable if one is satisfied with a near-optimal policy. Therefore, next we investigate *whether there exists a quantum algorithm that can obtain $\epsilon$-optimal policies and V-value functions for an MDP $\mathcal{M}$ but only requires $\tilde{O}(S^c poly(\sqrt{A}, H, \epsilon^{-1}))$ queries to $O_{\mathcal{QM}}$, where $0 < c < 2$.*

To achieve this optimization goal, we propose **QVI-2** in Algorithm 2, where the quantum subroutine **QMEBO**, as used in the fifth step, is defined in Algorithm 3. The main difference between **QVI-1** and **QVI-2** is that we compute an estimate of the expectation of $P_{h|s,a}^{\mathrm{T}}\hat{V}_{h+1}$ rather than its precise value in each time step $h$ in **QVI-2**. Since the oracle $O_{\mathcal{QM}}$ that encodes the probability distribution $P_{h|s,a}$ is a binary oracle, we cannot directly apply the existing quantum mean estimation algorithms (Montanaro, 2015), which require an oracle that encodes the probability distribution in the amplitude (See Theorem 4.2). Hence, we design a new quantum subroutine in Algorithm 3, denoted as quantum mean estimation with binary oracles (**QMEBO**).

**Theorem 3.7** (Quantum mean estimation with binary oracles). *Let $\Omega$ be a finite set with cardinality $N$, $p = (p_x)_{x\in\Omega}$ a discrete probability distribution over $\Omega$, and $f : \Omega \to \mathbb{R}$ a function. Suppose we have access to a binary oracle $B_p$ encoding the probability distribution $p$ and a binary oracle $B_f$ encoding the function $f$. If the function $f$ satisfies*

---

**Algorithm 2** Quantum Value Iteration **QVI-2**$(\mathcal{M}, \epsilon, \delta)$

1: **Require:** MDP $\mathcal{M}$, quantum oracle $O_{\mathcal{QM}}$, maximum error $\epsilon \in (0, H]$, failure probability $\delta \in (0, 1)$.
2: **Initialize:** $\zeta \leftarrow \delta/(4\tilde{c}SA^{1.5}H\log(1/\delta))$, $\hat{V}_H \leftarrow \mathbf{0}$.
3: **for** $h := H-1, \ldots, 0$ **do**
4:     create a quantum oracle $B_{\tilde{V}_{h+1}}$ encoding $\tilde{V}_{h+1} \in [0,1]^{\mathcal{S}}$ defined by $\tilde{V}_{h+1} \leftarrow \hat{V}_{h+1}/H$
5:     $\forall s \in \mathcal{S}$: create a quantum oracle $B_{z_{h,s}}$ encoding $z_{h,s} \in \mathbb{R}^{\mathcal{A}}$ defined by
    $z_{h,s}(a) \leftarrow H\cdot\mathbf{QMEBO}_\zeta(P_{h|s,a}^{\mathrm{T}}\tilde{V}_{h+1}, O_{\mathcal{QM}}, B_{\tilde{V}_{h+1}}, \frac{\epsilon}{2H^2}) - \frac{\epsilon}{2H}$
6:     $\forall s \in \mathcal{S}$: create quantum oracle $B_{\hat{Q}_{h,s}}$ encoding $\hat{Q}_{h,s} \in \mathbb{R}^{\mathcal{A}}$ with $O_{\mathcal{QM}}$ and $B_{z_{h,s}}$ satisfying
      $\hat{Q}_{h,s}(a) \leftarrow \max\{r_h(s,a) + z_{h,s}(a), 0\}$
7:     $\forall s \in \mathcal{S}$: $\hat{\pi}(s,h) \leftarrow \mathbf{QMS}_\delta\{\hat{Q}_{h,s}(a) : a \in \mathcal{A}\}$
8:     $\forall s \in \mathcal{S}$: $\hat{V}_h(s) \leftarrow \hat{Q}_{h,s}(\hat{\pi}(s,h))$
9: **end for**
10: **Return:** $\hat{\pi}, \{\hat{V}_h\}_{h=0}^{H-1}$

---

$f(x) \in [0, 1]$ *for all $x \in \Omega$, then the algorithm **QMEBO** requires $O((\frac{\sqrt{N}}{\epsilon} + \sqrt{\frac{N}{\epsilon}})\log(1/\delta))$ queries to $B_p$ and $B_f$ to output an estimate $\hat{\mu}$ of $\mu := \mathbb{E}[f(x)|x \sim p] = p^{\mathrm{T}}f$ such that $Pr(|\tilde{\mu} - \mu| < \epsilon) > 1 - \delta$ for any $\delta > 0$.*

We denote $\mathbf{QMEBO}_\delta(p^{\mathrm{T}}f, B_p, B_f, \epsilon)$ as an estimate of $p^{\mathrm{T}}f$, to an error less than $\epsilon$ with probability at least $1 - \delta$ obtained by **QMEBO**. The key step of **QMEBO** lies in line 4, where a binary oracle $B_p$ is transformed into a unitary oracle $\hat{U}_p$. Unlike $B_p$, $\hat{U}_p$ encodes the information of the probability distribution $p$ in amplitude rather than in quantum state. Using $\hat{U}_p$, we prepare the state $|\psi^{(0)}\rangle$ defined as

$$\frac{1}{\sqrt{N}}\sum_{i=1}^{N}\sqrt{p_i}|i\rangle|0\rangle + \sqrt{\frac{N-1}{N}}\sum_{i=1}^{N}\sqrt{\frac{1-p_i}{N-1}}|i\rangle|1\rangle. \quad (2)$$

The transformation and the required query complexity are presented in Theorem A.3. After encoding the function $f$ in the amplitudes (lines 5-6), the amplitude estimation (Theorem A.5) is applied to compute an estimate $\mu_k$ of $p^{\mathrm{T}}f/N$ with an error of $\epsilon/N$ in the loop $k$. Finally, guaranteed by the Powering lemma (Lemma A.4), the output $\hat{\mu}$ is an $\epsilon$-estimate of $p^{\mathrm{T}}f$ with probability at least $1 - \delta$. The complete version and full analysis of **QMEBO** is presented in Appendix A.4.

With **QMEBO**, it only requires $O(\sqrt{S}/\epsilon)$ queries to the oracle $O_{\mathcal{QM}}$ to obtain an $\epsilon$-estimate of $P_{h|s,a}^{\mathrm{T}}V$ for any $V \in [0,1]^{\mathcal{S}}$. Compared with computing the precise value of $P_{h|s,a}^{\mathrm{T}}V$ with $O(S)$ queries to $O_{\mathcal{QM}}$, **QMEBO** reduces the query complexity from $O(S)$ to $O(\sqrt{S})$. Finally, **QVI-2** only requires $\tilde{O}(S^{1.5}\mathrm{poly}(\sqrt{A}, H, 1/\epsilon))$ queries to $O_{\mathcal{QM}}$ (Theorem 3.9). By suitably controlling the error induced by **QMEBO**, one can ensure that **QVI-2** can obtain $\epsilon$-optimal policies and V-value functions (Theorem 3.8). Note that we subtract the $H$ times error induced by **QMEBO** in line 5

**Algorithm 3** Quantum Mean Estimation with Binary Oracles $\mathbf{QMEBO}_\delta(p^\mathrm{T} f, B_p, B_f, \epsilon)$

---

1: **Require:** $B_p$ encoding a probability distribution $p = (p_i)_{i \in \Omega}$ on a finite set $\Omega$ with cardinality $N$, $B_f$ encoding a function $f = (f_i)_{i \in \Omega}$ where $f_i \in [0, 1]$, maximum error $\epsilon$, maximum failure probability $\delta \in (0, 1)$.

2: **Initialize:** $K = O(\log 1/\delta), T = O(\frac{\sqrt{N}}{\epsilon} + \sqrt{\frac{N}{\epsilon}})$

3: **for** $k \in [K]$ **do**

4:     Prepare state $|\psi^{(0)}\rangle = \hat{U}_p |0\rangle |0\rangle$ using $B_p$

5:     Attach $|0\rangle^{\otimes(q+1)}$ qubits on $|\psi^{(0)}\rangle$ and apply $B_f$
$|\psi^{(1)}\rangle = \frac{1}{\sqrt{N}} \sum_{i=1}^{N} \sqrt{p_i} |i\rangle |0\rangle |\mathrm{Bi}[f_i]\rangle |0\rangle + |\Phi^{(1)}\rangle$

6:     Perform controlled rotation $R_f$ based on $|\mathrm{Bi}[f_i]\rangle$ and revert $B_f$
$|\psi^{(2)}\rangle = \frac{1}{\sqrt{N}} \sum_{i=1}^{N} \sqrt{p_i f_i} |i\rangle |000\rangle + |\Phi^{(2)}\rangle$

7:     Apply $T$ iterations of amplitude estimation with state $|\psi\rangle = |\psi^{(2)}\rangle$, operator $U = 2 |\psi\rangle \langle\psi| - I$, and projector $P = I \otimes |000\rangle \langle000|$ to obtain $\mu_k$

8: **end for**

9: **Return:** $\hat{\mu} = N \cdot \mathrm{Median}(\{\mu_k\}_{k \in [K]})$

---

of **QVI-2** which allows the estimates $z_{h,s}(a)$ to have an one-sided error. This is a variant of *the monotonicity technique* which was originally proposed for solving the infinite-horizon MDPs more efficiently (Sidford et al., 2018). This technique ensures that the value function $\hat{V}_h$ is bounded by the value function of the policy $V_h^{\hat{\pi}}$ at the same time step.

**Theorem 3.8** (Correctness of **QVI-2**). *The outputs $\hat{\pi}$ and $\{\hat{V}_h\}_{h=0}^{H-1}$ satisfy that $V_h^* - \epsilon \leq \hat{V}_h \leq V_h^{\hat{\pi}} \leq V_h^*$ for all $h \in [H]$ with a success probability at least $1 - \delta$.*

**Theorem 3.9** (Complexity of **QVI-2**). *The quantum query complexity of **QVI-2** in terms of the quantum oracle of MDPs $O_{\mathcal{QM}}$ is*

$$O\big(\frac{S^{1.5}\sqrt{A}H^3 \log(SA^{1.5}H/\delta)}{\epsilon}\big). \tag{3}$$

## 4. Generative Model Setting

Even though the exact dynamic model allows precise calculation of optimal policies and values, such a model is not always readily available in a complex environment. In this section, we focus on the generative model setting as studied in (Li et al., 2020). Specifically, we assume that the agent lacks full access to the transition probabilities but can query a generative model to sample transitions for specific state-action pairs. Note that similar models have often been used in the classical setting, where one is assumed to have access to a generative model $G$, which can generate $N$ independent samples for each $(s, a, h) \in \mathcal{S} \times \mathcal{A} \times [H]$ satisfying

$$s_h^i(s, a) \overset{\text{i.i.d.}}{\sim} P_{h|s,a}, \quad i = 1, \dots, N. \tag{4}$$

Correspondingly, we define a quantum generative model $\mathcal{G}$ for an MDP $\mathcal{M}$ in Definition 4.1. It is important to point

out that the quantum state output by $\mathcal{G}$ is similar to a sample drawn from the probability distribution $P_{h|s,a}$ in Eq. (4).

**Definition 4.1** (Quantum generative model of an MDP). The quantum generative model of a time-dependent and finite-horizon MDP $\mathcal{M}$ is a unitary matrix $\mathcal{G} : \mathbb{C}^S \otimes \mathbb{C}^A \otimes \mathbb{C}^H \otimes \mathbb{C}^S \otimes \mathbb{C}^J \to \mathbb{C}^S \otimes \mathbb{C}^A \otimes \mathbb{C}^H \otimes \mathbb{C}^S \otimes \mathbb{C}^J$ satisfying

$$\mathcal{G} : |s\rangle |a\rangle |h\rangle |0\rangle |0\rangle$$
$$\mapsto |s\rangle |a\rangle |h\rangle \left( \sum_{s'} \sqrt{P_{h|s,a}(s')} |s'\rangle |j_{s'}\rangle \right), \tag{5}$$

where $J \geq 0$ is an arbitrary integer and $|j_{s'}\rangle \in \mathbb{C}^J$ are arbitrary auxiliary states.

We define the number of calls that an algorithm makes to the quantum generative model $\mathcal{G}$ or classical generative model $G$ as its quantum or classical sample complexity. Note that in Section 3.1, we have argued that comparing the time complexities of using a quantum oracle and using a classical oracle can be reduced to comparing the quantum and classical query complexities. In this section, although the quantum generative model $\mathcal{G}$ that we use is different from the quantum oracle $O_{\mathcal{QM}}$, the same reduction from time complexity to sample complexity still holds. The reason is that $\mathcal{G}$ and $G$ have similar costs at the elementary gate level, assuming access to the classical circuit implementing $G$. In addition, it only incurs logarithmic overhead for $\mathcal{G}$ to encode the transition probabilities into quantum amplitudes, provided that quantum random access memory (QRAM) (Giovannetti et al., 2008) is available. Finally, the quantum time complexities match the quantum sample complexities up to log factors, provided that these assumptions hold and $G$ can be called in constant time.

We formally state the optimization goals in this setting. For a given time-dependent and finite-horizon MDP $\mathcal{M}$, $\epsilon \in (0, H]$ and $\delta \in (0, 1)$, we want to obtain $\epsilon$-optimal policies, V-value functions and Q-value functions with probability at least $1 - \delta$. With these objectives, we aim to design algorithms that require as few queries to the quantum generative model $\mathcal{G}$ as possible.

Before delving into our algorithms, we first introduce another important quantum subroutine, quantum mean estimation, in Theorem 4.2. Quantum mean estimation consists of two similar quantum algorithms, which are **QME1** and **QME2**. Both of them are referred to as **QME**.

**Theorem 4.2** (Quantum mean estimation (Montanaro, 2015)). *There are two quantum algorithms, denoted as **QME1** and **QME2**, with the following properties. Let $\Omega$ be a finite set, $p = (p_x)_{x \in \Omega}$ a discrete probability distribution over $\Omega$, and $f : \Omega \to \mathbb{R}$ a function. Assume access to a quantum oracle $U_p$ for the probability distribution $p$ satisfying $U_p : |0\rangle |0\rangle \mapsto \sum_{x \in \Omega} \sqrt{p_x} |x\rangle |j_x\rangle$ where $|j_x\rangle$ are arbitrary auxiliary states, as well as an oracle $B_f$ for the function $f$. Then,*

*1. Taking $u, \epsilon > 0$ as additional inputs, along with the assumption that $0 \le f(x) \le u$ for all $x \in \Omega$, **QME1** requires $O\big(\frac{u}{\epsilon} + \sqrt{\frac{u}{\epsilon}}\big)$ queries to $U_p$ and $B_f$,*

*2. Taking $\sigma > 0$ and $\epsilon \in (0, 4\sigma)$ as additional inputs, along with the assumption that $Var[f(x) \mid x \sim p] \le \sigma^2$, **QME2** needs $O\big(\frac{\sigma}{\epsilon} \log^2(\frac{\sigma}{\epsilon})\big)$ queries to $U_p$ and $B_f$,*

*to output an estimate $\tilde{\mu}$ of $\mu = \mathbb{E}[f(x) \mid x \sim p] = p^T f$ satisfying $\Pr(|\tilde{\mu} - \mu| > \epsilon) < 1/3$. Furthermore, by repeating either **QME1** or **QME2** a total of $O(\log(1/\delta))$ times and taking the median of the outputs, one can obtain another estimate $\hat{\mu}$ of $\mu$ such that $\Pr(|\hat{\mu} - \mu| < \epsilon) > 1 - \delta$.*

We denote $\textbf{QME}\{i\}_\delta(p^T f, \epsilon)$ as an estimate of $p^T f$ to an error at most $\epsilon$ with probability at least $1 - \delta$, obtained via **QME**$\{i\}$ for $i \in \{1, 2\}$. Roughly speaking, **QME1** is a quantum version of Hoeffding's inequality, while **QME2** corresponds to the Chebyshev's (or Bernstein's) inequality. For example, for a random variable $X \in [0, u]$, Hoeffding's inequality implies that $O(u^2/\epsilon^2)$ samples are required to obtain an $\epsilon$-estimation of $\mathbb{E}[X]$. In comparison, **QME1** only requires $O(u/\epsilon)$ quantum samples when $\epsilon \in (0, u]$.

Next, we will discuss how to apply the quantum subroutines, **QME** and **QMS**, into the model-free algorithms for finite-horizon MDPs by (Sidford et al., 2023) and (Sidford et al., 2018), and propose two quantum algorithms **QVI-3** and **QVI-4** which have significantly less sample complexity than the SOTA classical algorithms (Li et al., 2020).

### 4.1. Technical Overview of QVI-3

We first briefly review the main idea of the classical algorithm `RandomizedFiniteHorizonVI` proposed in (Sidford et al., 2023). In the standard value iteration algorithm (See Algorithm 6), we initialize $V_H \in \mathbb{R}^{\mathcal{S}}$ with all zero entries and repeatedly apply the Bellman recursion $V_h = \mathcal{T}^h(V_{h+1})$ starting from the last time step and moving backward to the first, where the Bellman value operator $\mathcal{T}^h : \mathbb{R}^{\mathcal{S}} \to \mathbb{R}^{\mathcal{S}}$ is defined as

$$[\mathcal{T}^h(V_{h+1})]_s := \max_{a \in \mathcal{A}}\{r_h(s,a) + P_{h|s,a}^{\mathrm{T}} V_{h+1}\}, \quad (6)$$

for all $s \in \mathcal{S}$. Instead of computing the exact value, they obtain an approximation of $[\mathcal{T}^h(V_{h+1})]_s$ by estimating $P_{h|s,a}^{\mathrm{T}} V_{h+1}$ via sampling from the classical generative model $G$ and taking maximum over the action space $\mathcal{A}$. In order to obtain $\epsilon$-optimal policies and V-value functions, they control the error of estimating $P_{h|s,a}^{\mathrm{T}} V_{h+1}$ to be $\epsilon/H$. If it also holds that $\|V_{h+1}\|_\infty \le H$, then it requires $O\big(SAH^2/(\epsilon^2/H^2)\big) = O(SAH^4/\epsilon^2)$ queries to $G$ at each time step $h$ to obtain the estimates of $P_{h|s,a}^{\mathrm{T}} V_{h+1}$ for all state-action pairs, according to the Hoeffding's inequality. Finally, the classical sample complexity of obtaining near optimal policy and values would be $O(SAH^5/\epsilon^2)$. The

---

**Algorithm 4** Quantum Value Iteration **QVI-3**$(\mathcal{M}, \epsilon, \delta)$

1: **Require:** MDP $\mathcal{M}$, quantum generative model $\mathcal{G}$, maximum error $\epsilon \in (0, H]$, maximum failure probability $\delta \in (0, 1)$.
2: **Initialize:** $\zeta \leftarrow \delta/\big(4\tilde{c}SA^{1.5}H\log(1/\delta)\big)$, $\hat{V}_H \leftarrow \mathbf{0}$.
3: **for** $h := H - 1, \dots, 0$ **do**
4: $\quad$ create a quantum oracle $B_{\hat{V}_{h+1}}$ encoding $\hat{V}_{h+1} \in \mathbb{R}^{\mathcal{S}}$
5: $\quad$ $\forall s \in \mathcal{S}$ : create a quantum oracle $B_{z_{h,s}}$ encoding $z_{h,s} \in \mathbb{R}^{\mathcal{A}}$ with $\mathcal{G}$ and $B_{\hat{V}_{h+1}}$ satisfying
$$z_{h,s}(a) \leftarrow \textbf{QME1}_\zeta\big((P_{h|s,a}^{\mathrm{T}} \hat{V}_{h+1}), \tfrac{\epsilon}{2H}\big) - \tfrac{\epsilon}{2H}$$
6: $\quad$ create a quantum oracle $B_{r_h}$ encoding $r_h \in \mathbb{R}^{\mathcal{S} \times \mathcal{A}}$
7: $\quad$ $\forall s \in \mathcal{S}$ : create a quantum oracle $B_{\hat{Q}_{h,s}}$ encoding $\hat{Q}_{h,s} \in \mathbb{R}^{\mathcal{A}}$ with $B_{r_h}$ and $B_{z_{h,s}}$ satisfying
$$\hat{Q}_{h,s}(a) \leftarrow \max\{r_h(s,a) + z_{h,s}(a), 0\}$$
8: $\quad$ $\forall s \in \mathcal{S} : \hat{\pi}(s,h) \leftarrow \textbf{QMS}_\delta\{\hat{Q}_{h,s}(a) : a \in \mathcal{A}\}$
9: $\quad$ $\forall s \in \mathcal{S} : \hat{V}_h(s) \leftarrow \hat{Q}_{h,s}\big(\hat{\pi}(s,h)\big)$
10: **end for**
11: **Return:** $\hat{\pi}, \{\hat{V}_h\}_{h=0}^{H-1}$

---

sample complexity derived from above informal analysis matches the sample complexity of the algorithm in (Sidford et al., 2023) (up to log-factors).

Now, we show how to achieve speedup in $A, H$ and $\epsilon$ by using the quantum subroutines **QME** and **QMS**. By using quantum mean estimation **QME1**, it only requires $O\big(SA\sqrt{H^2/(\epsilon^2/H^2)}\big) = O(SAH^2/\epsilon)$ queries to the quantum generative oracle $\mathcal{G}$ to obtain $\epsilon$-approximations of $P_{h|s,a}^{\mathrm{T}} V_{h+1}$ for all pairs $(s,a) \in \mathcal{S} \times \mathcal{A}$ at each time step. Hence, the total quantum query complexity in $H$ iterations becomes $O(SAH^3/\epsilon)$. Furthermore, we apply the quantum maximum searching **QMS** in the Bellman recursion to maximize the value on the RHS of Eq. (6). Then, the query complexity further reduces to $O(S\sqrt{A}H^3/\epsilon)$. These are the fundamental ideas of Algorithm 4, denoted as **QVI-3**. In order to correctly apply **QME**, we also apply the monotonicity technique in **QVI-3** by subtracting the error induced by **QME1** so that the V values $\hat{V}_h$ at each time step are bounded in $[0, H]$. Finally, **QVI-3** can obtain not only an $\epsilon$-optimal policy $\hat{\pi}$ but also $\epsilon$-optimal V-value functions $\{\hat{V}_h\}_{h=0}^{H-1}$ (Theorem 4.3) with probability at least $1 - \delta$, which requires only $\tilde{O}(S\sqrt{A}H^3/\epsilon)$ queries to the oracle $\mathcal{G}$ (Theorem 4.4). The rigorous proof of the correctness and complexity of **QVI-3** are provided in Appendix B.1.

**Theorem 4.3** (Correctness of **QVI-3**). *The outputs $\hat{\pi}$ and $\{\hat{V}_h\}_{h=0}^{H-1}$ satisfy that $V_h^* - \epsilon \le \hat{V}_h \le V_h^{\hat{\pi}} \le V_h^*$ for all $h \in [H]$ with a success probability at least $1 - \delta$.*

**Theorem 4.4** (Complexity of **QVI-3**). *The quantum query complexity of **QVI-3** in terms of the quantum generative oracle $\mathcal{G}$ is*

$$O\big(\frac{S\sqrt{A}H^3 \log(SA^{1.5}H/\delta)}{\epsilon}\big). \quad (7)$$

---

**Algorithm 5** Quantum Value Iteration **QVI-4**$(\mathcal{M}, \epsilon, \delta)$

---

1: **Require:** MDP $\mathcal{M}$, quantum generative model $\mathcal{G}$, maximum error $\epsilon \in (0, \sqrt{H}]$, maximum failure probability $\delta \in (0, 1)$.
2: **Initialize:** $K \leftarrow \lceil \log_2(H/\epsilon) \rceil + 1, \zeta \leftarrow \delta/4KHSA, c = 0.001, b = 1$
3: **Initialize:** $\forall h \in [H] : V_{0,h}^{(0)} \leftarrow \mathbf{0}; \forall s \in \mathcal{S}, h \in [H] : \pi_0^{(0)}(s, h) \leftarrow$ arbitrary action $a \in \mathcal{A}$.
4: **for** $k = 0, \ldots, K-1$ **do**
5: $\quad \epsilon_k \leftarrow H/2^k, V_{k,H} \leftarrow \mathbf{0}, V_{k,H}^{(0)} \leftarrow \mathbf{0}$
6: $\quad \forall (s, a, h) \in \mathcal{S} \times \mathcal{A} \times [H] : y_{k,h}(s, a) \leftarrow \max\left\{ \textbf{QME1}_\zeta \left( P_{h|s,a}^{\mathrm{T}} (V_{k,h+1}^{(0)})^2, b \right) - \left( \textbf{QME1}_\zeta (P_{h|s,a}^{\mathrm{T}} V_{k,h+1}^{(0)}, b/H) \right)^2, 0 \right\}$
7: $\quad \forall (s, a, h) \in \mathcal{S} \times \mathcal{A} \times [H] : x_{k,h}(s, a) \leftarrow \textbf{QME2}_\zeta \left( P_{h|s,a}^{\mathrm{T}} V_{k,h+1}^{(0)}, \frac{c\epsilon}{H^{1.5}} \sqrt{y_{k,h}(s, a) + 4b} \right) - \frac{c\epsilon}{H^{1.5}} \sqrt{y_{k,h}(s, a) + 4b}$
8: $\quad$ **for** $h := H-1, \ldots, 0$ **do**
9: $\qquad \forall (s, a) \in \mathcal{S} \times \mathcal{A} : g_{k,h}(s, a) \leftarrow \textbf{QME1}_\zeta \left( P_{h|s,a}^{\mathrm{T}} (V_{k,h+1} - V_{k,h+1}^{(0)}), cH^{-1}\epsilon_k \right) - cH^{-1}\epsilon_k$
10: $\qquad \forall (s, a) \in \mathcal{S} \times \mathcal{A} : Q_{k,h}(s, a) \leftarrow \max\{r_h(s, a) + x_{k,h}(s, a) + g_{k,h}(s, a), 0\}$
11: $\qquad \forall s \in \mathcal{S} : \tilde{V}_{k,h}(s) \leftarrow V_{k,h}(s) \leftarrow [V(Q_{k,h})]_s, \tilde{\pi}_k(s, h) \leftarrow \pi_k(s, h) \leftarrow [\pi(Q_{k,h})]_s$
12: $\qquad \forall s \in \mathcal{S} :$ if $\tilde{V}_{k,h}(s) \leq V_{k,h}^{(0)}(s)$, then $V_{k,h}(s) \leftarrow V_{k,h}^{(0)}(s)$ and $\pi_k(s, h) \leftarrow \pi_k^{(0)}(s, h)$
13: $\quad$ **end for**
14: $\quad \forall h \in [H] : V_{k+1,h}^{(0)} \leftarrow V_{k,h}$ and $\pi_{k+1}^{(0)}(\cdot, h) \leftarrow \pi_k(\cdot, h)$
15: **end for**
16: **Return:** $\hat{\pi} := \pi_{K-1}, \{\hat{V}_h\}_{h=0}^{H-1} := \{V_{K-1,h}\}_{h=0}^{H-1}, \{\hat{Q}_h\}_{h=0}^{H-1} := \{Q_{K-1,h}\}_{h=0}^{H-1}$

---

## 4.2. Technical Overview of QVI-4

Note that **QVI-3** can only obtain $\epsilon$-optimal policy and V-value functions. Below, we will introduce another algorithm, **QVI-4** in Algorithm 5, which can obtain not only $\epsilon$-optimal policies and V-value functions, but also Q-value functions. In this setting, although we can no longer attain a speed-up in $A$, we attain a speed-up in $H$ by utilizing two additional techniques in (Sidford et al., 2018): "*variance reduction*" and "*total variance*".

We now introduce the essential ideas of the two new techniques and show how to integrate these techniques with the quantum mean estimation **QME** to reduce the sample complexity. First, the main idea of variance reduction technique is that, instead of using the standard value iteration algorithm (Algorithm 6) directly for a target approximation error $\epsilon$, one repeats the value iteration algorithm for $K = O(\log(H/\epsilon))$ epochs with decreasing $\epsilon_k$ satisfying $\epsilon_k = \epsilon_{k-1}/2$ and $\epsilon_K = \epsilon$. In each epoch $k$, we obtain $\epsilon_k$-optimal V-value functions $\{V_{k,h}\}_{h=0}^{H-1}$, Q-value functions $\{Q_{k,h}\}_{h=0}^{H-1}$ and policy $\pi_k$. Note that, at the time step $h$ in epoch $k$, the second term on the RHS of Eq. (6) can be rewritten as follows

$$P_{h|s,a}^{\mathrm{T}} V_{k,h+1} = P_{h|s,a}^{\mathrm{T}} (V_{k,h+1} - V_{k,h+1}^{(0)}) + P_{h|s,a}^{\mathrm{T}} V_{k,h+1}^{(0)}, \quad (8)$$

where $V_{k,h+1}^{(0)} \in \mathbb{R}^\mathcal{S}$ is defined as an initial V-value function for the time step $h+1$ from the previous epoch $k-1$. Note that there are a total $SA$ of these equations, each of which is corresponding to a pair $(s, a) \in \mathcal{S} \times \mathcal{A}$. Rather than directly obtaining $\epsilon_k/H$-estimation of $P_{h|s,a}^{\mathrm{T}} V_{k,h+1}$, we instead obtain $\epsilon_k/(2H)$ estimations of both $P_{h|s,a}^{\mathrm{T}} (V_{k,h+1} - V_{k,h+1}^{(0)})$ and $P_{h|s,a}^{\mathrm{T}} V_{k,h+1}^{(0)}$. For the first estimation, if we have

$\mathbf{0} \leq V_{k,h+1} - V_{k,h+1}^{(0)} \leq \tilde{c}\epsilon_k$ for some constant $\tilde{c} > 0$, it can be done up to error $\epsilon_k/(2H)$ using only $O(H^2)$ classical samples or $O(H)$ quantum samples by the Hoeffding's bound or **QME1**, respectively. Similarly, for the second estimation, if it holds that $\mathbf{0} \leq V_{k,h+1}^{(0)} \leq H$, it requires $O(H^4/\epsilon_k^2)$ classical samples or $O(H^2/\epsilon_k)$ quantum samples. The overall classical sample complexity is $O\left( KHSA(H^4/\epsilon_k^2 + H^2) \right) = \tilde{O}(SAH^5/\epsilon_k^2)$, while the quantum sample complexity is $O\left( KHSA(H^2/\epsilon_k + H) \right) = \tilde{O}(SAH^3/\epsilon_k)$. Note that unlike Section 4.1, we do not expect a speedup from $A$ to $\sqrt{A}$ here, since we need to estimate the Q-values for all actions (instead of finding the action with the highest Q-value). Although the variance reduction technique alone does not achieve a speedup in $H$ compared with **QVI-3**, we will see the advantage when combined with the subsequent total variance technique.

The total variance technique stems from the observation that the actual error propagation across time steps is much smaller than previously assumed. Previously, the error in estimating $\mu_{k,h}^{s,a} := P_{h|s,a}^{\mathrm{T}} V_{k,h+1}^{(0)}$ at each time step was set to $\epsilon_k/(2H)$, ensuring that the total error accumulated over $H$ iterations remains bounded by $\epsilon_k/2$. In fact, the per-step error can be further relaxed to $\epsilon_k \sigma_{k,h}^{s,a}/(2H^{1.5})$, where $\sigma_{k,h}^{s,a} := [\sigma_h(V_{k,h+1}^{(0)})](s, a)$. This error value can reach up to $\epsilon_k/(2\sqrt{H})$. As the cumulative standard deviation $\sum_{h=0}^{H-1} \sigma_{k,h}^{s,a}$ is associated with an expression that can be non-trivially upper-bounded by $H^{1.5}$ (Lemma B.2), the total error remains $\epsilon_k/2$. With classical algorithms, $\mu_{k,h}^{s,a}$ can be estimated with an error $\epsilon\sigma_{k,h}^{s,a}$ without explicitly knowing $\sigma_{k,h}^{s,a}$. This requires overall $O\left( SA(\epsilon/H^{1.5})^{-2} \right) = O(SAH^3/\epsilon^2)$ classical samples per time step at each epoch, as guaranteed

by Chebyshev's (or Bernstein's) inequality. When combined with the variance reduction technique in estimating the first term on the RHS of Eq. (8), this approach achieves an overall classical sample complexity of $\tilde{O}(SAH^4/\epsilon^2)$, matching the complexity of the algorithm in (Sidford et al., 2018)[1].

Inspired by (Wang et al., 2021), we can adapt the total variance technique in the quantum setting. The main challenge is that we cannot directly apply the quantum mean estimation **QME2** like its classical counterpart. First, **QME2** cannot estimate $\mu_{k,h}^{s,a}$ to an error of $\epsilon\sigma_{k,h}^{s,a}/(2H^{1.5})$ without prior knowledge of $\sigma_{k,h}^{s,a}$. To address this, we can use **QME1** to obtain an estimate $(\hat{\sigma}_{k,h}^{s,a})^2$ of $(\sigma_{k,h}^{s,a})^2$ with an error $4b > 0$, then use **QME2** to estimate $\mu_{k,h}^{s,a}$ with an error $\epsilon\underline{\sigma}_{k,h}^{s,a}/(2H^{1.5})$, where $\underline{\sigma}_{k,h}^{s,a} := \sqrt{(\hat{\sigma}_{k,h}^{s,a})^2 - 4b} \leq \sigma_{k,h}^{s,a}$, to maintain the correctness. Second, **QME2** also requires upper bounds $C \in \mathbb{R}$ on $\sigma_{k,h}^{s,a}$. Observing that its sample complexity $O(C/\epsilon)$ can be inefficient for large $C$, an ideal way is to use $\overline{\sigma}_{k,h}^{s,a} := \sqrt{(\hat{\sigma}_{k,h}^{s,a})^2 + 4b}$ as $C$. However, this may lead to an unbounded complexity ratio $\big((\hat{\sigma}_{k,h}^{s,a})^2 + 4b\big)/\big((\hat{\sigma}_{k,h}^{s,a})^2 - 4b\big)$. To resolve this, we estimate $\mu_{k,h}^{s,a}$ with an error proportional to $\overline{\sigma}_{k,h}^{s,a}$, ensuring $C/\overline{\sigma}_{k,h}^{s,a} = 1$. Although the correctness may not hold due to $\overline{\sigma}_{k,h}^{s,a} > \sigma_{k,h}^{s,a}$, we can bound $\overline{\sigma}_{k,h}^{s,a} \leq \sigma_{k,h}^{s,a} + \sqrt{7b}$ and suppress the extra error by setting $b$ and the parameter $c$ in **QVI-4** as small constants. Ultimately, **QVI-4** can obtain $\epsilon$-optimal policies, V-value functions and Q-value functions (Theorem 4.5) with $\tilde{O}(SAH^{2.5}/\epsilon)$ queries to the quantum generative oracle $\mathcal{G}$ (Theorem 4.6), which holds for $\epsilon = O(1/\sqrt{H})$. The proof of the correctness and complexity of **QVI-4** is presented in Appendix B.2.

**Theorem 4.5** (Correctness of **QVI-4**). *The outputs $\hat{\pi}$, $\{\hat{V}_h\}_{h=0}^H$ and $\{\hat{Q}_h\}_{h=0}^H$ satisfy that*

$$V_h^* - \epsilon \leq \hat{V}_h \leq V_h^{\hat{\pi}} \leq V_h^*, \tag{9}$$

$$Q_h^* - \epsilon \leq \hat{Q}_h \leq Q_h^{\hat{\pi}} \leq Q_h^*, \tag{10}$$

*for all $h \in [H]$ with a success probability at least $1 - \delta$.*

**Theorem 4.6** (Complexity of **QVI-4**). *The quantum query complexity of **QVI-4** in terms of the quantum generative oracle $\mathcal{G}$ is*

$$O\big(SA(\frac{H^{2.5}}{\epsilon} + H^3)\log^2(\frac{H^{1.5}}{\epsilon})\log(\log(\frac{H}{\epsilon})HSA/\delta)\big). \tag{11}$$

### 4.3. Quantum Lower Bound for Finite-horizon MDPs

We now state the quantum lower bound of the sample complexity for obtaining the $\epsilon$-optimal policy, V-value func-

tions and Q-value functions for a finite-horizon and time-dependent MDP $\mathcal{M}$. Our proof idea is to reduce an infinite-horizon MDP problem to a finite-horizon MDP problem. Specifically, we show that, if there is an algorithm that can obtain an $\epsilon$-optimal V-value function for the finite-horizon MDP, it also can give an $2\epsilon$-optimal V-value function to the infinite-horizon MDP. Therefore, the lower bound of solving finite-horizon MDP with a quantum generative oracle inherits from that of the infinite-horizon MDP. The full analysis is presented in Appendix B.3. Note that our achievable quantum sample complexities of **QVI-3** and **QVI-4** differ from the quantum lower bounds only by a factor of $H$ or $H^{1.5}$, up to logarithmic factors.

**Theorem 4.7** (Lower bounds for finite-horizon MDPs). *Let $\mathcal{S}$ and $\mathcal{A}$ be finite sets of states and actions. Let $H > 0$ be a positive integer and $\epsilon \in (0, 1/2)$ be an error parameter. We consider the following time-dependent and finite-horizon MDP $\mathcal{M} = (\mathcal{S}, \mathcal{A}, \{P_h\}_{h=0}^{H-1}, \{r_h\}_{h=0}^{H-1}, H)$, where $r_h \in [0,1]^{\mathcal{S} \times \mathcal{A}}$ for all $h \in [H]$.*

- *Given access to a classical generative oracle $G$, any algorithm $\mathcal{K}$, which takes $\mathcal{M}$ as an input and outputs $\epsilon$-approximations of $\{Q_h^*\}_{h=0}^{H-1}$ $\{V_h^*\}_{h=0}^{H-1}$ or $\pi^*$ with probability at least $0.9$, must call $G$ at least $\Omega\big(\frac{SAH^3}{\epsilon^2 \log^3(\epsilon^{-1})}\big)$ times on the worst case of input $\mathcal{M}$.*

- *Given access to a quantum generative oracle $\mathcal{G}$, any algorithm $\mathcal{K}$, which takes $\mathcal{M}$ as an input and outputs $\epsilon$-approximations of $\{Q_h^*\}_{h=0}^{H-1}$ with probability at least $0.9$, must call $\mathcal{G}$ at least $\Omega\big(\frac{SAH^{1.5}}{\epsilon \log^{1.5}(\epsilon^{-1})}\big)$ times on the worst case of input $\mathcal{M}$. Besides, any algorithm $\mathcal{K}$ that outputs $\epsilon$-approximations of $\{V_h^*\}_{h=0}^{H-1}$ or $\pi^*$ with probability at least $0.9$ must call $\mathcal{G}$ at least $\Omega\big(\frac{S\sqrt{A}H^{1.5}}{\epsilon \log^{1.5}(\epsilon^{-1})}\big)$ times on the worst case of input $\mathcal{M}$.*

## 5. Conclusion

To the best of our knowledge, this is the first work to rigorously study quantum algorithms for solving "time-dependent" and "finite-horizon" MDPs. In the exact dynamics setting, our quantum value iteration algorithm **QVI-1** achieves a quadratic speedup in the size of the action space ($A$) for computing the optimal policy and V-value function, while **QVI-2** achieves an additional speedup in the size of the state space ($S$) for computing near-optimal policy and V-value functions. Besides, our classical lower bounds show that no classical algorithm can attain comparable query complexities of **QVI-1** and **QVI-2** in terms of the dependences on $S$ and $A$. In the generative model setting, our algorithms **QVI-3** and **QVI-4** achieve speedups in $A$, time horizon ($H$), and approximation error ($\epsilon$) over the SOTA classical algorithm and are asymptotically optimal, up to log terms, for computing near-optimal policies, V-value functions, and Q-value functions, provided a constant time horizon.

---

[1]The result in (Sidford et al., 2018) was originally presented for the time-independent case. We adapt it here for the time-dependent case with an additional factor of $H$.

## Acknowledgements

We especially thank Zongqi Wan for providing insightful guidance on quantum subroutines, including quantum maximum searching (Durr & Hoyer, 1999) and quantum mean estimation algorithms (Montanaro, 2015), and for suggesting helpful references, including (Cornelissen, 2018). The work of John C.S. Lui was supported in part by the RGC SRFS2122-4S02.

## Impact Statement

This paper presents work whose goal is to advance the field of Machine Learning and AI via quantum computing. There are many potential societal consequences of our work, none of which we feel must be specifically highlighted here.

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

# A. Exact Dynamics Setting

## A.1. Classical Algorithm for Finite-horizon MDPs

For the completeness, we restate the classical value iteration (or backward induction) algorithm in (Puterman, 2014).

---

**Algorithm 6** Value Iteration (Backward Induction) Algorithm for Finite Horizon MDPs

---

1: **Require:** MDP $\mathcal{M}$.
2: **Initialize:** $V_H \leftarrow \mathbf{0}$
3: **for** $h := H - 1, \ldots, 0$ **do**
4:    **for** each $s \in \mathcal{S}$ **do**
5:       **for** each $a \in \mathcal{A}$ **do**
6:          $Q_h(s, a) = r_h(s, a) + \sum_{s' \in \mathcal{S}} P_{h|s,a}(s') V_{h+1}(s')$
7:       **end for**
8:       $\pi(s, h) = \underset{a \in \mathcal{A}}{\mathrm{argmax}}\, Q_h(s, a)$
9:       $V_h(s) = Q_h\big(s, \pi(s, h)\big)$
10:    **end for**
11: **end for**
12: **Return:** $\pi, V_0$

---

## A.2. Classical Lower Bounds

**Proof.** We define two sets of hard instances of finite-horizon MDP $M_1$ and $M_2$ which are the same as those in Section 4.1 in (Chen & Wang, 2017). Specifically, suppose that the state space $\mathcal{S}$ can be divided into four parts $\mathcal{S} = \mathcal{S}_U \cup \mathcal{S}_G \cup \mathcal{S}_B \cup \{s_N\}$, where the cardinalities of the sets $\mathcal{S}_U, \mathcal{S}_G$ and $\mathcal{S}_B$ satisfy $S_U = S_G = S_B = \frac{S-1}{3}$, and $s_N$ is a single action. Let the action space be $\mathcal{A} = \mathcal{A}_U \cup \{a_N\}$, where the cardinality of the set $\mathcal{A}_U$ satisfies $A_U = A - 1$ and $a_N$ is a single action. We now construct two sets of MDP instances $M_1$ and $M_2$ that are hard to distinguish.

- Let $M_1$ be the set of instances satisfying the following conditions.

  - $P_h = P \in [0,1]^{\mathcal{S} \times \mathcal{A} \times \mathcal{S}}$ for all $h \in [H]$ and $r_h = r \in [0,1]^{\mathcal{S} \times \mathcal{A}}$, where $H \geq 2$;
  - For any $(s, a)$ satisfies $s \in \mathcal{S}_G \cup \mathcal{S}_B \cup \{s_N\}$ and $a \in \mathcal{A}$, the transition probabilities satisfy $P(s'|s, a) = 1$ if $s' = s$ and $P(s'|s, a) = 0$ if $s' \neq s$, i.e., the states in $\mathcal{S}_G \cup \mathcal{S}_B \cup \{s_N\}$ are absorbing states. Besides, the reward functions satisfy

$$r(s, a) = \begin{cases} 1, & \text{if } s \in \mathcal{S}_G, a \in \mathcal{A} \\ 0, & \text{if } s \in \mathcal{S}_B, a \in \mathcal{A} \\ \frac{1}{2}, & \text{if } s = s_N, a \in \mathcal{A} \end{cases} . \tag{12}$$

  - For any $a \in \mathcal{A}_U$ and $s \in \mathcal{S}_U$, the transition probability satisfies $P(s'|s, a) = 1$ if $s' \in \mathcal{S}_B$ and $P(s'|s, a) = 0$ otherwise, while the reward satisfies $r(s, a) = 0$.
  - For any $a = a_N$ and $s \in \mathcal{S}_U$, the transition probability satisfies $P(s'|s, a) = 1$ if $s' = s_N$ and $P(s'|s, a) = 0$ otherwise, while the reward satisfies $r(s, a) = 0$.

- Let $M_2$ be the set of instances that are different from those in $M_1$ at one state-action pair, which we denote by $(\overline{s}, \overline{a}) \in \mathcal{S}_U \times \mathcal{A}_U$.

  - When $(s, a) = (\overline{s}, \overline{a})$, the transition probability satisfies $P(s'|s, a) = 1$ for some $s' \in \mathcal{S}_G$ and $P(s'|s, a) = 0$ otherwise, while the reward satisfies $r(s, a) = 0$.

From the above definitions, we can know that the cardinalities of $M_1$ and $M_2$ are $|\mathcal{S}_B|^{|\mathcal{S}_U \times \mathcal{A}_U|}$ and $|\mathcal{S}_U \times \mathcal{A}_U| \times |\mathcal{S}_B|^{|\mathcal{S}_U \times \mathcal{A}_U|}$, respectively. We now compute the optimal V-value function $V^*_{0, \mathcal{M}_1}$ for any finite-horizon MDP $\mathcal{M}_1 \in M_1$.

- For $s \in \mathcal{S}_G$, $V^*_{H-1, \mathcal{M}_1}(s) = \max_{a \in \mathcal{A}} \{r(s, a) + P^{\mathrm{T}}_{H|s,a} V^*_{H, \mathcal{M}_1}\} = 1$, because $V^*_{H, \mathcal{M}_1} = \mathbf{0}$. Further, since $s$ is an absorbing state, $V^*_{h, \mathcal{M}_1}(s) = 1 + V^*_{h+1, \mathcal{M}_1}(s) = H - h$. Hence, we can compute $V^*_{0, \mathcal{M}_1}(s) = H$.

- For $s \in \mathcal{S}_B$, since $r(s,a) = 0$ for all $a \in \mathcal{A}$ and $s$ is an absorbing state, we can compute $V^*_{h,\mathcal{M}_1}(s) = 0$ for all $h \in [H]$.

- When $s = s_N$, since $r(s,a) = \frac{1}{2}$ for all $a \in \mathcal{A}$ and $s$ is also an absorbing state, we can compute $V^*_{h,\mathcal{M}_1}(s) = \frac{1}{2} + V^*_{h+1,\mathcal{M}_1}(s) = \frac{H-h}{2}$ and $V^*_{0,\mathcal{M}_1}(s) = \frac{H}{2}$.

- For $s \in \mathcal{S}_U$, we can compute $V^*_{H-1,\mathcal{M}_1}(s) = \max_{a \in \mathcal{A}}\{r(s,a)\} = 0$. Further, by the Bellman optimality equation (Bellman, 1958), we can compute

$$
\begin{aligned}
V^*_{H-2,\mathcal{M}_1}(s) &= \max_{a \in \mathcal{A}}\{r(s,a) + \sum_{s' \in \mathcal{S}} P(s'|s,a)V^*_{H-1,\mathcal{M}_1}(s')\} \\
&= \max\{V^*_{H-1,\mathcal{M}_1}(s)\mathbb{1}\{s \in \mathcal{S}_B\}, V^*_{H-1,\mathcal{M}_1}(s_N)\} \\
&= \max\{0, \frac{1}{2}\}.
\end{aligned}
\tag{13}
$$

The second line comes from the fact that $r(s,a) = 0$ for any $a \in \mathcal{A}$ and state $s$ will transition to $s_N$ if $a = a_N$ or transition to some state $s \in \mathcal{S}B$ if $a \in \mathcal{A}_U$. By induction, we know that $V^*_{h,\mathcal{M}_1}(s) = \max\{V^*_{h+1,\mathcal{M}_1}(s)\mathbb{1}\{s \in \mathcal{S}_B\}, V^*_{h+1,\mathcal{M}_1}(s_N)\} = \max\{0, \frac{H-h+1}{2}\}$ and $V^*_{0,\mathcal{M}_1}(s) = \frac{H-1}{2}$.

Similarly, we can compute the optimal V-value function $V^*_{0,\mathcal{M}_2}$ for any finite-horizon MDP $\mathcal{M}_2 \in M_2$. Since $\mathcal{M}_2$ only differs from $\mathcal{M}_1$ on the state-action pair $(\overline{s}, \overline{a}) \in \mathcal{S}_U \times \mathcal{A}_U$ and the states $s \in \mathcal{S}_G \times \mathcal{S}_B \times \{s_N\}$ are absorbing states, $V^*_{h,\mathcal{M}_2}(s)$ only differs from $V^*_{h,\mathcal{M}_1}(s)$ on state $\overline{s}$. Specifically, $V^*_{H-1,\mathcal{M}_2}(\overline{s}) = \max_{a \in \mathcal{A}}\{r(\overline{s},a)\} = 0$ and

$$
\begin{aligned}
V^*_{h,\mathcal{M}_2}(\overline{s}) &= \max_{a \in \mathcal{A}}\{r(\overline{s},a) + \sum_{s' \in \mathcal{S}} P(s'|\overline{s},a)V^*_{h+1,\mathcal{M}_2}(s')\} \\
&= \max\{V^*_{h+1,\mathcal{M}_2}(s)\mathbb{1}\{s \in \mathcal{S}_B\}, V^*_{h+1,\mathcal{M}_2}(s)\mathbb{1}\{s \in \mathcal{S}_G\}, V^*_{h+1,\mathcal{M}_2}(s_N)\} \\
&= \max\{0, H-h+1, \frac{H-h+1}{2}\} \\
&= H-h+1.
\end{aligned}
\tag{14}
$$

The first line comes from the Bellman optimality equation (Bellman, 1958). The second line comes from the fact that $r(\overline{s},a) = 0$ for all $a \in \mathcal{A}$ and the state $\overline{s}$ will transition to some state $s' \in \mathcal{S}_B$, $s' \in \mathcal{S}_G$ or $s' = s_N$ under the action $a \in \mathcal{A}_U \setminus \{\overline{a}\}$, $a = \overline{a}$ or $a = a_N$. Hence, it implies that $V^*_{0,\mathcal{M}_2}(\overline{s}) = H - 1$. However, $V^*_{0,\mathcal{M}_1}(\overline{s}) = \frac{H-1}{2}$. Therefore, we can see that $\|V^*_{0,\mathcal{M}_1} - V^*_{0,\mathcal{M}_2}\|_\infty = \frac{H-1}{2}$. Using the same proof in Section 5.2 in (Chen & Wang, 2017), we can know that, to achieve $\frac{H-1}{4}$-optimal $V_0$ with high probability, any algorithm must distinguish $\mathcal{M}_1$ from $\mathcal{M}_2$, requiring to search for two discrepancies in an array of size $|\mathcal{S}_U \times \mathcal{A}_U \times \mathcal{S}_B| = \Omega(S^2 A)$ by quering the classical oracle $O_\mathcal{M}$. Therefore, given the classical oracle $O_\mathcal{M}$, the classical lower bound of query complexity for computing an $\epsilon$-optimal $V_0$ for the time-independent and finite-horizon MDP is $\Omega(S^2 A)$ for $\epsilon \in (0, \frac{H-1}{4})$. This implies the classical lower bound of query complexity for obtaining an $\epsilon$-optimal policy or $\epsilon$-optimal V-value functions for the time-dependent and finite-horizon MDP is $\Omega(S^2 A)$. ∎

### A.3. Correctness, Complexity and Qubit Cost of QVI-1 (Algorithm 1)

A.3.1. CORRECTNESS OF **QVI-1** (PROOF OF THEOREM 3.5)

**Proof.** First, we consider the failure probability of the algorithm to achieve above goal. Every **QMS** is performed with maximum failure probability $\zeta = \delta/(SH)$ and **QMS** is called $SH$ times when running Algorithm 1 one time. By the union bound, the probability that there exists an incorrect output is at most $\delta$.

Now, we assume the ideal scenario when **QMS** is always successful to find the action $a^* = \arg\max_{a \in \mathcal{A}} \hat{Q}_{h,s}(a)$, i.e., $\hat{\pi}(s,h) = \arg\max_{a \in \mathcal{A}} \hat{Q}_{h,s}(a)$. Note that we assume $\hat{V}_H(s) = 0$ for all $s \in \mathcal{S}$. Then, we have $\hat{Q}_{H-1,s}(a) = r_{H-1}(s,a)$ for any policy $\pi$, indicating that $\hat{Q}_{H-1,s}(a) = Q^*_{H-1}(s,a) = Q^{\hat{\pi}}_{H-1}(s,a)$.

Assume that with our policy $\hat{\pi}(s,h) = \arg\max_{a \in \mathcal{A}} \hat{Q}_{h,s}(a)$ for all $s \in \mathcal{S}, h \in [H]$, we have $\hat{Q}_{h,s}(a) = Q^*_h(s,a) = Q^{\hat{\pi}}_h(s,a)$ for all $s \in \mathcal{S}, a \in \mathcal{A}, h \in [H]$. Besides, we define $\hat{\pi}_h(a|s)$ as the probability that the agent choose action $a$ in the state $s$ at time $h$. Note that $\hat{\pi}_h(a|s) = 1$ if $a = \hat{\pi}(s,h)$ and $\hat{\pi}_h(a|s) = 0$ otherwise. By Bellman equations, we have

$V_h^{\hat{\pi}}(s) = \sum_{a \in \mathcal{A}} Q_h^{\hat{\pi}}(s,a) \hat{\pi}_h(a|s)$ and $Q_h^{\hat{\pi}}(s,a) = r_h(s,a) + P_{h|s,a}^{\mathrm{T}} V_{h+1}^{\hat{\pi}}$. Then, we can know that, for all $s \in \mathcal{S}$ and $h \in [H]$,

$$
\begin{aligned}
V_h^*(s) &= \max_{\hat{\pi}_h} \max_{\hat{\pi}_{h+1} \cdots \hat{\pi}_{H-1}} \sum_{a \in \mathcal{A}} Q_h^{\hat{\pi}}(s,a) \hat{\pi}_h(a|s) \\
&= \max_{\hat{\pi}_h} \sum_{a \in \mathcal{A}} Q_h^*(s,a) \hat{\pi}_h(a|s) \\
&= Q_h^* \big(s, \hat{\pi}(s,h)\big) \\
&= \hat{Q}_{h,s} \big(\hat{\pi}(s,h)\big) \\
&= \hat{V}_h(s).
\end{aligned}
\tag{15}
$$

Besides, since $Q_h^*(s, \hat{\pi}(s,h)) = Q_h^{\hat{\pi}}(s, \hat{\pi}(s,h))$ by the assumption, then $V_h^*(s) = V_h^{\hat{\pi}}(s)$ for all $s \in \mathcal{S}$ and $h \in [H]$. Similarly, assume that with our policy $\hat{\pi}(s,h) = \arg\max_{a \in \mathcal{A}} \hat{Q}_{h,s}(a)$ for all $s \in \mathcal{S}, h \in [H]$, we have $\hat{V}_h(s) = V_h^*(s) = V_h^{\hat{\pi}}(s)$ for all $s \in \mathcal{S}, h \in [H]$. Then, we have

$$
\begin{aligned}
Q_h^*(s,a) &= r_h(s,a) + \max_{\pi} \sum_{s' \in \mathcal{S}} P_{h|s,a}(s') V_{h+1}^{\pi}(s') \\
&= r_h(s,a) + \sum_{s' \in \mathcal{S}} P_{h|s,a}(s') V_{h+1}^*(s') \\
&= r_h(s,a) + \sum_{s' \in \mathcal{S}} P_{h|s,a}(s') \hat{V}_{h+1}(s') \\
&= \hat{Q}_{h,s}(a).
\end{aligned}
\tag{16}
$$

Note that we also have $\hat{V}_h = V_h^{\hat{\pi}}$ for all $h \in [H]$. Then, it also holds that $Q_h^* = r_h + P_{h|s,a}^{\mathrm{T}} \hat{V}_{h+1} = r_h + P_{h|s,a}^{\mathrm{T}} V_{h+1}^{\hat{\pi}} = Q_h^{\hat{\pi}}$ Since $\hat{Q}_{H-1,s}(a) = Q_{H-1}^*(s,a) = Q_{H-1}^{\hat{\pi}}(s,a)$ for all $s \in \mathcal{S}, a \in \mathcal{A}$, then we can know that $\hat{V}_{H-1}(s) = V_{H-1}^*(s) = V_{H-1}^{\hat{\pi}}(s)$ for all $s \in \mathcal{S}$. Furthermore, since $\hat{V}_{H-1}(s) = V_{H-1}^*(s) = V_{H-1}^{\hat{\pi}}(s)$ holds for all $s \in \mathcal{S}$, then we can deduce that $\hat{Q}_{H-2,s}(a) = Q_{H-2}^*(s,a) = Q_{H-2}^{\hat{\pi}}(s,a)$ for all $s \in \mathcal{S}$ and $a \in \mathcal{A}$. In the end, we can conclude that $\hat{V}_0(s) = V_0^*(s) = V_0^{\hat{\pi}}(s)$ for all $s \in \mathcal{S}$ which implies $\hat{\pi}$ is an optimal policy. ■

### A.3.2. COMPLEXITY OF **QVI-1** (PROOF OF THEOREM 3.6)

**Proof.** We first assume that all **QMS** are successful to find the optimal actions, up to the specified error, because the probability that this does not hold is at most $\delta$. Let $C$ be the complexity of **QVI-1**$(\mathcal{M}, \delta)$ as if all **QMS** are carried out with maximum failure probabilities set to constant. Then, since the actual maximum failure probabilities are set to $\zeta = \delta/(SH)$, the actual complexity of **QVI-1**$(\mathcal{M}, \delta)$ is

$$
O\big(C \log(SH/\delta)\big). \tag{17}
$$

Now, we check each line of **QVI-1**$(\mathcal{M}, \delta)$ to bound $C$.

In line 4, we encode the vector $\hat{V}_{h+1}$ to an oracle $B_{\hat{V}_{h+1}}$. This process does not need to query $O_{\mathcal{QM}}$ and only needs to access the classical vector $\hat{V}_{h+1}$. Therefore, the query complexity of $B_{\hat{V}_{h+1}}$ in terms of $O_{\mathcal{QM}}$ is $O(1)$.

In line 5, we need to construct the quantum oracle $B_{\hat{Q}_{h,s}}$ with $O_{\mathcal{QM}}$. Since we need to obtain $|P_{h|s,a}(s')\rangle$ for all $s' \in \mathcal{S}$ and calculate the weighted sum $|\sum_{s' \in \mathcal{S}} P_{h|s,a}(s') \hat{V}_{h+1}(s')\rangle$, it requires $O(S)$ query cost of the oracle $O_{\mathcal{QM}}$. Note that the quantum addition and quantum multiplication can be performed by various quantum circuits, such as quantum Fourier transform techniques (Ruiz-Perez & Garcia-Escartin, 2017; Draper, 2000). Therefore, the query complexity of $B_{\hat{Q}_{h,s}}$ in terms of $O_{\mathcal{QM}}$ is $O(S)$.

In line 6, we can use quantum maximum searching algorithm **QMS** in Theorem 3.4, resulting in a query cost of order $O(\sqrt{A})$ to the oracle $B_{\hat{Q}_{h,s}}$ for all $s \in \mathcal{S}$ in each loop $h \in [H]$.

Therefore, it induces an overall query cost of $C = O(S^2 \sqrt{A} H)$ to the oracle $O_{\mathcal{QM}}$. Combining with (17), the overall

quantum query complexity of **QVI-1**$(\mathcal{M}, \delta)$ in terms of $O_{\mathcal{QM}}$ is

$$O\big(S^2\sqrt{A}H\log(SH/\delta)\big). \tag{18}$$

∎

### A.3.3. ANALYSIS ON THE COST OF QUBIT RESOURCES

Considering that qubits are still scarce resources in a quantum computer, it is necessary to minimize the qubits resources required in a quantum algorithm. Note that the line 5 of Algorithm 1 is the main source of consuming qubits in the whole algorithm. Constructing the oracle $B_{\hat{Q}_{h,s}}$ for all $s \in \mathcal{S}$ and $h \in [H]$ requires a large number of auxiliary qubits. This is because the process involves storing information about the vector $\hat{V}_{h+1} \in \mathbb{R}^S$ and the transition probabilities $P_{h|s,a}(s')$ for all $s' \in \mathcal{S}$, which are obtained by querying the quantum oracles $B_{\hat{V}_{h+1}}$ and $O_{\mathcal{QM}}$. After encoding the classical information into qubits, we need to compute the weighted sum $\sum_{s' \in \mathcal{S}} P_{h|s,a}(s')\hat{V}_{h+1}(s')$. This process requires (non-modular) quantum adder(Ruiz-Perez & Garcia-Escartin, 2017; Draper, 2000; Vedral et al., 1996) and (non-modular) quantum multiplier (Ruiz-Perez & Garcia-Escartin, 2017; Vedral et al., 1996) to compute the additions and multiplications with additional auxiliary qubits. Specifically, with the fixed-point representation, (Ruiz-Perez & Garcia-Escartin, 2017) constructs quantum circuits of a non-modular quantum adder $U_{\mathrm{qAdd}} : |\mathrm{Bi}[a]\rangle_q |\mathrm{Bi}[b]\rangle_{q+1} \mapsto |\mathrm{Bi}[a]\rangle_q |\mathrm{Bi}[a] + \mathrm{Bi}[b]\rangle_{q+1}$ and a non-modular quantum multiplier $U_{\mathrm{qMul}} : |\mathrm{Bi}[a]\rangle_q |\mathrm{Bi}[b]\rangle_q |0\rangle_{2q} \mapsto |\mathrm{Bi}[a]\rangle_q |\mathrm{Bi}[b]\rangle_q |\mathrm{Bi}[a]\mathrm{Bi}[b]\rangle_{2q}$, which can compute the non-modular sum and multiplication of two non-negative real numbers $a$ and $b$. We refer readers to (Wang et al., 2024) for a comprehensive overview of the existing work on quantum arithmetic circuits.

Inspired by the quantum circuit for computing the controlled weighted sum proposed in Section 8 in (Ruiz-Perez & Garcia-Escartin, 2017), we design a QFT-based circuit to reduce the qubits consumption in constructing the oracle $B_{\hat{Q}_{h,s}}$. We first prepare the following qubits

$$|a\rangle |s\rangle |h\rangle |0\rangle^{\otimes 4q+q_s+1}, \tag{19}$$

where $q_s = \lceil \log_2(S) \rceil$. Then we apply the rotation matrix $U_{s'}$ to transform the $|0\rangle$ to the target state $|s'\rangle$. Since we encode the state space into orthonormal bases, then $U_{s'}$ is unitary. Hence, it returns the output state

$$|a\rangle |s\rangle |h\rangle |s'\rangle |0\rangle^{\otimes 4q+1}. \tag{20}$$

By applying the unitary oracle $O_{\mathcal{QM}}$ and $B_{\hat{V}_{h+1}}$, we can obtain the following state

$$|a\rangle |s\rangle |h\rangle |s'\rangle |\mathrm{Bi}[r_h(s,a)]\rangle |\mathrm{Bi}[P_{h|s,a}(s')]\rangle |\mathrm{Bi}[\hat{V}_{h+1}(s')]\rangle |0\rangle^{\otimes q+1}. \tag{21}$$

Then we compute the quantum Fourier transform of $|0\rangle^{\otimes q+1}$ where

$$\mathrm{QFT} |0\rangle^{\otimes q+1} = \frac{1}{\sqrt{2^{q+1}}} \sum_{k=0}^{2^{q+1}-1} e^{i\frac{2\pi 0 k}{2^{q+1}}} |k\rangle = |\phi(0)\rangle, \tag{22}$$

we obtain the output state

$$|a\rangle |s\rangle |h\rangle |s'\rangle |\mathrm{Bi}[r_h(s,a)]\rangle |\mathrm{Bi}[P_{h|s,a}(s')]\rangle |\mathrm{Bi}[\hat{V}_{h+1}(s')]\rangle |\phi(0)\rangle. \tag{23}$$

Then we apply the multiplication block $U_{2^{-p}P_{h|s,a}(s')\hat{V}_{h+1}(s')}$ defined in the Fig. 4 in (Ruiz-Perez & Garcia-Escartin, 2017) and obtain the output state

$$|a\rangle |s\rangle |h\rangle |s'\rangle |\mathrm{Bi}[r_h(s,a)]\rangle |\mathrm{Bi}[P_{h|s,a}(s')]\rangle |\mathrm{Bi}[\hat{V}_{h+1}(s')]\rangle |\phi(0 + \mathrm{Bi}[P_{h|s,a}(s')]\mathrm{Bi}[\hat{V}_{h+1}(s')])\rangle. \tag{24}$$

By applying the unitary matrix $B_{\hat{V}_{h+1}}^{\dagger}$, $O_{\mathcal{QM}}^{\dagger}$ and $U_{s'}^{\dagger}$ in sequence, we can undo the operations on auxiliary qubits and obtain the following state

$$|a\rangle |s\rangle |h\rangle |0\rangle^{\otimes 3q+q_s} |\phi(0 + \mathrm{Bi}[P_{h|s,a}(s')]\mathrm{Bi}[\hat{V}_{h+1}(s')])\rangle. \tag{25}$$

We can repeat the above operations for all $s' \in \mathcal{S}$ and obtain

$$|a\rangle |s\rangle |h\rangle |0\rangle^{\otimes 3q+q_s} |\phi(0 + \sum_{s' \in \mathcal{S}} \mathrm{Bi}[P_{h|s,a}(s')]\mathrm{Bi}[\hat{V}_{h+1}(s')])\rangle. \tag{26}$$

By applying the inverse quantum Fourier transform on the state $|\phi(0 + \sum_{s' \in \mathcal{S}} \text{Bi}[P_{h|s,a}(s')]\text{Bi}[\hat{V}_{h+1}(s')])\rangle$, we can obtain

$$|a\rangle |s\rangle |h\rangle |0\rangle^{\otimes 3q + q_s} | \sum_{s' \in \mathcal{S}} \text{Bi}[P_{h|s,a}(s')]\text{Bi}[\hat{V}_{h+1}(s')]\rangle. \tag{27}$$

Since $\sum_{s' \in \mathcal{S}} P_{h|s,a}(s') = 1$ and $P_{h|s,a}(s') \in [0,1]$ for all $s' \in \mathcal{S}$, there is no overflow when computing the weighted sum. Hence, the weighted sum is non-modular. Further, we apply the rotation matrix $U_{s'}$ and the oracle $O_{\mathcal{QM}}$ in sequence to obtain the state

$$|a\rangle |s\rangle |h\rangle |s'\rangle |\text{Bi}[r_h(s,a)]\rangle |\text{Bi}[P_{h|s,a}(s')]\rangle |0\rangle^{\otimes q} | \sum_{s' \in \mathcal{S}} \text{Bi}[P_{h|s,a}(s')]\text{Bi}[\hat{V}_{h+1}(s')]\rangle. \tag{28}$$

Then we apply the quantum adder $U_{\text{qAdd}}$ to obtain the state

$$|a\rangle |s\rangle |h\rangle |s'\rangle |\text{Bi}[r_h(s,a)]\rangle |\text{Bi}[P_{h|s,a}(s')]\rangle |0\rangle^{\otimes q} |\text{Bi}[r_h(s,a)] + \sum_{s' \in \mathcal{S}} \text{Bi}[P_{h|s,a}(s')]\text{Bi}[\hat{V}_{h+1}(s')]\rangle. \tag{29}$$

Since there are $q + 1$ qubits in the last register in (28), then the sum of $\text{Bi}[r_h(s,a)]$ and $\sum_{s' \in \mathcal{S}} \text{Bi}[P_{h|s,a}(s')]\text{Bi}[\hat{V}_{h+1}(s')]$ has no overflow and the result is non-modular. Therefore, by applying $O_{\mathcal{QM}}^{\dagger}$ and $U_{s'}^{\dagger}$ in sequence, we can obtain the state

$$|a\rangle |s\rangle |h\rangle |0\rangle^{\otimes 3q + q_s} |\text{Bi}[r_h(s,a)] + \sum_{s' \in \mathcal{S}} \text{Bi}[P_{h|s,a}(s')]\text{Bi}[\hat{V}_{h+1}(s')]\rangle. \tag{30}$$

**Remark:** Since the above operations are unitary, then the oracle $B_{\hat{Q}_{h,s}}$ constructed in this way is unitary. Instead of preparing $|\text{Bi}[P_{h|s,a}(1)]\rangle |\text{Bi}[\hat{V}_{h+1}(1)]\rangle \cdots |\text{Bi}[P_{h|s,a}(S)]\rangle |\text{Bi}[\hat{V}_{h+1}(S)]\rangle |0\rangle^{\otimes q+1}$ and computing the weighted sum on the last register, which requires $q(2S + 1) + 1$ qubits as shown in Section 8 in (Ruiz-Perez & Garcia-Escartin, 2017), our method significantly reduces the number of required qubits, needing only $3q + 1$ qubits to compute the weighted sum. The main idea is to reuse the $2q$ auxiliary qubits $|\text{Bi}[P_{h|s,a}(s')]\rangle |\text{Bi}[\hat{V}_{h+1}(s')]\rangle$ by leveraging the invertible property of the unitary matrices $O_{\mathcal{QM}}$ and $B_{\hat{V}_{h+1}}$ for all $h \in [H]$. However, this comes at the cost of an additional $S$ queries to $O_{\mathcal{QM}}^{\dagger}$. We summarize the above results in creating the oracle $B_{\hat{Q}_{h,s}}$ in the following Theorem A.1.

**Theorem A.1** (Number of Qubits Required for the Construction of Oracle $B_{\hat{Q}_{h,s}}$). *The total number of qubits required for creating a quantum oracle $B_{\hat{Q}_{h,s}}$ for each $h \in [H]$ and $s \in \mathcal{S}$ in Algorithm 1 is $4q + 2q_s + q_h + q_a + 1$, among which $3q + 2q_s + q_h$ are auxiliary qubits, not counting the auxiliary qubits necessary to implement the oracle $O_{\mathcal{QM}}$ and $B_{\hat{V}_{h+1}}$ for all $h \in [H]$, where $q_s = \lceil \log_2 S \rceil$, $q_a = \lceil \log_2 A \rceil$ and $q_h = \lceil \log_2 H \rceil$.*

**Theorem A.2** (Number of Qubits Required for **QVI-1** (Algorithm 1)). *The total number of qubits required for Algorithm 1 is $11q + 4q_s + 2q_h + 4q_a + 2$, not counting the auxiliary qubits necessary to implement the oracle $O_{\mathcal{QM}}$ and $B_{\hat{V}_{h+1}}$ for all $h \in [H]$, where $q_s = \lceil \log_2 S \rceil$, $q_a = \lceil \log_2 A \rceil$ and $q_h = \lceil \log_2 H \rceil$.*

**Proof.** In line 6 of Algorithm 1, we apply **QMS** algorithm to achieve a quadratic speedup in searching the optimal action which achieves maximum value of the vector $\hat{Q}_{h,s}$ for all $s \in \mathcal{S}$ and $h \in [H]$. In this **QMS** algorithm, it requires an oracle $O_{\text{QMS}}$ to mark the indexes $a$ of the vector $\hat{Q}_{h,s}$ which satisfy $\hat{Q}_{h,s}(a) > \hat{Q}_{h,s}(a')$, where $a'$ is a threshold index, by flipping the phase of the $|a\rangle$. Therefore, the oracle $O_{\text{QMS}}$ is defined as

$$O_{\text{QMS}} : |a\rangle |a'\rangle |-\rangle \mapsto (-1)^{f_{a'}(a)} |a\rangle |a'\rangle |-\rangle, \tag{31}$$

where $f_{a'}(a) = 1$ if $\hat{Q}_{h,s}(a) > \hat{Q}_{h,s}(a')$ and $f_{a'}(a) = 0$ otherwise, and $|-\rangle = \frac{1}{\sqrt{2}}(|0\rangle - |1\rangle)$. Figure 8 in (Oliveira & Ramos, 2007) showed a way to construct the corresponding unitary quantum circuit of the oracle $O_{\text{QMS}}$ based on the quantum bit string comparator (QBSC) $U_{\text{QBSC}}$. Given two real number $a$ and $b$, $U_{\text{QBSC}}$ works as

$$U_{\text{QBSC}} : |\text{Bi}[a]\rangle |\text{Bi}[b]\rangle |0\rangle^{\otimes 3(q-1)} |0\rangle |0\rangle \mapsto |\text{Bi}[a]\rangle |\text{Bi}[b]\rangle |\psi\rangle |x\rangle |y\rangle, \tag{32}$$

where $|\psi\rangle$ is a $3(q-1)$-qubit garbage state and the last two qubits store the comparison result. Specifically, we define that if $a = b$ then $x = y = 0$, if $a > b$ then $x = 1$ and $y = 0$, and if $a < b$ then $x = 0$ and $y = 1$. We restate the construction process under the background of our algorithm here. The first step to construct $O_{\text{QMS}}$ is to prepare the following qubits

$$|a\rangle |0\rangle^{\otimes q} |a'\rangle |0\rangle^{\otimes q} |0\rangle^{\otimes 3(q-1)} |0\rangle |0\rangle |-\rangle. \tag{33}$$

Then we apply $B_{\hat{Q}_{h,s}}$ created in line 5 to the first and third register and obtain the following state

$$|a\rangle \, |\text{Bi}[\hat{Q}_{h,s}(a)]\rangle \, |a'\rangle \, |\text{Bi}[\hat{Q}_{h,s}(a')]\rangle \, |0\rangle^{\otimes 3(q-1)} \, |0\rangle \, |0\rangle \, |-\rangle. \tag{34}$$

Then we apply $U_{\text{QBSC}}$ to compare $\hat{Q}_{h,s}(a)$ and $\hat{Q}_{h,s}(a')$

$$|a\rangle \, |\text{Bi}[\hat{Q}_{h,s}(a)]\rangle \, |a'\rangle \, |\text{Bi}[\hat{Q}_{h,s}(a')]\rangle \, |\psi\rangle \, |x\rangle \, |y\rangle \, |-\rangle. \tag{35}$$

Further, we apply the controlled unitary matrix $U_c = (I \otimes \sigma_1 \otimes I)T(I \otimes \sigma_1 \otimes I)$ to the last three qubits and obtain the following state

$$(-1)^{x(1-y)} \, |a\rangle \, |\text{Bi}[\hat{Q}_{h,s}(a)]\rangle \, |a'\rangle \, |\text{Bi}[\hat{Q}_{h,s}(a')]\rangle \, |\psi\rangle \, |x\rangle \, |y\rangle \, |-\rangle, \tag{36}$$

where $\sigma_1$ is the Pauli-X gate and $T$ is the Tofolli gate. By applying $U_{\text{QBSC}}^\dagger$ and $B_{\hat{Q}_{h,s}}^\dagger$, we can undo the operations on $|\text{Bi}[\hat{Q}_{h,s}(a)]\rangle$ and $|\text{Bi}[\hat{Q}_{h,s}(a')]\rangle$ and obtain the following state

$$(-1)^{x(1-y)} \, |a\rangle \, |0\rangle^{\otimes q} \, |a'\rangle \, |0\rangle^{\otimes q} \, |0\rangle^{\otimes 3(q-1)} \, |0\rangle \, |0\rangle \, |-\rangle. \tag{37}$$

From the above steps, we can see that the construction of the $O_{\text{QMS}}$ requires one query to $B_{\hat{Q}_{h,s}}$ and one query to $B_{\hat{Q}_{h,s}}^\dagger$ as shown in (34) and (37). By Theorem A.1, we know that it requires $4q + 2q_s + q_h + q_a + 1$ qubits to construct the oracle $B_{\hat{Q}_{h,s}}$. Then it requires $2(4q + 2q_s + q_h + q_a + 1) + 2q_a + 3(q-1) + 3 = 11q + 4q_s + 2q_h + 4q_a + 2$ qubits to construct the oracle $O_{\text{QMS}}$, among which $2(4q + 2q_s + q_h + q_a + 1) + 3(q-1) + 2 = 11q + 4q_s + 2q_h + 2q_a + 1$ are auxiliary qubits. ∎

## A.4. Correctness and Complexity of QVI-2 (Algorithm 2)

### A.4.1. PROOF OF THEOREM A.3

**Theorem A.3.** *Let $\Omega$ be a finite set with cardinality $N$, $p = (p_x)_{x \in \Omega}$ a discrete probability distribution over $\Omega$. Suppose we have access to a binary oracle $B_p : |i\rangle \, |0\rangle \mapsto |i\rangle \, |Bi[p_i]\rangle$. By using $O(1)$ invocations of the oracle $B_p$ and $B_p^\dagger$, we can implement a unitary oracle $\hat{U}_p : \mathbb{C}^N \otimes \mathbb{C}^2 \to \mathbb{C}^N \otimes \mathbb{C}^2$ satisfying*

$$\hat{U}_p : |i\rangle \, |0\rangle \mapsto \frac{1}{\sqrt{N}} \sum_{i=1}^{N} \sqrt{p_i} \, |i\rangle \, |0\rangle + \sqrt{\frac{N-1}{N}} \sum_{i=1}^{N} \sqrt{\frac{1-p_i}{N-1}} \, |i\rangle \, |1\rangle. \tag{38}$$

**Proof.** First, we need to create the uniform superposition by applying Hadamard gates and query oracle $B_p$

$$|i\rangle \, |0\rangle \xrightarrow{H^{\otimes n}} \frac{1}{\sqrt{N}} \sum_{i=1}^{N} |i\rangle \, |0\rangle \xrightarrow{B_p} \frac{1}{\sqrt{N}} \sum_{i=1}^{N} |i\rangle \, |\text{Bi}[p_i]\rangle. \tag{39}$$

Second, we add a single auxiliary qubit and perform a controlled rotation $R_p$ based on the value stored in $|\text{Bi}[p_i]\rangle$ defined as $R_p : |\text{Bi}[p_i]\rangle \, |0\rangle \mapsto |\text{Bi}[p_i]\rangle \, (\sqrt{p_i} \, |0\rangle + \sqrt{1-p_i} \, |1\rangle)$:

$$\xrightarrow{I \otimes R_p} \frac{1}{\sqrt{N}} \sum_{i=1}^{N} |i\rangle \, |\text{Bi}[p_i]\rangle \, \left(\sqrt{p_i} \, |0\rangle + \sqrt{1-p_i} \, |1\rangle\right). \tag{40}$$

Third, we undo the oracle $B_p$ and drop the auxilliary qubit $|0\rangle$ in Eq. $(a)$ to obtain the desired result.

$$\begin{aligned}
&\xrightarrow{B_p^\dagger} \frac{1}{\sqrt{N}} \sum_{i=1}^{N} |i\rangle \, |0\rangle \, (\sqrt{p_i} \, |0\rangle + \sqrt{1-p_i} \, |1\rangle) \\
&\stackrel{(a)}{=} \frac{1}{\sqrt{N}} \sum_{i=1}^{N} \sqrt{p_i} \, |i\rangle \, |0\rangle + \frac{1}{\sqrt{N}} \sum_{i=1}^{N} \sqrt{1-p_i} \, |i\rangle \, |1\rangle \\
&= \frac{1}{\sqrt{N}} \sum_{i=1}^{N} \sqrt{p_i} \, |i\rangle \, |0\rangle + \sqrt{\frac{N-1}{N}} \sum_{i=1}^{N} \sqrt{\frac{1-p_i}{N-1}} \, |i\rangle \, |1\rangle.
\end{aligned} \tag{41}$$

**Lemma A.4** (Powering lemma (Jerrum et al., 1986)). *Let $\mathcal{K}$ be a classical or quantum algorithm designed to estimate a quantity $\mu$, where its output $\tilde{\mu}$ satisfies $|\mu - \tilde{\mu}| \leq \epsilon$ with probability at least $1 - \gamma$, for some fixed $\gamma < 1/2$. Then, for any $\delta > 0$, by repeating $\mathcal{K}$ $O(\log(1/\delta))$ times and taking the median of the outputs, one can obtain an estimate $\hat{\mu}$ such that $|\hat{\mu} - \mu| < \epsilon$ with probability at least $1 - \delta$.*

**Theorem A.5** (Amplitude estimation (Brassard et al., 2002)). *The amplitude estimation algorithm is designed to estimate the amplitude $a = \langle \psi | P | \psi \rangle \in [0, 1]$ of a quantum state $|\psi\rangle$. It takes the following inputs, a quantum state $|\psi\rangle$, two unitary operators: $U = 2 |\psi\rangle \langle \psi| - I$ and $V = I - 2P$, where $P$ is some suitable projector, and an integer $T$, which determines the number of repetitions. The algorithm outputs an estimate $\tilde{a} \in [0, 1]$ for the amplitude $a$. The estimate satisfies the error bound:*

$$|\tilde{a} - a| \leq 2\pi \frac{\sqrt{a(1-a)}}{T} + \frac{\pi^2}{T^2}, \tag{42}$$

*with a success probability of at least $8/\pi^2$. To achieve this, the unitary operators $U$ and $V$ are applied $T$ times each.*

### A.4.2. COMPLETE VERSION OF QUANTUM MEAN ESTIMATION WITH BINARY ORACLE **QMEBO**

In Section 3, we provide a simplified version of **QMEBO** by hiding the details of some auxiliary states and operators. For clarity, we provide a complete version in Algorithm 7. Based the Algorithm 7, the auxiliary state in line 5 of **QMEBO** in Algorithm 3 should be

$$|\Phi^{(1)}\rangle = \frac{1}{\sqrt{N}} \sum_{i=1}^{N} \sqrt{1 - p_i} |i\rangle |1\rangle |\mathbf{Bi}[f_i]\rangle |0\rangle, \tag{43}$$

and the auxiliary state in line 6 satisfies

$$|\Phi^{(2)}\rangle = \frac{1}{\sqrt{N}} \sum_{i=1}^{N} \sqrt{p_i(1 - f_i)} |i\rangle |001\rangle + \frac{1}{\sqrt{N}} \sum_{i=1}^{N} \sqrt{1 - p_i} |i\rangle |1\rangle |0\rangle \left( \sqrt{f_i} |0\rangle + \sqrt{1 - f_i} |1\rangle \right) \tag{44}$$

### A.4.3. PROOF OF THEOREM 3.7

**Proof.** We first show the correctness of Algorithm 3. Note that we obtain

$$|\psi^{(2)}\rangle = \frac{1}{\sqrt{N}} \sum_{i=1}^{N} \sqrt{p_i} |i\rangle |0\rangle |0\rangle \left( \sqrt{f_i} |0\rangle + \sqrt{1 - f_i} |1\rangle \right) + \sqrt{\frac{N-1}{N}} |\Phi\rangle, \tag{45}$$

where $|\Phi\rangle = \sum_{i=1}^{N} \sqrt{\frac{1 - p_i}{N - 1}} |i\rangle |1\rangle |0\rangle \left( \sqrt{f_i} |0\rangle + \sqrt{1 - f_i} |1\rangle \right)$. Besides, we have

$$\langle \psi^{(2)} | P | \psi^{(2)} \rangle = \frac{1}{N} \sum_{i=1}^{N} p_i f_i = \frac{1}{N} E[f(x) | x \sim p] = \frac{1}{N} \mu, \tag{46}$$

where $P = I \otimes |000\rangle \langle 000|$. Hence, by Theorem A.5, we know that we can obtain $\mu_k$ in each loop $k \in [K]$ such that

$$\left| \mu_k - \frac{1}{N} \sum_{i=1}^{N} p_i f_i \right| \leq 2\pi \frac{\sqrt{\frac{\mu}{N} \left( 1 - \frac{\mu}{N} \right)}}{T} + \frac{\pi^2}{T^2}, \tag{47}$$

with probability at least $8/\pi^2$. Let $\hat{\mu} = N \cdot \tilde{\mu}$, where $\tilde{\mu} = \text{Median}(\mu_0, \ldots, \mu_{K-1})$. By Lemma A.4, we know that $\tilde{\mu} = \hat{\mu}/N$ satisfies

$$\left| \frac{\hat{\mu}}{N} - \frac{1}{N} \sum_{i=1}^{N} p_i f_i \right| \leq 2\pi \frac{\sqrt{\frac{\mu}{N} \left( 1 - \frac{\mu}{N} \right)}}{T} + \frac{\pi^2}{T^2}, \tag{48}$$

---

**Algorithm 7** Quantum Mean Estimation with Binary Oracles $\mathbf{QMEBO}_\delta(p^{\mathrm{T}}f, B_p, B_f, \epsilon)$

---

1: **Require:** $B_p$ encoding a probability distribution $p = (p_i)_{i \in \Omega}$ on a finite set $\Omega$ with cardinality $N$, $B_f$ encoding a function $f = (f_i)_{i \in \Omega}$ where $f_i \in [0, 1]$, maximum error $\epsilon$, maximum failure probability $\delta \in (0, 1)$.

2: **Output:** $\hat{\mu}$ satisfying $|\hat{\mu} - p^{\mathrm{T}}f| \leq \epsilon$

3: **Initialize:** $K = O\big(\log(1/\delta)\big), T = O\Big(\frac{\sqrt{N}}{\epsilon} + \sqrt{\frac{N}{\epsilon}}\Big)$

4: **for** $k \in [K]$ **do**

5:   create a quantum oracle $\hat{U}_p$ with $B_p$ and obtain the following state $|\psi^{(0)}\rangle = \hat{U}_p |0\rangle |0\rangle$:

$$|\psi^{(0)}\rangle = \hat{U}_p |0\rangle |0\rangle = \frac{1}{\sqrt{N}} \sum_{i=1}^{N} \sqrt{p_i} |i\rangle |0\rangle + \sqrt{\frac{N-1}{N}} \sum_{i=1}^{N} \sqrt{\frac{1-p_i}{N-1}} |i\rangle |1\rangle.$$

6:   Attach $|0\rangle^{\otimes(q+1)}$ qubits on $|\psi^{(0)}\rangle$ and apply $B_f$ on $|0\rangle^{\otimes q}$ to obtain $|\psi^{(1)}\rangle = B_f |\psi^{(0)}\rangle |0\rangle^{\otimes(q+1)}$:

$$|\psi^{(1)}\rangle = B_f |\psi^{(0)}\rangle |0\rangle^{\otimes q+1}$$

$$= \frac{1}{\sqrt{N}} \sum_{i=1}^{N} \sqrt{p_i} |i\rangle |0\rangle |\mathbf{Bi}[f_i]\rangle |0\rangle + \underbrace{\frac{1}{\sqrt{N}} \sum_{i=1}^{N} \sqrt{1-p_i} |i\rangle |1\rangle |\mathbf{Bi}[f_i]\rangle |0\rangle}_{=|\Phi^{(1)}\rangle}.$$

7:   Apply the controlled rotation $R_f$ defined by $R_f : |\mathbf{Bi}[f_i]\rangle |0\rangle \mapsto |\mathbf{Bi}[f_i]\rangle (\sqrt{f_i} |0\rangle + \sqrt{1-f_i} |1\rangle)$ and undo the oracle $B_f$:
$|\psi^{(2)}\rangle = (B_f^\dagger \otimes I)(I \otimes R_f) |\psi^{(1)}\rangle$

$$= \frac{1}{\sqrt{N}} \sum_{i=1}^{N} \sqrt{p_i} |i\rangle |0\rangle |0\rangle \left( \sqrt{f_i} |0\rangle + \sqrt{1-f_i} |1\rangle \right) + \frac{1}{\sqrt{N}} \sum_{i=1}^{N} \sqrt{1-p_i} |i\rangle |1\rangle |0\rangle \left( \sqrt{f_i} |0\rangle + \sqrt{1-f_i} |1\rangle \right)$$

$$= \frac{1}{\sqrt{N}} \sum_{i=1}^{N} \sqrt{p_i f_i} |i\rangle |000\rangle + \underbrace{\frac{1}{\sqrt{N}} \sum_{i=1}^{N} (\sqrt{p_i(1-f_i)} |i\rangle |001\rangle + \sqrt{1-p_i} |i\rangle |10\rangle (\sqrt{f_i} |0\rangle + \sqrt{1-f_i} |1\rangle))}_{=|\Phi^{(2)}\rangle}.$$

8:   Apply $T$ iterations of amplitude estimation by setting $|\psi\rangle = |\psi^{(2)}\rangle, U = 2 |\psi\rangle \langle\psi| - I$ and $P = I \otimes |000\rangle \langle000|$ to obtain $\mu_k$

9: **end for**

10: **Return:** $\hat{\mu} = N \cdot \mathrm{Median}(\{\mu_k\}_{k \in [K]})$

---

with probability at least $1 - \delta$ for any $\delta > 0$. We proceed to focus on the complexity cost of Algorithm 3. Note that $\mu \in [0, 1]$ because $f_i \in [0, 1]$ for all $i = 1, \ldots, N$. Hence, we further have that

$$2\pi \frac{\sqrt{\frac{\mu}{N}\left(1 - \frac{\mu}{N}\right)}}{T} + \frac{\pi^2}{T^2} < \pi^2 \left(\frac{1}{\sqrt{N}T} + \frac{1}{T^2}\right), \tag{49}$$

In order to let $\hat{\mu}/N$ be an $\epsilon/N$ approximation of $\frac{1}{N}\sum_{i=1}^{N} p_i f_i$, it suffices to let

$$\pi^2 \left(\frac{1}{\sqrt{N}T} + \frac{1}{T^2}\right) \leq \frac{\epsilon}{N}, \tag{50}$$

which is equivalent to $\epsilon T^2 - \pi^2 \sqrt{N} T - \pi^2 N \geq 0$. Then it suffices to let $T = O\left(\frac{\sqrt{N}}{\epsilon} + \sqrt{\frac{N}{\epsilon}}\right)$ such that $\left|\frac{\hat{\mu}}{N} - \frac{1}{N}\sum_{i=1}^{N} p_i f_i\right| \leq \epsilon/N$. This implies that $\left|\hat{\mu} - \sum_{i=1}^{N} p_i f_i\right| \leq \epsilon$.

By Theorem A.3, we know that the query complexity of $\hat{U}_p$ in terms of $B_p$ is $O(1)$. Therefore, Algorithm 3 calls $B_p$ and $B_f$ $O\left(\left(\frac{\sqrt{N}}{\epsilon} + \sqrt{\frac{N}{\epsilon}}\right)\log(1/\delta)\right)$ times each. ∎

### A.4.4. PROOF OF LEMMA A.6

**Lemma A.6.** *QVI-2$(\mathcal{M}, \epsilon, \delta)$ holds that $\mathcal{T}_{\hat{\pi}}^h(\hat{V}_{h+1}) - \frac{\epsilon}{H} \leq \hat{V}_h \leq \mathcal{T}_{\hat{\pi}}^h(\hat{V}_{h+1})$ for all $h \in [H]$ with a success probability at least $1 - \delta$.*

**Proof.** The analysis on success probability is the same as Theorem 3.8 and hence we omit it here. For all $s \in \mathcal{S}, a \in \mathcal{A}$ and $h \in [H]$ we have that

$$\left|\frac{z_{h,s}(a)}{H} + \frac{\epsilon}{2H^2} - P_{h|s,a}^{\mathrm{T}}\tilde{V}_{h+1}\right| \leq \frac{\epsilon}{2H^2}. \tag{51}$$

This implies that

$$\left|z_{h,s}(a) + \frac{\epsilon}{2H} - P_{h|s,a}^{\mathrm{T}}\hat{V}_{h+1}\right| \leq \frac{\epsilon}{2H}, \tag{52}$$

and

$$P_{h|s,a}^{\mathrm{T}}\hat{V}_{h+1} - \frac{\epsilon}{H} \leq z_{h,s}(a) \leq P_{h|s,a}^{\mathrm{T}}\hat{V}_{h+1}. \tag{53}$$

Now, for all $s \in \mathcal{S}, a \in \mathcal{A}$ and $h \in [H]$, we let

$$\tilde{Q}_h(s, a) := r_h(s, a) + P_{h|s,a}^{\mathrm{T}}\hat{V}_{h+1}. \tag{54}$$

Note that $\mathcal{T}_{\hat{\pi}}^h(\hat{V}_{h+1}) = \tilde{Q}_h(s, \hat{\pi}(s, h)) = r_h(s, \hat{\pi}(s, h)) + P_{h|s,\hat{\pi}(s,h)}^T \hat{V}_{h+1}$. Therefore, for all $s \in \mathcal{S}, a \in \mathcal{A}$ and $h \in [H]$ we have

$$\hat{Q}_{h,s}(a) - \tilde{Q}_h(s, a) = \max\{r_h(s, a) + z_{h,s}(a), 0\} - \left(r_h(s, a) + P_{h|s,a}^{\mathrm{T}}\hat{V}_{h+1}\right). \tag{55}$$

On one hand, since $z_{h,s}(a) \leq P_{h|s,a}^{\mathrm{T}}\hat{V}_{h+1}$ and $\hat{V}_{h+1} \geq 0$, then we have

$$\begin{aligned}
\hat{Q}_{h,s}(a) - \tilde{Q}_h(s, a) &\leq \max\left\{r_h(s, a) + P_{h|s,a}^{\mathrm{T}}\hat{V}_{h+1}, 0\right\} - \left(r_h(s, a) + P_{h|s,a}^{\mathrm{T}}\hat{V}_{h+1}\right) \\
&= r_h(s, a) + P_{h|s,a}^{\mathrm{T}}\hat{V}_{h+1} - \left(r_h(s, a) + P_{h|s,a}^{\mathrm{T}}\hat{V}_{h+1}\right) \\
&= 0.
\end{aligned} \tag{56}$$

On the other hand, we also have

$$\begin{aligned}
\hat{Q}_{h,s}(a) - \tilde{Q}_h(s, a) &= \max\{r_h(s, a) + z_{h,s}(a), 0\} - \left(r_h(s, a) + P_{h|s,a}^{\mathrm{T}}\hat{V}_{h+1}\right) \\
&\geq r_h(s, a) + z_{h,s}(a) - \left(r_h(s, a) + P_{h|s,a}^{\mathrm{T}}\hat{V}_{h+1}\right) \\
&= z_{h,s}(a) - P_{h|s,a}^{\mathrm{T}}\hat{V}_{h+1} \\
&\geq -\frac{\epsilon}{H}.
\end{aligned} \tag{57}$$

The last line comes from Eq. (53). In summary, for all $s \in \mathcal{S}, a \in \mathcal{A}$ and $h \in [H]$, we have

$$-\frac{\epsilon}{H} \leq \hat{Q}_{h,s}(a) - \tilde{Q}_h(s,a) \leq 0. \tag{58}$$

Hence, by letting $a = \hat{\pi}(s, h)$, we will have

$$-\frac{\epsilon}{H} \leq \hat{V}_h - \mathcal{T}_{\hat{\pi}}^h(\hat{V}_{h+1}) = \hat{Q}_{h,s}(\hat{\pi}(s,h)) - \tilde{Q}_h(s, \hat{\pi}(s,h)) \leq 0, \tag{59}$$

$$\mathcal{T}_{\hat{\pi}}^h(\hat{V}_{h+1}) - \frac{\epsilon}{H} \leq \hat{V}_h \leq \mathcal{T}_{\hat{\pi}}^h(\hat{V}_{h+1}). \tag{60}$$

∎

### A.4.5. MONOTONICITY PROPERTY OF THE VALUE OPERATOR ASSOCIATED WITH A POLICY $\mathcal{T}_{\pi}^h(\cdot)$ IN DEFINITION 2.1

Suppose two vectors $u$ and $v$ satisfy $u \leq v \in \mathbb{R}^{\mathcal{S}}$, then it implies that $u(s) \leq v(s)$ for all $s \in \mathcal{S}$. Consequently, we must have, for any fixed policy $\pi$ and for all $s \in \mathcal{S}$ and $h \in [H]$,

$$\sum_{s' \in \mathcal{S}} P_{h|s,\pi(s,h)}(s')u(s') \leq \sum_{s' \in \mathcal{S}} P_{h|s,\pi(s,h)}(s')v(s'). \tag{61}$$

Further, we can know that

$$r(s, \pi(s,h)) + \sum_{s' \in \mathcal{S}} P_{h|s,\pi(s,h)}(s')u(s') \leq r(s, \pi(s,h)) + \sum_{s' \in \mathcal{S}} P_{h|s,\pi(s,h)}(s')v(s'). \tag{62}$$

By the definitions of $\mathcal{T}_{\pi}^h(u)$ and $\mathcal{T}_{\pi}^h(v)$, this implies that $[\mathcal{T}_{\pi}^h(u)]_s \leq [\mathcal{T}_{\pi}^h(v)]_s$ for all $s \in \mathcal{S}$ and $h \in [H]$. In other words, $\mathcal{T}_{\pi}^h(u) \leq \mathcal{T}_{\pi}^h(v)$. This implies that the operator $\mathcal{T}_{\pi}^h$ is monotonically increasing for any $\pi$ and $h \in [H]$ in the coordinate-wise order.

### A.4.6. CORRECTNESS OF **QVI-2** (PROOF OF THEOREM 3.8)

**Proof.** We start by examining the failure probability. The approach is similar to the analysis in Theorem 3.5, except that we must now account for quantum oracles that can fail. To address this, we use fundamental properties of unitary matrices, particularly a quantum analog of the union bound, which states that the failure probabilities of quantum operators (unitary matrices) combine linearly.

In line 5, since $B_{z_{h,s}}$ is constructed using **QMEBO** with a failure probability $\zeta$, it is within $2A\zeta$ of its "ideal version." Specifically, this means that there exists an ideal quantum oracle $B_{z_{h,s}}^{\text{ideal}}$ encoding $HP_{h|s,a}^{\mathrm{T}}\widetilde{\tilde{V}_{h+1}} - \epsilon/2H$, where $P_{h|s,a}^{\mathrm{T}}\widetilde{\tilde{V}_{h+1}}$ satisfies $\left\| P_{h|s,a}^{\mathrm{T}}\widetilde{\tilde{V}_{h+1}} - P_{h|s,a}^{\mathrm{T}}\tilde{V}_{h+1} \right\|_{\infty} \leq \epsilon/(2H^2)$, such that $\left\| B_{z_{h,s}}^{\text{ideal}} - B_{z_{h,s}} \right\|_{\text{op}} \leq 2A\zeta$. Since $B_{\hat{Q}_{h,s}}$ is formed using one call each to $B_{z_{h,s}}$ and $B_{z_{h,s}}^{\dagger}$, it is within $4A\zeta$ of its ideal counterpart $B_{\hat{Q}_{h,s}}^{\text{ideal}}$. By applying the quantum union bound and substituting the definition of $\zeta$, this shows that the quantum operation executed by **QMS** is $(\tilde{c}\sqrt{A}\log(1/\delta) \cdot 4A\zeta = \delta/SH)$-close to its ideal version. Consequently, the output of **QMS** is incorrect with a probability of at most $\delta/SH$. Given that **QMS** is invoked a total of $SH$ times, the overall failure probability is bounded by $\delta$, as ensured by the standard union bound.

In line 5, we apply **QMEBO** to obtain an approximate value $z_{h,s}(a)/H$ of the inner product $P_{h|s,a}^{\mathrm{T}}\tilde{V}_{h+1}$. Theorem 3.7 guarantees the output in the line 5 satisfying $|z_{h,s}(a)/H + \epsilon/2H^2 - P_{h|s,a}^{\mathrm{T}}\tilde{V}_{h+1}| \leq \epsilon/2H^2$ for all $s \in \mathcal{S}, a \in \mathcal{A}$ and $h \in [H]$. This implies that $|z_{h,s}(a) - P_{h|s,a}^{\mathrm{T}}\hat{V}_{h+1}| \leq \epsilon/H$. Hence, it holds for all $s \in \mathcal{S}$ and $a \in \mathcal{A}$ in every loop $h \in [H]$ that

$$\begin{aligned}
|\hat{Q}_{h,s}(a) - Q_h^*(s,a)| &= \left| r_h(s,a) + z_{h,s}(a) - (r_h(s,a) + P_{h|s,a}^{\mathrm{T}}V_{h+1}^*) \right| \\
&\leq \left| z_{h,s}(a) - P_{h|s,a}^{\mathrm{T}}(V_{h+1}^* - \hat{V}_{h+1}) - P_{h|s,a}^{\mathrm{T}}\hat{V}_{h+1} \right| \\
&\leq \left| z_{h,s}(a) - P_{h|s,a}^{\mathrm{T}}\hat{V}_{h+1} \right| + \left| P_{h|s,a}^{\mathrm{T}}(V_{h+1}^* - \hat{V}_{h+1}) \right| \\
&\leq \frac{\epsilon}{H} + \max_{s \in \mathcal{S}} \left| \hat{V}_{h+1}(s) - V_{h+1}^*(s) \right| \\
&= \frac{\epsilon}{H} + \left\| \hat{V}_{h+1} - V_{h+1}^* \right\|_{\infty}.
\end{aligned} \tag{63}$$

Furthermore, we have

$$\begin{aligned}
\left\|\hat{V}_h - V_h^*\right\|_\infty &= \left\|\hat{Q}_{h,s}\big(\hat{\pi}(s,h)\big) - \max_{a \in \mathcal{A}} Q_h^*(s,a)\right\|_\infty \\
&= \left\|\max_{a \in \mathcal{A}} \hat{Q}_{h,s}(a) - \max_{a \in \mathcal{A}} Q_h^*(s,a)\right\|_\infty \\
&\le \left\|\max_{a \in \mathcal{A}} |\hat{Q}_{h,s}(a) - Q_h^*(s,a)|\right\|_\infty \\
&\le \max_{s \in \mathcal{S}} \max_{a \in \mathcal{A}} |\hat{Q}_{h,s}(a) - Q_h^*(s,a)| \\
&\le \frac{\epsilon}{H} + \left\|\hat{V}_{h+1} - V_{h+1}^*\right\|_\infty .
\end{aligned} \tag{64}$$

Since it holds that $\hat{V}_H(s) = V_H^*(s) = 0$ for all $s \in \mathcal{S}$, we can induce that

$$\left\|\hat{V}_h - V_h^*\right\|_\infty \le \frac{(H-h)\epsilon}{H} + \left\|\hat{V}_H - V_H^*\right\|_\infty = \frac{(H-h)\epsilon}{H}. \tag{65}$$

Then, we know that $\left\|\hat{V}_h - V_h^*\right\|_\infty \le \epsilon$ for all $h \in [H]$. In particular, it implies that $V_h^*(s) - \epsilon \le \hat{V}_h(s)$ for all $s \in \mathcal{S}$ and $h \in [H]$.

Now, we proceed to prove the $\hat{V}_h(s) \le V_h^{\hat{\pi}}(s)$ for all $s \in \mathcal{S}$ and $h \in [H]$. By Lemma A.6, we know that $\hat{V}_h \le \mathcal{T}_{\hat{\pi}}^h(\hat{V}_{h+1})$ for all $h \in [H]$. Therefore, $\hat{V}_{H-1}(s) \le [\mathcal{T}_{\hat{\pi}}^{H-1}(\hat{V}_H)]_s = [\mathcal{T}_{\hat{\pi}}^{H-1}(0)]_s = r_{H-1}(s, \hat{\pi}(s, H-1)) = V_{H-1}^{\hat{\pi}}(s)$. By the monotonicity of the operators $\mathcal{T}_{\hat{\pi}}^h$, where $h \in [H]$, we have $\hat{V}_h \le \mathcal{T}_{\hat{\pi}}^h(\hat{V}_{h+1}) \le \mathcal{T}_{\hat{\pi}}^h(\mathcal{T}_{\hat{\pi}}^{h+1}(\hat{V}_{h+2})) \le \cdots \le V_h^{\hat{\pi}}$ for all $h \in [H]$. Due to the definition of $V_h^*(s)$, we must have $V_h^*(s) = \max_{\pi \in \Pi} V_h^\pi(s) \ge V_h^{\hat{\pi}}(s)$ for all $s \in \mathcal{S}$ and $h \in [H]$. ∎

### A.4.7. COMPLEXITY OF **QVI-2** (PROOF OF THEOREM 3.9)

**Proof.** We first assume that all **QMS** and **QMEBO** are correct, up to the specified error, because the probability that this does not hold is at most $\delta$. Let $C$ be the complexity of **QVI-2**$(\mathcal{M}, \epsilon, \delta)$ as if all **QMS** and **QMEBO** are carried out with maximum failure probabilities set to constant. Then, since the failure probabilities are set to $\zeta = \delta/(4cHSA^{1.5}\log(1/\delta))$, the actual complexity of **QVI-2**$(\mathcal{M}, \epsilon, \delta)$ is

$$O\Big(C \log\big(SA^{1.5}H\log(1/\delta)/\delta\big)\Big) = O\big(C \log(SA^{1.5}H/\delta)\big). \tag{66}$$

Now, we check each line of **QVI-2**$(\mathcal{M}, \epsilon, \delta)$ to bound $C$.

In line 4, we encode the vector $\tilde{V}_{h+1} = \hat{V}_{h+1}/H$ to an oracle $B_{\tilde{V}_{h+1}}$. This process does not need to query $O_{Q\mathcal{M}}$ and only needs to access the classical vector $\hat{V}_{h+1}$.

In line 5, we implement **QMEBO** to compute an estimate of $P_{h|s,a}^{\mathrm{T}} \tilde{V}_{h+1}$ with error $\epsilon/2H^2$. Besides, the correctness analysis shows that $\hat{V}_{h+1}(s) \le V_{h+1}^*(s) \le H$ for all $s \in \mathcal{S}$ and the definition of $\hat{Q}_{h,s}(a)$ of **QVI2** implies that $0 \le \hat{V}_h(s)$ for all $s \in \mathcal{S}$, so it also holds that $0 \le \tilde{V}_{h+1}(s) = \hat{V}_{h+1}(s)/H \le 1$ for all $s \in \mathcal{S}$ and $h \in [H]$. By Theorem 3.7, we know that **QMEBO** needs $O(\sqrt{S}(H^2/\epsilon + \sqrt{H^2/\epsilon})) = O(\sqrt{S}H^2/\epsilon)$ queries to $O_{Q\mathcal{M}}$ for each $s \in \mathcal{S}$ at each time step $h \in [H]$, provided $0 < \epsilon \le H^2$. Since we have assumed that $\epsilon \le H$ on the input $\epsilon$, so this holds.

In line 6, $B_{\hat{Q}_{h,s}}$ needs to call $B_{z_{h,s}}$ and $B_{z_{h,s}}^\dagger$ once. Then the query complexity of $B_{\hat{Q}_{h,s}}$ in terms of $B_{z_{h,s}}$ is $O(1)$.

In line 7, we use **QMS** in accelerating the searching for the optimal action $\hat{\pi}(s,h)$ for all $s \in \mathcal{S}$. By Theorem 3.4, **QMS** requires $O(\sqrt{A})$ queries to the oracle $B_{\hat{Q}_{h,s}}$ for all $s \in \mathcal{S}$ and $h \in [H]$. Therefore, after summing up $H$ iterations, it induces an overall query cost of

$$C = O\!\left(S \cdot \sqrt{A} \cdot H \cdot \frac{\sqrt{S}H^2}{\epsilon}\right) = O\!\left(\frac{S^{\frac{3}{2}}\sqrt{A}H^3}{\epsilon}\right). \tag{67}$$

Combining the above equation with Eq. (66), the overall quantum query complexity of **QVI-2**$(\mathcal{M}, \epsilon, \delta)$ is

$$O\left(\frac{S^{\frac{3}{2}}\sqrt{A}H^3 \log(SA^{1.5}H/\delta)}{\epsilon}\right). \tag{68}$$

■

## B. Generative Model Setting

### B.1. Correctness and Complexity of QVI-3 (Algorithm 4)

#### B.1.1. PROOF OF LEMMA B.1

**Lemma B.1.** *QVI-3*$(\mathcal{M}, \epsilon, \delta)$ *holds that* $\mathcal{T}_{\hat{\pi}}^h(\hat{V}_{h+1}) - \frac{\epsilon}{H} \leq \hat{V}_h \leq \mathcal{T}_{\hat{\pi}}^h(\hat{V}_{h+1})$ *for all* $h \in [H]$ *with a success probability at least* $1 - \delta$.

**Proof.** The analysis of success probability is the same as Theorem 4.3 and hence is omitted here. We proceed to show the correctness of the claim. For all $s \in \mathcal{S}, a \in \mathcal{A}$ and $h \in [H]$ we have that

$$\left|z_{h,s}(a) + \frac{\epsilon}{2H} - P_{h|s,a}^T \hat{V}_{h+1}\right| \leq \frac{\epsilon}{2H}. \tag{69}$$

This implies, for all $s \in \mathcal{S}, a \in \mathcal{A}$ and $h \in [H]$,

$$P_{h|s,a}^T \hat{V}_{h+1} - \frac{\epsilon}{H} \leq z_{h,s}(a) \leq P_{h|s,a}^T \hat{V}_{h+1}. \tag{70}$$

Now, for all $s \in \mathcal{S}, a \in \mathcal{A}$ and $h \in [H]$, we let

$$\tilde{Q}_h(s,a) := r_h(s,a) + P_{h|s,a}^T \hat{V}_{h+1} \text{ and } \tilde{V}_h(s) := \max_{a \in \mathcal{A}} \tilde{Q}_h(s,a). \tag{71}$$

Note that $\mathcal{T}_{\hat{\pi}}^h(\hat{V}_{h+1}) = \tilde{Q}_h(s, \hat{\pi}(s,h)) = r_h(s, \hat{\pi}(s,h)) + P_{h|s,\hat{\pi}(s,h)}^T \hat{V}_{h+1}$. Therefore, for all $s \in \mathcal{S}, a \in \mathcal{A}$ and $h \in [H]$, we have

$$\hat{Q}_{h,s}(a) - \tilde{Q}_h(s,a) = \max\{r_h(s,a) + z_{h,s}(a), 0\} - (r_h(s,a) + P_{h|s,a}^T \hat{V}_{h+1}). \tag{72}$$

On one hand, since $z_{h,s}(a) \leq P_{h|s,a}^T \hat{V}_{h+1}$ and $\hat{V}_{h+1} \geq 0$, then we have

$$\begin{aligned}
\hat{Q}_{h,s}(a) - \tilde{Q}_h(s,a) &\leq \max\{r_h(s,a) + P_{h|s,a}^T \hat{V}_{h+1}, 0\} - (r_h(s,a) + P_{h|s,a}^T \hat{V}_{h+1}) \\
&= r_h(s,a) + P_{h|s,a}^T \hat{V}_{h+1} - (r_h(s,a) + P_{h|s,a}^T \hat{V}_{h+1}) \\
&= 0.
\end{aligned} \tag{73}$$

On the other hand, we also have

$$\begin{aligned}
\hat{Q}_{h,s}(a) - \tilde{Q}_h(s,a) &= \max\{r_h(s,a) + z_{h,s}(a), 0\} - (r_h(s,a) + P_{h|s,a}^T \hat{V}_{h+1}), \\
&\geq r_h(s,a) + z_{h,s}(a) - (r_h(s,a) + P_{h|s,a}^T \hat{V}_{h+1}), \\
&= z_{h,s}(a) - P_{h|s,a}^T \hat{V}_{h+1}, \\
&\geq -\frac{\epsilon}{H}.
\end{aligned} \tag{74}$$

The last line comes from Eq. (70). In summary, for all $s \in \mathcal{S}, a \in \mathcal{A}$ and $h \in [H]$, we have

$$-\frac{\epsilon}{H} \leq \hat{Q}_{h,s}(a) - \tilde{Q}_h(s,a) \leq 0, \tag{75}$$

Hence, by letting $a = \hat{\pi}(s,h)$, we will have

$$-\frac{\epsilon}{H} \leq \hat{V}_h - \mathcal{T}_{\hat{\pi}}^h(\hat{V}_{h+1}) = \hat{Q}_{h,s}(\hat{\pi}(s,h)) - \tilde{Q}_h(s, \hat{\pi}(s,h)) \leq 0, \tag{76}$$

$$\mathcal{T}_{\hat{\pi}}^h(\hat{V}_{h+1}) - \frac{\epsilon}{H} \le \hat{V}_h \le \mathcal{T}_{\hat{\pi}}^h(\hat{V}_{h+1}). \tag{77}$$

∎

### B.1.2. CORRECTNESS OF **QVI-3** (PROOF OF THEOREM 4.3)

**Proof.** We start by examining the failure probability. The analysis is similar to Theorem 3.8 where we need to consider quantum oracles that can fail. Again, we use the quantum union bound for quantum operators here.

In line 5, since $B_{z_{h,s}}$ is constructed using **QME1** with a failure probability $\zeta$, it is within $2A\zeta$ of its "ideal version." Specifically, this means that there exists an ideal quantum oracle $B_{z_{h,s}}^{\text{ideal}}$ encoding $\widetilde{P_{h|s,a}^{\mathrm{T}} \hat{V}_{h+1}} - \epsilon/2H$, where $\widetilde{P_{h|s,a}^{\mathrm{T}} \hat{V}_{h+1}}$ satisfies $\left\| \widetilde{P_{h|s,a}^{\mathrm{T}} \hat{V}_{h+1}} - P_{h|s,a}^{\mathrm{T}} \hat{V}_{h+1} \right\|_{\infty} \le \epsilon/(2H)$, such that $\left\| B_{z_{h,s}}^{\text{ideal}} - B_{z_{h,s}} \right\|_{\text{op}} \le 2A\zeta$. Since $B_{\hat{Q}_{h,s}}$ is formed using one call each to $B_{z_{h,s}}$ and $B_{z_{h,s}}^{\dagger}$, it is within $4A\zeta$ of its ideal counterpart $B_{\hat{Q}_{h,s}}^{\text{ideal}}$. By applying the quantum union bound and substituting the definition of $\zeta$, this shows that the quantum operation executed by **QMS** is $(\tilde{c}\sqrt{A} \log(1/\delta) \cdot 4A\zeta = \delta/SH)$-close to its ideal version. Consequently, the output of **QMS** is incorrect with a probability of at most $\delta/SH$. Given that **QMS** is invoked a total of $SH$ times, the overall failure probability is bounded by $\delta$, as ensured by the standard union bound.

In line 5, we apply **QME1** to obtain an approximate value $z_{h,s}(a)$ of the inner product $P_{h|s,a}^{\mathrm{T}} \hat{V}_{h+1}$. Theorem 4.2 guarantees the output in the line 5 satisfying $|z_{h,s}(a) - P_{h|s,a}^{\mathrm{T}} \hat{V}_{h+1}| \le \epsilon/H$ for all $s \in \mathcal{S}, a \in \mathcal{A}$ and $h \in [H]$. Hence, it holds for all $s \in \mathcal{S}$ and $a \in \mathcal{A}$ in every loop $h \in [H]$ that

$$
\begin{aligned}
\left| \hat{Q}_{h,s}(a) - Q_h^*(s,a) \right| &= \left| r_h(s,a) + z_{h,s}(a) - \left( r_h(s,a) + P_{h|s,a}^{\mathrm{T}} V_{h+1}^* \right) \right| \\
&\le \left| z_{h,s}(a) - P_{h|s,a}^{\mathrm{T}}(V_{h+1}^* - \hat{V}_{h+1}) - P_{h|s,a}^{\mathrm{T}} \hat{V}_{h+1} \right| \\
&\le \left| z_{h,s}(a) - P_{h|s,a}^{\mathrm{T}} \hat{V}_{h+1} \right| + \left| P_{h|s,a}^{\mathrm{T}}(V_{h+1}^* - \hat{V}_{h+1}) \right| \\
&\le \frac{\epsilon}{H} + \max_{s \in \mathcal{S}} \left| \hat{V}_{h+1}(s) - V_{h+1}^*(s) \right| \\
&= \frac{\epsilon}{H} + \left\| \hat{V}_{h+1} - V_{h+1}^* \right\|_{\infty}.
\end{aligned}
\tag{78}
$$

Further, we have

$$
\begin{aligned}
\left\| \hat{V}_h - V_h^* \right\|_{\infty} &= \left\| \hat{Q}_{h,s}\left( \hat{\pi}(s,h) \right) - \max_{a \in \mathcal{A}} Q_h^*(s,a) \right\|_{\infty} \\
&= \left\| \max_{a \in \mathcal{A}} \hat{Q}_{h,s}(a) - \max_{a \in \mathcal{A}} Q_h^*(s,a) \right\|_{\infty} \\
&\le \left\| \max_{a \in \mathcal{A}} \left| \hat{Q}_{h,s}(a) - Q_h^*(s,a) \right| \right\|_{\infty} \\
&\le \max_{s \in \mathcal{S}} \max_{a \in \mathcal{A}} \left| \hat{Q}_{h,s}(a) - Q_h^*(s,a) \right| \\
&\le \frac{\epsilon}{H} + \left\| \hat{V}_{h+1} - V_{h+1}^* \right\|_{\infty}.
\end{aligned}
\tag{79}
$$

Since it holds that $\hat{V}_h(s) = V_h^*(s) = 0$ for all $s \in \mathcal{S}$, we can induce that

$$\left\| \hat{V}_h - V_h^* \right\|_{\infty} \le \frac{(H-h)\epsilon}{H} + \left\| \hat{V}_H - V_H^* \right\|_{\infty} = \frac{(H-h)\epsilon}{H}. \tag{80}$$

Then, we know that $\left\| \hat{V}_h - V_h^* \right\|_{\infty} \le \epsilon$ for all $h \in [H]$. In particular, it implies that $V_h^*(s) - \epsilon \le \hat{V}_h(s)$ for all $s \in \mathcal{S}$ and $h \in [H]$. Now, we proceed to prove the $\hat{V}_h(s) \le V_h^{\hat{\pi}}(s)$ for all $s \in \mathcal{S}$ and $h \in [H]$. By Lemma B.1, we know that $\hat{V}_h \le \mathcal{T}_{\hat{\pi}}^h(\hat{V}_{h+1})$ for all $h \in [H]$. Therefore, $\hat{V}_{H-1}(s) \le [\mathcal{T}_{\hat{\pi}}^{H-1}(\hat{V}_H)]_s = [\mathcal{T}_{\hat{\pi}}^{H-1}(0)]_s = r_{H-1}\left(s, \hat{\pi}(s, H-1)\right) = V_{H-1}^{\hat{\pi}}(s)$. By the monotonicity of the operators $\mathcal{T}_{\hat{\pi}}^h$, where $h \in [H]$, we have $\hat{V}_h \le \mathcal{T}_{\hat{\pi}}^h(\hat{V}_{h+1}) \le \mathcal{T}_{\hat{\pi}}^h(\mathcal{T}_{\hat{\pi}}^{h+1}(\hat{V}_{h+2})) \le \cdots \le V_h^{\hat{\pi}}$ for

all $h \in [H]$. Due to the definition of $V_h^*(s)$, we must have $V_h^*(s) = \max_{\pi \in \Pi} V_h^\pi(s) \geq V_h^{\hat{\pi}}(s)$ for all $s \in \mathcal{S}$ and $h \in [H]$. ∎

### B.1.3. COMPLEXITY OF **QVI-3** (PROOF OF THEOREM 4.4)

**Proof.** We first assume that all **QMS** and **QME1** are correct, up to the specified error, because the probability that this does not hold is at most $\delta$. Let $C$ be the complexity of **QVI-3**$(\mathcal{M}, \epsilon, \delta)$ as if all **QMS** and **QME1** are carried out with maximum failure probabilities set to constant. Then, since the actual failure probabilities are set to $\zeta = \delta/(4cSA^{1.5}H \log(1/\delta))$, the actual complexity of **QVI-3**$(\mathcal{M}, \epsilon, \delta)$ is

$$O\Big(C \log\big(SA^{1.5}H \log(1/\delta)/\delta\big)\Big) = O\big(C \log(SA^{1.5}H/\delta)\big). \tag{81}$$

Now, we check each line of **QVI-3**$(\mathcal{M}, \epsilon, \delta)$ to bound $C$.

In line 4, we encode the vector $\hat{V}_{h+1}$ to an oracle $B_{\hat{V}_{h+1}}$. This process does not need to query $\mathcal{G}$ and only needs to access the classical vector $\hat{V}_{h+1}$.

In line 5, we implement **QME1** to compute the approximate inner product of $P_{h|s,a}^{\mathrm{T}} \hat{V}_{h+1}$ with error $\epsilon/H$. Besides, the correctness analysis shows that it holds that $\hat{V}_{h+1}(s) \leq V_{h+1}^*(s) \leq H$ for all $s \in \mathcal{S}$ and $h \in [H]$ by Theorem 4.3 and $\hat{V}_{h+1}(s) \geq 0$ for all $s \in \mathcal{S}$ by the definition of itself, $\hat{Q}_{h+1,s}(a)$ and $z_{h+1,s}(a)$ in **QVI-3**. By Theorem 4.2, we know that **QME1** needs $O\big(H^2/\epsilon + \sqrt{H^2/\epsilon}\big) = O(H^2/\epsilon)$ queries to $\mathcal{G}$ for each $s \in \mathcal{S}$, provided $0 < \epsilon \leq H^2$. Since we have assumed that $\epsilon \leq H$ on the input $\epsilon$, so this holds.

In line 7, $B_{\hat{Q}_{h,s}}$ needs to call $B_{z_{h,s}}$ and $B_{z_{h,s}}^\dagger$ once. Then the query complexity of $B_{\hat{Q}_{h,s}}$ in terms of $B_{z_{h,s}}$ is $O(1)$.

In line 8, we use **QMS** in accelerating the searching for the optimal action $\hat{\pi}(s,h)$ for all $s \in \mathcal{S}$. By Theorem 3.4, **QMS** requires $O(\sqrt{A})$ queries to the oracle $B_{\hat{Q}_{h,s}}$ for all $s \in \mathcal{S}$ and $h \in [H]$. Therefore, it induces an overall query cost of

$$C = O\left(S \cdot \sqrt{A} \cdot H \cdot \frac{H^2}{\epsilon}\right) = O\left(\frac{S\sqrt{A}H^3}{\epsilon}\right), \tag{82}$$

in $H$ iterations. Combining the above equation with Equation (81), the overall quantum query complexity of **QVI-3**$(\mathcal{M}, \epsilon, \delta)$ is

$$O\left(\frac{S\sqrt{A}H^3 \log(SA^{1.5}H/\delta)}{\epsilon}\right). \tag{83}$$

∎

## B.2. Correctness and Complexity of QVI-4

**Lemma B.2** (Upper Bound on Variance (Sidford et al., 2018)). *For any policy $\pi : \mathcal{S} \times [H] \to \mathcal{A}$, it must hold that*

$$\left\|\sum_{h'=h}^{H-1}\left(\prod_{i=h+1}^{h'} P_i^\pi\right)\sigma_{h'}(V_{h'+1}^\pi)\right\|_\infty \leq H^{3/2}, \tag{84}$$

*where $\sigma_{h'}(V_{h'+1}^\pi) = \sqrt{P_{h'}(V_{h'+1}^\pi)^2 - (P_{h'}V_{h'+1}^\pi)^2}$.*

### B.2.1. PROOF OF LEMMA B.3

**Lemma B.3.** *For all $k \in [K]$ and $h \in [H]$, Algorithm 5 holds that*

$$V_{k,h} \leq V_h^{\pi_k} \leq V_h^*, \tag{85}$$
$$Q_{k,h} \leq Q_h^{\pi_k} \leq Q_h^*, \tag{86}$$

*with probability at least $1 - \delta$.*

**Proof.** We first consider the success probability. Note that all the quantum subroutines **QME1** and **QME2** are implemented with maximum failure probability $\zeta = \delta/4KHSA$. In total, **QME1** and **QME2** are implemented $4KHSA$ times in line 6, 7 and 9. By the union bound, the probability that there exists an incorrect estimate is at most $\delta$.

Now, we proceed to prove the inequalities

$$V_{k,h} \leq V_h^{\pi_k} \leq V_h^*. \tag{87}$$

Note that the second inequality is trivial due to the definition of $V_h^* = \max_{\pi \in \Pi} V_h^\pi$. Therefore, we only need to prove the inequality $V_{k,h} \leq V_h^{\pi_k}$ for all $h \in [H]$ and $k \in [K]$. In fact, it suffices to show that for all $k \in [K]$, we have

$$V_{k,h} \leq \mathcal{T}_{\pi_k}^h(V_{k,h+1}). \tag{88}$$

First, by the definition of $x_{k,h}$ and $g_{k,h}$ in line 7 and line 9 respectively, we have, for all $(s,a) \in \mathcal{S} \times \mathcal{A}$,

$$x_{k,h}(s,a) \leq P_{h|s,a}^{\mathrm{T}} V_{k,h+1}^{(0)}, \tag{89}$$

$$g_{k,h}(s,a) \leq P_{h|s,a}^{\mathrm{T}}(V_{k,h+1} - V_{k,h+1}^{(0)}). \tag{90}$$

We continue to prove Eq. (88) by induction on $k$. We first consider the base case where $k = 0$. For any $h \in [H]$, if there exists some $s \in \mathcal{S}$ such that $\pi_k(s,h) \neq \pi_k^{(0)}(s,h)$, then we have

$$
\begin{aligned}
V_{k,h}(s) &= \tilde{V}_{k,h}(s) \\
&= Q_{k,h}\big(s, \pi_k(s,h)\big) \\
&= \max\Big\{ r_h\big(s, \pi_k(s,h)\big) + x_{k,h}\big(s, \pi_k(s,h)\big) + g_{k,h}\big(s, \pi_k(s,h)\big), 0 \Big\} \\
&\leq \max\Big\{ r_h\big(s, \pi_k(s,h)\big) + P_{h|s,\pi_k(s,h)}^{\mathrm{T}} V_{k,h+1}^{(0)} + P_{h|s,\pi_k(s,h)}^{\mathrm{T}}(V_{k,h+1} - V_{k,h+1}^{(0)}), 0 \Big\} \\
&= \max\Big\{ r_h\big(s, \pi_k(s,h)\big) + P_{h|s,\pi_k(s,h)}^{\mathrm{T}} V_{k,h+1}, 0 \Big\} \\
&= r_h\big(s, \pi_k(s,h)\big) + P_{h|s,\pi_k(s,h)}^{\mathrm{T}} V_{k,h+1} \\
&= \big[\mathcal{T}_{\pi_k}^h(V_{k,h+1})\big]_s.
\end{aligned} \tag{91}
$$

If there exists some $s \in \mathcal{S}$ such that $\pi_k(s,h) = \pi_k^{(0)}(s,h)$, then we have $V_{k,h}(s) = V_{k,h}^{(0)}(s) = V_{0,h}^{(0)}(s) = 0$. Since $V_{k,h+1}(s) \geq V_{k,h+1}^{(0)}(s) = V_{0,h+1}^{(0)}(s) = 0$ for all $s \in \mathcal{S}$, then we must have $V_{k,h}(s) = 0 \leq [\mathcal{T}_{\pi_k}^h(V_{k,h+1})]_s$. Therefore, when $k = 0$, it holds that $V_{k,h} \leq \mathcal{T}_{\pi_k}^h(V_{k,h+1})$ for all $h \in [H]$. We assume that for any $k' = 0, 1, \ldots, k-1$, it also holds that $V_{k',h} \leq \mathcal{T}_{\pi_k}^h(V_{k',h+1})$ for all $h \in [H]$. Next, we show the above statement holds for $k' = k$. In fact, if there exists some $s \in \mathcal{S}$ such that $\pi_k(s,h) \neq \pi_k^{(0)}(s,h)$, then we also have $V_{k,h}(s) \leq [\mathcal{T}_\pi^h(V_{k,h+1})]_s$ by following the same analysis in the case of $k = 0$. If there exists some $s \in \mathcal{S}$ such that $\pi_k(s,h) = \pi_k^{(0)}(s,h)$, then we have

$$V_{k,h}(s) = V_{k,h}^{(0)}(s) = V_{k-1,h}(s) \leq \big[\mathcal{T}_{\pi_{k-1}}^h(V_{k-1,h+1})\big]_s = \big[\mathcal{T}_{\pi_{k-1}}^h(V_{k,h+1}^{(0)})\big]_s \leq \big[\mathcal{T}_{\pi_{k-1}}^h(V_{k,h+1})\big]_s = \big[\mathcal{T}_{\pi_k}^h(V_{k,h+1})\big]_s. \tag{92}$$

The first inequality comes from the induction hypothesis. The second inequality comes from the fact that $V_{k,h+1}^{(0)} \leq V_{k,h+1}$. The last equation comes from the fact that $\pi_k(s,h) = \pi_k^{(0)}(s,h) = \pi_{k-1}(s,h)$. Therefore, we already showed that $V_{k,h} \leq \mathcal{T}_{\pi_k}^h(V_{k,h+1})$ for the case $k' = k$ and finish the induction. Since we have $V_{k,h} \leq \mathcal{T}_{\pi_k}^h(V_{k,h+1})$ for all $k \in [K]$ and $h \in [H]$ and $V_{k,H}(s) = 0, \forall s \in \mathcal{S}$, then for any fixed $k \in [K]$, $V_{k,h} \leq \mathcal{T}_{\pi_k}^h(\cdots \mathcal{T}_{\pi_k}^{H-1}(V_{k,H})) = V_h^{\pi_k}$ for all $h \in [H]$.

Furthermore, since we already proved $V_{k,h} \leq V_h^{\pi_k}$ for all $h \in [H]$ and $k \in [K]$, we also have, for all $(s,a) \in \mathcal{S} \times \mathcal{A}$,

$$Q_{k,h}(s,a) \leq r_h(s,a) + P_{h|s,a}^{\mathrm{T}} V_{k,h+1} \leq r_h(s,a) + P_{h|s,a}^{\mathrm{T}} V_{h+1}^{\pi_k} = Q_h^{\pi_k}(s,a) \leq Q_h^*(s,a). \tag{93}$$

The first inequality follows from Eq. (89) and (90). ∎

B.2.2. PROOF OF LEMMA B.4

**Lemma B.4.** *For all $k \in [K]$ and $h \in [H]$, Algorithm 5 holds that*

$$V_h^* - \epsilon_k \leq V_{k,h}, \tag{94}$$

$$Q_h^* - \epsilon_k \leq Q_{k,h}, \tag{95}$$

*with the probability at least $1 - \delta$.*

**Proof.** The success probability analysis is the same as Lemma B.3, so we omit it here. We continue to use induction on $k$ to prove Eq. (94). First, we consider the base case where $k = 0$ and show (94) holds for all $h \in [H]$. By the definition of $x_{k,h}$ and $g_{k,h}$ in line 7 and line 9 of **QVI-4**, we know that, for all $(s, a) \in \mathcal{S} \times \mathcal{A}$,

$$x_{k,h}(s,a) \geq P_{h|s,a}^{\mathrm{T}} V_{k,h+1}^{(0)} - 2cH^{-1.5}\epsilon\sqrt{y_{k,h}(s,a) + 4b}, \tag{96}$$

$$g_{k,h}(s,a) \geq P_{h|s,a}^{\mathrm{T}}(V_{k,h+1} - V_{k,h+1}^{(0)}) - 2cH^{-1}\epsilon_k. \tag{97}$$

We define $\xi_{k,h}(s,a) := 2cH^{-1}\epsilon_k + 2cH^{-1.5}\epsilon\sqrt{y_{k,h}(s,a) + 4b}$. Then, we can show that

$$
\begin{aligned}
Q_h^* - Q_{k,h} &= r_h + P_h V_{h+1}^* - \max\{r_h + x_{k,h} + g_{k,h}, 0\} \\
&\leq P_h V_{h+1}^* - x_{k,h} - g_{k,h} \\
&\leq P_h V_{h+1}^* - P_h V_{k,h+1} + 2cH^{-1}\epsilon_k + 2cH^{-1.5}\epsilon\sqrt{y_{k,h} + 4b} \\
&= P_h(V_{h+1}^* - V_{k,h+1}) + \xi_{k,h} \\
&= P_h V(Q_{h+1}^*) - P_h V_{k,h+1} + \xi_{k,h}.
\end{aligned} \tag{98}
$$

Since we have $V(Q_{k,h+1}) \leq V_{k,h+1}$, then it holds that

$$
\begin{aligned}
Q_h^* - Q_{k,h} &\leq P_h V(Q_{h+1}^*) - P_h V(Q_{k,h+1}) + \xi_{k,h} \\
&= P_h^{\pi^*} Q_{h+1}^* - P_h V(Q_{k,h+1}) + \xi_{k,h} \\
&\leq P_h^{\pi^*} Q_{h+1}^* - P_h^{\pi^*} Q_{k,h+1} + \xi_{k,h}.
\end{aligned} \tag{99}
$$

The second line comes from the fact that $V_h^*(s) = Q_{h+1}^*(s, \pi^*(s,h))$ for all $s \in \mathcal{S}$ and $h \in [H]$. The last line comes from the fact that $\pi^*(s,h)$ may not be the same as $\arg\max_{a \in \mathcal{A}} Q_{k,h+1}(s,a)$ for some $s \in \mathcal{S}$. Since it must hold that $V_H^*(s) = 0, \forall s \in \mathcal{S}$ and we require that $V_{k,H}(s) = 0, \forall s \in \mathcal{S}$, then we have $V_H^*(s) - V_{k,H}(s) = 0, \forall s \in \mathcal{S}$. By solving the recursion on $Q_h^* - Q_{k,h}$, we can obtain

$$Q_h^* - Q_{k,h} \leq \sum_{h'=h}^{H-1} \left( \prod_{i=h+1}^{h'} P_i^{\pi^*} \right) \xi_{k,h'}, \tag{100}$$

where $\xi_{k,h'}(s,a) = 2cH^{-1}\epsilon_k + 2cH^{-1.5}\epsilon\sqrt{y_{k,h'}(s,a) + 4b}$ for all $(s,a) \in \mathcal{S} \times \mathcal{A}$. Note that a product over an empty index set evaluates to 1. Now, we try to bound $\sqrt{y_{k,h'}(s,a) + 4b}$ for all $(s,a) \in \mathcal{S} \times \mathcal{A}$. By the definition of $y_{k,h}(s,a)$ in line 6 of **QVI-4**, we know that there exists a $b'$ satisfying $|b'| \leq b$ such that

$$
\begin{aligned}
\sqrt{y_{k,h'}(s,a) + 4b} &\leq \max\left\{ \left( P_{h'|s,a}^{\mathrm{T}}(V_{k,h'+1}^{(0)})^2 + b - (P_{h'|s,a}^{\mathrm{T}} V_{k,h'+1}^{(0)} - b'/H)^2 + 4b \right)^{1/2}, \sqrt{4b} \right\} \\
&\leq \left( \sigma_{h'}^2(V_{k,h'+1}^{(0)}) + 5b + 2bH^{-1} P_{h'|s,a}^{\mathrm{T}} V_{k,h'+1}^{(0)} \right)^{1/2} \\
&\leq \left( \sigma_{h'}^2(V_{k,h'+1}^{(0)}) + 7b \right)^{1/2}.
\end{aligned} \tag{101}
$$

Since it holds that $V_{k,h'+1}^{(0)}(s) = 0$ for all $s \in \mathcal{S}$ and $h' \in [H]$ when $k = 0$, then $\sigma_{h'}^2(V_{k,h'+1}^{(0)}) = 0$. This implies that

$\sqrt{y_{k,h'}(s,a)+4b} \le \sqrt{7b}$. Then we can show that

$$
\begin{aligned}
Q_h^* - Q_{k,h} &\le \sum_{h'=h}^{H-1}\left(\prod_{i=h+1}^{h'} P_i^{\pi^*}\right)\left(2cH^{-1}\epsilon_k + 2cH^{-1.5}\epsilon\sqrt{y_{k,h'}+4b}\right)\\
&\le 2c\epsilon_k + 2cH^{-0.5}\epsilon\sqrt{7b}\\
&\le 2c\epsilon_k + 2c\epsilon\sqrt{7b}\\
&\le \left(2c + 4c\sqrt{7b}\right)\epsilon_k\\
&\le \epsilon_k.
\end{aligned}
\tag{102}
$$

The second line comes from the fact that $\left\|\sum_{h'=h}^{H-1}\left(\prod_{i=h+1}^{h'} P_i^{\pi^*}\right)\mathbf{1}\right\|_\infty \le H - h \le H$ for all $h \in [H]$. The third line comes from the fact that $H \ge 1$. The fourth line comes from the fact that $\epsilon \le 2\epsilon_k = 2\epsilon_0 = 2H$. The last line comes from the fact that $c = 0.001$ and $b = 1$. Therefore, we have $V_{k,h}(s) \ge V(Q_{k,h})(s) = \max_{a\in\mathcal{A}} Q_{k,h}(s,a) \ge \max_{a\in\mathcal{A}}\{Q_h^*(s,a) - \epsilon_k\} = V_h^*(s) - \epsilon_k$ for the base case $k = 0$.

Now, we assume that for any $k' = 1, \ldots, k-1$, it also holds that $V_{k,h}(s) \ge V_h^*(s) - \epsilon_k$ for all $h \in H$. Then, we proceed to prove the claim for the case of $k' = k$. In fact, the analysis for the case of $k' = k$ is quite similar to the base case, except for the part of the upper bound for $\sqrt{y_{k,h'}(s,a)+4b}$. We can show that there exists a $b'$ satisfying $|b'| \le b$

$$
\begin{aligned}
\sqrt{y_{k,h'}(s,a)+4b} &\le \max\left\{\left((P_{h'|s,a}^{\mathrm{T}}(V_{k,h+1}^{(0)})^2 + b - (P_{h'|s,a}^{\mathrm{T}}V_{k,h+1}^{(0)} - b'/H)^2 + 4b\right)^{1/2}, \sqrt{4b}\right\}\\
&\le \left(\sigma_{h'}^2(V_{k,h'+1}^{(0)}) + 5b + 2bH^{-1}P_{h'|s,a}^{\mathrm{T}}V_{k,h'+1}^{(0)}\right)^{1/2}\\
&\le \left(\sigma_{h'}^2(V_{k,h'+1}^{(0)}) + 7b\right)^{1/2}\\
&\le \sigma_{h'}(V_{k,h'+1}^{(0)}) + \sqrt{7b}\\
&\le \sigma_{h'}(V_{h'+1}^*) + \sigma(V_{k,h'+1}^{(0)} - V_{h'+1}^*) + \sqrt{7b}.
\end{aligned}
\tag{103}
$$

The third line comes from the fact that $V_{k,h'+1}^{(0)}(s) \le H$ for all $s \in \mathcal{S}$. The fourth line comes from the fact that $\sqrt{a+b} \le \sqrt{a} + \sqrt{b}$ when $a, b \ge 0$. The last line comes from the fact that, for any random variables $X$ and $Y$, we must have $\sigma^2(X+Y) = \mathrm{Var}[X+Y] = \mathrm{Var}[X] + \mathrm{Var}[Y] + 2\mathrm{Cov}[X,Y] \le (\sqrt{\mathrm{Var}[X]} + \sqrt{\mathrm{Var}[Y]})^2 = (\sigma(X) + \sigma(Y))^2$. Note that $\sigma(V_{k,h'+1}^{(0)} - V_{h'+1}^*) \le \left\|V_{k,h'+1}^{(0)} - V_{h'+1}^*\right\|_\infty = \left\|V_{k-1,h'+1} - V_{h'+1}^*\right\|_\infty \le \epsilon_{k-1} = 2\epsilon_k$ for all $h' \in [H]$. Therefore, we can show that

$$
\begin{aligned}
Q_h^* - Q_{k,h} &\le \sum_{h'=h}^{H-1}\left(\prod_{i=h+1}^{h'} P_i^{\pi^*}\right)\left(2cH^{-1}\epsilon_k + 2cH^{-1.5}\epsilon\sqrt{y_{k,h'}+4b}\right)\\
&\le 2c\epsilon_k + 2cH^{-1.5}\epsilon\sum_{h'=h}^{H-1}\left(\prod_{i=h+1}^{h'} P_i^{\pi^*}\right)\left(\sigma(V_{h'+1}^*) + \sigma(V_{k,h'+1}^{(0)} - V_{h'+1}^*) + \sqrt{7b}\right)\\
&\le 2c\epsilon_k + 2cH^{-1.5}\epsilon\sum_{h'=h}^{H-1}\left(\prod_{i=h+1}^{h'} P_i^{\pi^*}\right)\left(\sigma(V_{h'+1}^*) + 2\epsilon_k + \sqrt{7b}\right)\\
&\le 2c\epsilon_k + 2c\epsilon + 2cH^{-0.5}\epsilon\left(2\epsilon_k + \sqrt{7b}\right)\\
&\le 2c\left(1 + 2 + 2 + \sqrt{7}\right)\epsilon_k\\
&\le \epsilon_k.
\end{aligned}
\tag{104}
$$

The fourth line comes from the Lemma B.2 and the fact that $\left\|\sum_{h'=h}^{H-1}\left(\prod_{i=h+1}^{h'} P_i^{\pi^*}\right)\mathbf{1}\right\|_\infty \le H - h \le H$ for all $h \in [H]$. The fifth line comes from the fact that we require the input $\epsilon \in (0, \sqrt{H}]$. The last line comes from the fact that $c = 0.001$.

Therefore, we have $V_{k,h}(s) \geq V(Q_{k,h})(s) = \max_{a \in \mathcal{A}} Q_{k,h}(s,a) \geq \max_{a \in \mathcal{A}} \{Q_h^*(s,a) - \epsilon_k\} = V_h^*(s) - \epsilon_k$ for the case of $k' = k$. ∎

### B.2.3. CORRECTNESS OF **QVI-4** (PROOF OF THEOREM 4.5)

By combining Lemma B.4 and Lemma B.3, we can obtain that, for all $k \in [K]$,

$$V_h^* - \epsilon_k \leq V_{k,h} \leq V_h^{\pi_k} \leq V_h^*, \tag{105}$$

$$Q_h^* - \epsilon_k \leq Q_{k,h} \leq Q_h^{\pi_k} \leq Q_h^*, \tag{106}$$

with probability at least $1 - \delta$. When $k = K - 1 = \lceil \log_2(H/\epsilon) \rceil \geq \log_2(H/\epsilon)$, $\epsilon_k = H/2^k \leq \epsilon$. Therefore, it implies that

$$V_h^* - \epsilon \leq V_h^* - \epsilon_{K-1} \leq V_{K-1,h} = \hat{V}_h \leq V_h^{\pi_{K-1}} = V_h^{\hat{\pi}} \leq V_h^*, \tag{107}$$

$$Q_h^* - \epsilon \leq Q_h^* - \epsilon_{K-1} \leq Q_{K-1,h} = \hat{Q}_h \leq Q_h^{\pi_{K-1}} = Q_h^{\hat{\pi}} \leq Q_h^*, \tag{108}$$

with probability at least $1 - \delta$. ∎

### B.2.4. COMPLEXITY OF **QVI-4** (PROOF OF THEOREM 4.6)

**Proof.** The success probability analysis is analogous to Lemma B.3. Hence, we omit it here. We first assume that all estimations are correct, up to the specified error, because the probability that this does not hold is at most $\delta$. Let $C$ be the complexity of **QVI-4**$(\mathcal{M}, \epsilon, \delta)$ as if all estimations are carried out with maximum failure probabilities set to constant. Then, since the actual maximum failure probabilities are set to $\zeta = \delta/(4KHSA)$, the actual complexity of **QVI-4**$(\mathcal{M}, \epsilon, \delta)$ is

$$O\big(C \log(KHSA/\delta)\big). \tag{109}$$

Now, we check each line of **QVI-4**$(\mathcal{M}, \epsilon, \delta)$ to bound $C$.

In line 6, since we have $0 \leq V_{k,h+1}^{(0)}(s) = V_{k-1,h+1}(s) \leq V_{h+1}^*(s) \leq H$ for all $k > 0$ and $0 = V_{k,h+1}^{(0)}(s) = V_{k-1,h+1}(s) \leq V_{h+1}^*(s) \leq H$ for all $s \in \mathcal{S}$ when $k = 0$, therefore, we can use quantum mean estimation algorithm **QME1**, which induces a total query complexity in the order

$$KHSA\left( H^2/b + \sqrt{H^2/b} + H^2/b + \sqrt{H^2/b} \right) = O(KSAH^3). \tag{110}$$

Now, we focus on line 7. By the definition of $y_{k,h}(s,a)$ in line 6, we know that there exists a $b'$ satisfying $|b'| \leq b$ such that

$$
\begin{aligned}
y_{k,h}(s,a) &\geq \max\left\{ P_{h|s,a}^{\mathrm{T}}(V_{k,h+1}^{(0)})^2 - b - \big(P_{h|s,a}^{\mathrm{T}}V_{k,h+1}^{(0)} + b'/H\big)^2, 0 \right\} \\
&\geq P_{h|s,a}^{\mathrm{T}}(V_{k,h+1}^{(0)})^2 - b - \big(P_{h|s,a}^{\mathrm{T}}V_{k,h+1}^{(0)} + b'/H\big)^2 \\
&= \left[\sigma^2(V_{k,h+1}^{(0)})\right]_{(s,a)} - b - (2b'/H)P_{h|s,a}^{\mathrm{T}}V_{k,h+1}^{(0)} - (b')^2/H^2.
\end{aligned}
\tag{111}
$$

This implies that

$$\left[\sigma^2(V_{k,h+1}^{(0)})\right]_{(s,a)} \leq y_{k,h}(s,a) + b + (2b/H)P_{h|s,a}^{\mathrm{T}}V_{k,h+1}^{(0)} + b^2/H^2 \leq y_{k,h}(s,a) + 4b. \tag{112}$$

The last inequality follows from $b = 1$ and $V_{k,h+1}^{(0)}(s) = V_{k-1,h+1}(s) \leq V_{h+1}^*(s) \leq H$ for all $s \in \mathcal{S}$ when $k \geq 1$ and $V_{0,h+1}^{(0)}(s) = 0$ for all $s \in \mathcal{S}$. We also note that, since we have $y_{k,h}(s,a) \geq 0$ (by the definition in line 6), then it holds that $0 < cH^{-1.5}\epsilon\sqrt{y_{k,h}(s,a) + 4b} < 4\sqrt{y_{k,h}(s,a) + 4b}$. Therefore, we can use quantum mean estimation algorithm **QME2** with error $cH^{-1.5}\epsilon\sqrt{y_{k,h}(s,a) + 4b}$ and variance upper bound set to $y_{k,h}(s,a) + 4b$, which induces a total query complexity of order

$$KH \sum_{(s,a) \in \mathcal{S} \times \mathcal{A}} w(s,a) \log^2\big(w(s,a)\big) = O\big(KSAH^{2.5}\epsilon^{-1} \log^2(H^{1.5}/\epsilon)\big), \tag{113}$$

where $w(s,a) = \left(\sqrt{y_{k,h}(s,a) + 4b}\right)\left(cH^{-1.5}\epsilon\sqrt{y_{k,h}(s,a) + 4b}\right)^{-1} = O(H^{1.5}/\epsilon)$.

In line 9, we can bound $0 \leq V_{k,h+1}(s) - V_{k,h+1}^{(0)}(s) \leq V_{h+1}^*(s) - V_{k,h+1}^{(0)}(s) = V_{h+1}^*(s) - V_{k-1,h+1}^{(0)}(s) \leq \epsilon_{k-1} = 2\epsilon_k$ for all $s \in \mathcal{S}$ and $k \geq 1$. When $k = 0$, since $V_{0,h+1}(s) \geq V_{0,h+1}^{(0)}(s) = 0$, then we also have $0 \leq V_{0,h+1}(s) - V_{0,h+1}^{(0)}(s) = V_{0,h+1}(s) \leq V_{h+1}^*(s) \leq H = \epsilon_0$ for all $s \in \mathcal{S}$. Therefore, we can use quantum mean estimation algorithm **QME1** which induces a total query complexity of order

$$KHSA\left(\frac{2\epsilon_k}{cH^{-1}\epsilon_k} + \sqrt{\frac{2\epsilon_k}{cH^{-1}\epsilon_k}}\right) = O(KSAH^2). \tag{114}$$

Therefore, we can show that

$$C = O\left(KSA(H^{2.5}/\epsilon + H^3 + H^2)\log^2(H^{1.5}/\epsilon)\right) = O\left(SA(H^{2.5}/\epsilon + H^3)\log^2(H^{1.5}/\epsilon)\right). \tag{115}$$

Then the total query complexity is

$$O\left(SA(H^{2.5}/\epsilon + H^3)\log^2(H^{1.5}/\epsilon)\log\left(\log(H/\epsilon)HSA/\delta\right)\right). \tag{116}$$

∎

## B.3. Lower Bounds

### B.3.1. INFINITE-HORIZON MDPS

**Preliminaries of infinite-horizon MDPs:** An infinite-horizon MDP is formally defined as a tuple $\tilde{\mathcal{M}} := (\mathcal{S}, \mathcal{A}, P, r, \gamma)$, where $\mathcal{S}$ is a finite set of states representing the possible configurations of the environment, and $\mathcal{A}$ is a finite set of actions available to the agent at each state. The transition probability $P(s'|s,a)$ specifies the likelihood of transitioning to state $s'$ after taking action $a$ in state $s$, ensuring that $\sum_{s' \in \mathcal{S}} P(s'|s,a) = 1$ for all $s \in \mathcal{S}$ and $a \in \mathcal{A}$. The reward function $r(s,a)$, bounded within $[0,1]$, assigns a scalar reward for executing action $a$ in state $s$. Finally, the discount factor $\gamma \in [0,1)$ determines the relative importance of future rewards compared to immediate ones, with $\Gamma := \frac{1}{1-\gamma}$. Given such an MDP, the agent's objective is to select actions that maximize the expected sum of discounted rewards over an infinite time horizon. The primary goal is to compute a policy $\pi : \mathcal{S} \to \mathcal{A}$ that specifies the action $a = \pi(s)$ the agent should take in each state $s \in \mathcal{S}$ to optimize its performance with high probability. For a given policy $\pi$, the state-value function (or V-value) $V^\pi : \mathcal{S} \to [0, \Gamma]$ and the state-action-value function (or Q-value) $Q^\pi : \mathcal{S} \times \mathcal{A} \to [0, \Gamma]$ are defined as follows:

$$V^\pi(s) = \mathbb{E}\left[\sum_{t=0}^\infty \gamma^t r(s_t, a_t)\bigg|\pi, s_0 = s\right], \tag{117}$$

$$Q^\pi(s,a) = \mathbb{E}\left[\sum_{t=0}^\infty \gamma^t r(s_t, a_t)\bigg|\pi, s_0 = s, a_0 = a\right]. \tag{118}$$

A policy $\pi$ is an optimal policy $\pi^*$ if $V^\pi = \max_{\pi \in \Pi} V^\pi = V^{\pi^*}$ where $\Pi$ is the space of all policies. For simplicity, we denote $V^* := V^{\pi^*}$ and $Q^* := Q^{\pi^*}$.

**Optimization goals in infinite-horizon MDPs:** The primary computational objectives in infinite-horizon MDPs are as follows: given an infinite-horizon MDP $\tilde{\mathcal{M}}$, an approximation error $\epsilon$, and a failure probability $\delta$, the goal is to compute $\epsilon$-estimates $\hat{\pi}$, $\hat{V}$, and $\hat{Q}$ such that $\|V^{\hat{\pi}} - V^*\|_\infty \leq \epsilon$, $\|\hat{V} - V^*\|_\infty \leq \epsilon$, and $\|\hat{Q} - Q^*\|_\infty \leq \epsilon$ with a probability of at least $1 - \delta$.

**Classical generative model for infinite-horizon MDPs:** We denote the classical generative model for infinite-horizon MDPs as $\tilde{G}$. Assuming access to $\tilde{G}$, one can collect $N$ independent samples

$$s_{s,a}^i \overset{\text{i.i.d.}}{\sim} P(\cdot|s,a), \quad i = 1, \ldots, N,$$

for each state-action pair $(s,a) \in \mathcal{S} \times \mathcal{A}$.

**Theorem B.5** (Classical and quantum lower bounds for infinite-horizon MDP (Wang et al., 2021)). *Fix any integers $S, A \geq 2$ and $\gamma \in [0.9, 1)$. Let $\Gamma = (1 - \gamma)^{-1} \geq 10$ and fix any $\epsilon \in (0, \Gamma/4)$. There exists an infinite-horizon MDP $\mathcal{M} = (\mathcal{S}, \mathcal{A}, P, r, \gamma)$ with $S$ states, $A$ actions, and discount parameter $\gamma$ such that the following lower bound hold:*

- *Given access to a classical generative oracle $\tilde{G}$, any algorithm that computes an $\epsilon$-approximation to $Q^*$, $V^*$, or $\pi^*$ must make $\Omega(\frac{SA\Gamma^3}{\epsilon^2})$ queries.*

- *Given access to a quantum generative oracle $\tilde{\mathcal{G}}$ defined as*

$$\tilde{\mathcal{G}} : |s\rangle \otimes |a\rangle \otimes |0\rangle \otimes |0\rangle \mapsto |s\rangle \otimes |a\rangle \otimes \left( \sum_{s' \in \mathcal{S}} \sqrt{P(s'|s,a)} |s'\rangle \otimes |v_{s'}\rangle \right), \tag{119}$$

*where $|v_{s'}\rangle$ are arbitrary auxiliary states, any algorithm that computes an $\epsilon$-approximation to $Q^*$ must take $\Omega(\frac{SA\Gamma^{1.5}}{\epsilon})$ queries and any algorithm that computes an $\epsilon$-approximation to $V^*$ or $\pi^*$ must take $\Omega(\frac{S\sqrt{A}\Gamma^{1.5}}{\epsilon})$ queries.*

### B.3.2. FINITE-HORIZON MDPS

**Lemma B.6.** *Let $\mathcal{S}$ and $\mathcal{A}$ be finite sets of states and actions. Let $H > 0$ be a positive integer and $\epsilon \in (0, 1/2)$ be an error parameter. We consider the following finite-horizon MDP $\mathcal{M} := (\mathcal{S}, \mathcal{A}, \{P_h\}_{h=0}^{H-1}, \{r_h\}_{h=0}^{H-1}, H)$ where $P_h = P \in \mathbb{R}^{\mathcal{S} \times \mathcal{A} \times \mathcal{S}}$ and $r_h = r \in [0, 1]^{\mathcal{S} \times \mathcal{A}}$ for all $h \in [H]$.*

- *Given access to a classical generative model, any algorithm $\mathcal{K}$, which takes $\mathcal{M}$ as an input and outputs a value function $\hat{V}_0$ such that $\left\| \hat{V}_0 - V_0^* \right\|_\infty \leq \epsilon$ with probability at least $0.9$, needs to call the classical generative oracle at least*

$$\Omega\left( \frac{SAH^3}{\epsilon^2 \log^3(\epsilon^{-1})} \right) \tag{120}$$

*times on the worst case of input $\mathcal{M}$.*

- *Given access to a quantum generative oracle $\mathcal{G}$ defined in Definition 4.1 any algorithm $\mathcal{K}$, which takes $\mathcal{M}$ as an input and outputs a value function $\hat{V}_0$ such that $\left\| \hat{V}_0 - V_0^* \right\|_\infty \leq \epsilon$ with probability at least $0.9$, needs to call the quantum generative oracle at least*

$$\Omega\left( \frac{S\sqrt{A}H^{1.5}}{\epsilon \log^{1.5}(\epsilon^{-1})} \right) \tag{121}$$

*times on the worst case of input $\mathcal{M}$.*

**Proof.** We first introduce some definitions about infinite horizon MDPs. Let $s_0 \in \mathcal{S}$ to be a state. Suppose we have an infinite-horizon MDP $\tilde{\mathcal{M}} = (\tilde{\mathcal{S}}, \tilde{\mathcal{A}}, \tilde{P}, \tilde{r}, \gamma)$ with a quantum generative oracle, where $\tilde{\mathcal{S}} = \mathcal{S} \setminus \{s_0\}$ to be a subset of $\mathcal{S}$ and $\gamma \in [0, 1)$. For a better differentiation on the notations between finite-horizon and infinite-horizon MDPs, we let $\tilde{V}^* \in \mathbb{R}^{\mathcal{S}}$ represent the optimal V-value function of $\tilde{\mathcal{M}}$. First, we define a Bellman operator $\mathcal{T}$ for the infinite-horizon MDP $\tilde{\mathcal{M}}$ satisfying, for any $u \in \mathbb{R}^{\tilde{\mathcal{S}}}$ and $s \in \tilde{\mathcal{S}}$,

$$\mathcal{T}(u)_s = \max_{a \in \tilde{\mathcal{A}}} \left[ \tilde{r}(s, a) + \gamma \sum_{s' \in \tilde{\mathcal{S}}} \tilde{P}(s'|s, a) u(s') \right]. \tag{122}$$

Note that for any $u, v \in \mathbb{R}^{\tilde{\mathcal{S}}}$ satisfying $u(s) \leq v(s)$ for all $s \in \tilde{\mathcal{S}}$, we have $\mathcal{T}(u)_s \leq \mathcal{T}(v)_s$ for all $s \in \tilde{\mathcal{S}}$. This is the so-called monotonicity property of $\mathcal{T}$. Besides, it also holds that $\mathcal{T}(\tilde{V}^*)_s = \tilde{V}^*(s)$ for all $s \in \tilde{\mathcal{S}}$.

Now, we proceed to prove that obtaining an $2\epsilon$-approximation value of $\tilde{V}^*$ for any infinite horizon MDP $\tilde{\mathcal{M}}$ can be reduced to obtaining an $\epsilon$-approximation value of $V_0^*$ for a finite horizon MDP. We consider the following finite-horizon MDP $\mathcal{M} = (\mathcal{S}, \mathcal{A}, \{P_h\}_{h=0}^{H-1}, \{r_h\}_{h=0}^{H-1}, H)$ where $P_h = P \in \mathbb{R}^{\mathcal{S} \times \mathcal{A} \times \mathcal{S}}$ and $r_h = r \in [0, 1]^{\mathcal{S} \times \mathcal{A}}$. Besides, the time horizon $H$ satisfies $H = \lceil 2(1 - \gamma)^{-1} \log(2\epsilon^{-1}) \rceil = \Theta((1 - \gamma)^{-1} \log(\epsilon^{-1}))$. Besides, under any action $a \in \mathcal{A} = \tilde{\mathcal{A}}$, there is a $(1 - \gamma)$

probability for each state $s \in \tilde{\mathcal{S}}$ to transition to $s_0$ and $\gamma$ probability to follow the original transitions in $\tilde{\mathcal{M}}$. However, when the agent is in $s_0$, it can only transition to itself with probability 1, no matter which action $a \in \mathcal{A}$ it takes. Hence, $s_0$ is an absorbing state in $\mathcal{M}$. Overall, we have the following definitions for the transition probability kernel $P$ in $\mathcal{M}$.

$$\forall s, s' \in \tilde{\mathcal{S}}, a \in \mathcal{A}, P(s'|s,a) = \gamma \tilde{P}(s'|s,a), P(s_0|s,a) = (1-\gamma), \tag{123}$$

$$P(s'|s_0,a) = 0, P(s_0|s_0,a) = 1. \tag{124}$$

Besides, we define $r(s_0,a) = 0, r(s,a) = \tilde{r}(s,a) \in [0,1]$ for all $s \in \tilde{\mathcal{S}}$ and $a \in \mathcal{A}$.

Now, we proceed to prove that $\left\| V_0^*|_{\tilde{\mathcal{S}}} - \tilde{V}^* \right\|_\infty \leq \epsilon$, i.e., $|V_0^*(s) - \tilde{V}^*(s)| \leq \epsilon$ for all $s \in \tilde{\mathcal{S}}$. First, we note that $V_{H-1}^* = \max_{a \in \mathcal{A}} r(s,a) \leq \tilde{V}^*$. Then, by the monotonicity of the $\mathcal{T}$ operator, we have $\mathcal{T}(V_{H-1}^*)_s \leq \mathcal{T}(\tilde{V}^*)_s = \tilde{V}^*(s)$ for all $s \in \tilde{\mathcal{S}}$. In fact, by the definition of $P$ in $\mathcal{M}$, we have

$$
\begin{aligned}
\forall s \in \tilde{\mathcal{S}}, \mathcal{T}(V_{H-1}^*)_s &= \max_{a \in \mathcal{A}} \left[ \tilde{r}(s,a) + \gamma \sum_{s' \in \tilde{\mathcal{S}}} \tilde{P}(s'|s,a) V_{H-1}^*(s') \right] \\
&= \max_{a \in \mathcal{A}} \left[ r(s,a) + \sum_{s' \in \tilde{\mathcal{S}}} P(s'|s,a) V_{H-1}^*(s') + P(s_0|s,a) V_{H-1}^*(s_0) \right] \\
&= \max_{a \in \mathcal{A}} \left[ r(s,a) + \sum_{s' \in \mathcal{S}} P(s'|s,a) V_{H-1}^*(s') \right] \\
&= V_{H-2}^*(s).
\end{aligned}
\tag{125}
$$

The second line above comes from the fact that $V_{H-1}^*(s_0) = \max_{a \in \mathcal{A}} r(s_0,a) = 0$. By induction, we have $V_h^*(s_0) = \max_{a \in \mathcal{A}}[r(s_0,a) + P(s_0|s_0,a) V_{h+1}^*(s_0)] = 0$ for all $h \in [H]$. Hence, we have $V_{H-2}^*(s) \leq \tilde{V}^*(s)$ for all $s \in \tilde{\mathcal{S}}$. By induction, we have $V_h^*(s) \leq \tilde{V}^*(s)$ for all $h \in [H]$ and $s \in \tilde{\mathcal{S}}$. In particular, we have $V_0^*(s) \leq \tilde{V}^*(s)$ for all $s \in \tilde{\mathcal{S}}$. Let $\tilde{\pi}^* \in \mathcal{A}^{\mathcal{S}}$ be an optimal policy for the infinite-horizon MDP $\tilde{\mathcal{M}}$. However, $\tilde{\pi} \in \mathcal{A}^{\mathcal{S} \times [H]}$, where $\tilde{\pi}(\cdot, h) = \tilde{\pi}^*$ for all $h \in [H]$, may not be an optimal policy for finite-horizon MDP $\mathcal{M}$. Then we have $V_0^{\tilde{\pi}}(s) \leq V_0^*(s)$ for all $s \in \mathcal{S}$. In fact, for any $s \in \tilde{\mathcal{S}}$, we have

$$
\begin{aligned}
V_0^{\tilde{\pi}}(s) &= r\big(s, \tilde{\pi}^*(s)\big) + \sum_{s' \in \mathcal{S}} P(s'|s, \tilde{\pi}^*(s)) r\big(s, \tilde{\pi}^*(s)\big) + \cdots + \sum_{s' \in \mathcal{S}} P^H\big(s'|s, \tilde{\pi}^*(s)\big) r\big(s, \tilde{\pi}^*(s)\big) \\
&= \tilde{r}\big(s, \tilde{\pi}^*(s)\big) + \gamma \sum_{s' \in \tilde{\mathcal{S}}} \tilde{P}(s'|s, \tilde{\pi}^*(s)) \tilde{r}\big(s, \tilde{\pi}^*(s)\big) + \cdots + \gamma^H \sum_{s' \in \tilde{\mathcal{S}}} \tilde{P}^H\big(s'|s, \tilde{\pi}^*(s)\big) \tilde{r}\big(s, \tilde{\pi}^*(s)\big) \\
&= \tilde{V}_H^{\tilde{\pi}^*},
\end{aligned}
\tag{126}
$$

where $\tilde{V}_H^{\tilde{\pi}^*}$ is the V-value induced by the policy $\tilde{\pi}^*$ over $H$ iterations. Note that for any policy $\tilde{\pi}$ for the infinite horizon MDP $\tilde{\mathcal{M}}$, $\left\| \tilde{V}_k^{\pi} - \tilde{V}^{\pi} \right\|_\infty \leq \gamma^k \left\| \tilde{V}_0^{\pi} - \tilde{V}^{\pi} \right\|_\infty \leq \gamma^k \left( \left\| \tilde{V}_0^{\pi} \right\|_\infty + \left\| \tilde{V}^{\pi} \right\|_\infty \right) \leq 2\exp\big(-(1-\gamma)k\big)/(1-\gamma)$. The last inequality follows from $\left\| \tilde{V}_0^{\pi} \right\|_\infty \leq 1/(1-\gamma)$ and $\left\| \tilde{V}^{\pi} \right\|_\infty \leq 1/(1-\gamma)$. Besides, combining the fact that $\log(\gamma) \leq \gamma - 1$ for all $\gamma \in (0,1)$ and $\exp(x)$ is monotonically increasing, we can induce the inequalities $k \log(\gamma) \leq -k(1-\gamma)$ and $\gamma^k = \exp(k \log(\gamma)) \leq \exp(-k(1-\gamma))$. Then, $\forall \epsilon > 0$, it suffices to let $k \geq \log(2/((1-\gamma)\epsilon))/(1-\gamma)$ so that $\left\| \tilde{V}_k^{\pi} - \tilde{V}^{\pi} \right\|_\infty \leq \epsilon$. In fact,

$$
\begin{aligned}
\frac{1}{1-\gamma} \log\left( \frac{2}{(1-\gamma)\epsilon} \right) &= \frac{1}{1-\gamma} \left( \log\left( \frac{1}{1-\gamma} \right) + \log\left( \frac{2}{\epsilon} \right) \right) \\
&= \frac{1}{1-\gamma} \left( -\log(1-\gamma) + \log\left( \frac{2}{\epsilon} \right) \right) \\
&\leq \frac{1}{1-\gamma} \left( \gamma + \log\left( \frac{2}{\epsilon} \right) \right) \\
&\leq \frac{2}{1-\gamma} \log\left( \frac{2}{\epsilon} \right).
\end{aligned}
\tag{127}
$$

The third line comes from the fact that $\log(1-\gamma) \leq -\gamma, \forall \gamma \in [0,1)$. The last line comes from the fact that $\log(2/\epsilon) > 1, \forall \epsilon \in (0, 1/2)$. Since we have $H = \lceil \frac{2}{1-\gamma} \log\left(\frac{2}{\epsilon}\right) \rceil$ and $\tilde{\pi}^*$ is an optimal policy for $\tilde{\mathcal{M}}$, then we must have $V_0^{\tilde{\pi}}(s) = \tilde{V}_H^{\tilde{\pi}^*}(s) \geq \tilde{V}^{\tilde{\pi}^*}(s) - \epsilon = \tilde{V}^*(s) - \epsilon$. Therefore, we have $\tilde{V}^*(s) - \epsilon \leq V_0^*(s) \leq \tilde{V}^*(s)$ for all $s \in \tilde{\mathcal{S}}$, which implies $\left\| \hat{V}_0^*|_{\tilde{\mathcal{S}}} - \tilde{V}^* \right\|_\infty \leq \epsilon$.

Therefore, an $\epsilon$-approximation of $V_0^*$ will give an $2\epsilon$-approximation to $\tilde{V}^*$. Specifically, if we let $\hat{V}_0$ be an $\epsilon$-approximation of $V_0^*$, then

$$
\begin{aligned}
\left\| \hat{V}_0|_{\tilde{\mathcal{S}}} - \tilde{V}^* \right\|_\infty &\leq \left\| \hat{V}_0|_{\tilde{\mathcal{S}}} - V_0^*|_{\tilde{\mathcal{S}}} \right\|_\infty + \left\| V_0^*|_{\tilde{\mathcal{S}}} - \tilde{V}^* \right\|_\infty \\
&\leq \left\| \hat{V}_0 - V_0^* \right\|_\infty + \left\| V_0^*|_{\tilde{\mathcal{S}}} - \tilde{V}^* \right\|_\infty \\
&\leq 2\epsilon.
\end{aligned}
\tag{128}
$$

Therefore, obtaining $2\epsilon$-approximation $\tilde{V}^*$ for $\tilde{\mathcal{M}}$ with a quantum generative oracle reduced to obtaining $\epsilon$-approximation value $\hat{V}_0^*$ for $\mathcal{M}$ with a quantum generative oracle. Then, it implies that the algorithm $\mathcal{K}$ inherits the lower bound for obtaining $2\epsilon$-approximation $\tilde{V}^*$ for $\tilde{\mathcal{M}}$ with a quantum generative oracle. Note that $\mathcal{M}$ is a time-independent MDP. Then the quantum generative oracle $\mathcal{G}$ is the same as $\tilde{\mathcal{G}}$ defined in Theorem B.5. By Theorem B.5, we know that the lower bound for obtaining $2\epsilon$-approximation $\tilde{V}^*$ for $\tilde{\mathcal{M}}$ with a quantum generative oracle is $\Omega(S\sqrt{A}\Gamma^{1.5}/\epsilon)$. This implies the quantum lower bound for finite horizon MDP $\mathcal{M}$ to obtain an $\epsilon$-optimal value function $\hat{V}_0$ is $\Omega(S\sqrt{AH^{1.5}}/(\epsilon \log^{1.5}(\epsilon^{-1})))$.

Note that the above content also shows that obtaining $2\epsilon$-approximation $\tilde{V}^*$ for $\tilde{\mathcal{M}}$ with a classical generative oracle reduced to obtaining $\epsilon$-approximation value $\hat{V}_0^*$ for $\mathcal{M}$ with a classical generative oracle. By Theorem B.5, we know that the lower bound for obtaining $2\epsilon$-approximation $\tilde{V}^*$ for $\tilde{\mathcal{M}}$ with a classical generative oracle is $\Omega(SA\Gamma^3/\epsilon)$. Therefore, the classical lower bound for finite horizon MDP $\mathcal{M}$ to obtain an $\epsilon$-optimal value function $\hat{V}_0$ is $\Omega(SAH^3/(\epsilon^2 \log^3(\epsilon^{-1})))$. $\blacksquare$

**Lemma B.7.** *Let $\mathcal{S}$ and $\mathcal{A}$ be finite sets of states and actions. Let $H > 0$ be a positive integer and $\epsilon \in (0, 1/2)$ be an error parameter. We consider the following finite-horizon MDP $\mathcal{M} := (\mathcal{S}, \mathcal{A}, \{P_h\}_{h=0}^{H-1}, \{r_h\}_{h=0}^{H-1}, H)$ where $P_h = P \in \mathbb{R}^{\mathcal{S} \times \mathcal{A} \times \mathcal{S}}$ and $r_h = r \in [0,1]^{\mathcal{S} \times \mathcal{A}}$ for all $h \in [H]$.*

- *Given access to a classical generative oracle, any algorithm $\mathcal{K}$, which takes $\mathcal{M}$ as an input and outputs a value function $\hat{Q}_0$ such that $\left\| \hat{Q}_0 - Q_0^* \right\|_\infty \leq \epsilon$ with probability at least $0.9$, needs to call the classical generative oracle at least*

$$
\Omega\left( \frac{SAH^3}{\epsilon^2 \log^3(\epsilon^{-1})} \right)
\tag{129}
$$

  *times on the worst case of input $\mathcal{M}$.*

- *Given access to a quantum generative oracle $\mathcal{G}$ defined in Definition 4.1 any algorithm $\mathcal{K}$, which takes $\mathcal{M}$ as an input and outputs a value function $\hat{Q}_0$ such that $\left\| \hat{Q}_0 - Q_0^* \right\|_\infty \leq \epsilon$ with probability at least $0.9$, needs to call the quantum generative oracle at least*

$$
\Omega\left( \frac{SAH^{1.5}}{\epsilon \log^{1.5}(\epsilon^{-1})} \right)
\tag{130}
$$

  *times on the worst case of input $\mathcal{M}$.*

**Proof.** Following the same idea in Lemma B.6, we consider an infinite-horizon MDP $\tilde{\mathcal{M}} = (\tilde{\mathcal{S}}, \tilde{\mathcal{A}}, \tilde{P}, \tilde{r}, \gamma)$ with a quantum generative oracle, where $\tilde{\mathcal{S}} = \mathcal{S} \setminus \{s_0\}$ to be a subset of $\mathcal{S}$ and $\gamma \in [0,1)$. With a slight abuse of the notations for the infinite-horizon MDPs, we let $\tilde{V}^* \in \mathbb{R}^{\mathcal{S}}$ and $\tilde{Q}^* \in \mathbb{R}^{\mathcal{S} \times \mathcal{A}}$ be the optimal V-value and Q-value functions of $\tilde{\mathcal{M}}$. Now, we proceed to prove that obtaining an $2\epsilon$-approximation value of $\tilde{Q}^*$ for any infinite horizon MDP $\tilde{\mathcal{M}}$ can be reduced to obtaining an $\epsilon$-approximation value of $Q_0^*$ for a finite horizon MDP. We consider following finite-horizon MDP $\mathcal{M} = (\mathcal{S}, \mathcal{A}, \{P_h\}_{h=0}^{H-1}, \{r_h\}_{h=0}^{H-1}, H)$ where $P_h = P \in \mathbb{R}^{\mathcal{S} \times \mathcal{A} \times \mathcal{S}}$ and $r_h = r \in \mathbb{R}^{\mathcal{S} \times \mathcal{A}}$. Besides, the time horizon $H$ satisfies $H = \lceil 2(1-\gamma)^{-1} \log(\epsilon^{-1}) \rceil = \Theta((1-\gamma)^{-1} \log(\epsilon^{-1}))$. Besides, under any action $a \in \mathcal{A} = \tilde{\mathcal{A}}$, there is a $(1-\gamma)$ probability for each state $s \in \tilde{\mathcal{S}}$ to transition to $s_0$ and $\gamma$ probability to follow the original transitions in $\tilde{\mathcal{M}}$. However, when

the agent is in $s_0$, it can only transition to itself with probability 1, no matter which action $a \in \mathcal{A}$ it takes. Hence, $s_0$ is an absorbing state in $\mathcal{M}$. Overall, we have the following definitions for the transition probability kernel $P$ in $\mathcal{M}$.

$$\forall s, s' \in \tilde{\mathcal{S}}, a \in \mathcal{A}, P(s'|s,a) = \gamma \tilde{P}(s'|s,a), P(s_0|s,a) = (1 - \gamma), \tag{131}$$

$$P(s'|s_0, a) = 0, P(s_0|s_0, a) = 1. \tag{132}$$

Besides, we define $r(s_0, a) = 0, r(s,a) = \tilde{r}(s,a) \in [0,1]$ for all $s \in \tilde{\mathcal{S}}$ and $a \in \mathcal{A}$.

Now, we proceed to prove that $\left\| Q_0^*|_{\tilde{\mathcal{S}} \times \mathcal{A}} - \tilde{Q}^* \right\|_\infty \leq \epsilon$, i.e., $|Q_0^*(s,a) - \tilde{Q}^*(s,a)| \leq \epsilon$ for all $s \in \tilde{\mathcal{S}}$ and $a \in \tilde{\mathcal{A}} = \mathcal{A}$. First, we note that $Q_{H-1}^* = r(s,a) \leq \tilde{Q}^*$ by the definition of $\tilde{Q}^*$. In Lemma B.6, we see that it holds that $V_h^*(s) \leq \tilde{V}^*(s)$ for all $h \in [H]$ and $s \in \tilde{\mathcal{S}}$, and $V_{h+1}^*(s_0) = 0$ for all $h \in [H]$. Therefore, we have,

$$\begin{aligned}
Q_h^*(s,a) &= r(s,a) + \sum_{s' \in \mathcal{S}} P(s'|s,a) V_{h+1}^*(s') \\
&= r(s,a) + \sum_{s' \in \tilde{\mathcal{S}}} P(s'|s,a) V_{h+1}^*(s') + P(s_0|s,a) V_{h+1}^*(s_0) \\
&= r(s,a) + \sum_{s' \in \tilde{\mathcal{S}}} P(s'|s,a) V_{h+1}^*(s') \\
&= r(s,a) + \gamma \sum_{s' \in \tilde{\mathcal{S}}} \tilde{P}(s'|s,a) V_{h+1}^*(s') \\
&\leq r(s,a) + \gamma \sum_{s' \in \tilde{\mathcal{S}}} \tilde{P}(s'|s,a) \tilde{V}^*(s') \\
&= \tilde{Q}^*(s,a),
\end{aligned} \tag{133}$$

for all $h \in [H-1]$ and $(s,a) \in \tilde{\mathcal{S}} \times \mathcal{A}$. In particular, $Q_0^*(s,a) \leq \tilde{Q}^*(s,a)$ for all $s \in \tilde{\mathcal{S}}$ and $a \in \mathcal{A}$. Let $\tilde{\pi}^* \in \mathcal{A}^{\mathcal{S}}$ be an optimal policy for the infinite-horizon MDP $\tilde{\mathcal{M}}$. However, $\tilde{\pi} \in \mathcal{A}^{\mathcal{S} \times [H]}$, where $\tilde{\pi}(\cdot, h) = \tilde{\pi}^*$ for all $h \in [H]$, may not be an optimal policy for finite-horizon MDP $\mathcal{M}$. Then we have $Q_0^{\tilde{\pi}}(s,a) \leq Q_0^*(s,a)$ for all $s \in \mathcal{S}$ and $a \in \mathcal{A}$. In fact, for any $s \in \tilde{\mathcal{S}}$, we have

$$\begin{aligned}
Q_0^{\tilde{\pi}}(s,a) &= r(s,a) + \sum_{s' \in \mathcal{S}} P(s'|s,a) r\big(s, \tilde{\pi}^*(s)\big) + \cdots + \sum_{s' \in \mathcal{S}} P^H\big(s'|s, \tilde{\pi}^*(s)\big) r\big(s, \tilde{\pi}^*(s)\big) \\
&= \tilde{r}(s,a) + \gamma \sum_{s' \in \tilde{\mathcal{S}}} \tilde{P}(s'|s,a) \tilde{r}\big(s, \tilde{\pi}^*(s)\big) + \cdots + \gamma^H \sum_{s' \in \tilde{\mathcal{S}}} \tilde{P}^H(s'|s,a) \tilde{r}\big(s, \tilde{\pi}^*(s)\big) \\
&= \tilde{Q}_H^{\tilde{\pi}^*},
\end{aligned} \tag{134}$$

where $\tilde{Q}_H^{\tilde{\pi}^*}$ is the Q value of the infinite-horizon MDP $\tilde{\mathcal{M}}$ induced by the policy $\tilde{\pi}^*$ over $H$ iterations. Note that for any policy $\tilde{\pi}$ for the infinite horizon MDP $\tilde{\mathcal{M}}$, $\left\| \tilde{Q}_k^\pi - \tilde{Q}^\pi \right\|_\infty \leq \gamma^k \left\| \tilde{Q}_0^\pi - \tilde{Q}^\pi \right\|_\infty \leq 2\exp(-(1-\gamma)k)/(1-\gamma)$. Then, $\forall \epsilon$, it suffices to let $k \geq \log(2/((1-\gamma)\epsilon))/(1-\gamma)$ so that $\left\| \tilde{Q}_k^\pi - \tilde{Q}^\pi \right\|_\infty \leq \epsilon$. In fact,

$$\begin{aligned}
\frac{1}{1-\gamma} \log\left(\frac{2}{(1-\gamma)\epsilon}\right) &= \frac{1}{1-\gamma}\left(\log\left(\frac{1}{1-\gamma}\right) + \log\left(\frac{2}{\epsilon}\right)\right) \\
&= \frac{1}{1-\gamma}\left(-\log(1-\gamma) + \log\left(\frac{2}{\epsilon}\right)\right) \\
&\leq \frac{1}{1-\gamma}\left(\gamma + \log\left(\frac{2}{\epsilon}\right)\right) \\
&\leq 2\frac{1}{1-\gamma} \log\left(\frac{2}{\epsilon}\right).
\end{aligned} \tag{135}$$

The third line comes from the fact that $\log(1-\gamma) \leq -\gamma, \forall \gamma \in [0,1)$. The last line comes from the fact that $\log(2/\epsilon) > 1, \forall \epsilon \in (0, 1/2)$. Since $H = \lceil 2\frac{1}{1-\gamma} \log\left(\frac{2}{\epsilon}\right) \rceil$ and $\tilde{\pi}^*$ is an optimal policy for $\tilde{\mathcal{M}}$, then we must have $\tilde{Q}_H^{\tilde{\pi}^*}(s,a) \geq$

$\tilde{Q}^{\tilde{\pi}^*}(s, a) - \epsilon = \tilde{Q}^*(s, a) - \epsilon$. Therefore, we have $\tilde{Q}^*(s, a) - \epsilon \le Q_0^*(s, a) \le \tilde{Q}^*(s, a)$ for all $s \in \tilde{\mathcal{S}}, a \in \mathcal{A}$, which implies $\left\| \hat{Q}_0^*|_{\tilde{\mathcal{S}} \times \mathcal{A}} - \tilde{Q}^* \right\|_\infty \le \epsilon$. Therefore, an $\epsilon$-approximation of $Q_0^*$ will give an $2\epsilon$-approximation to $\tilde{Q}^*$. Specifically, if we let $\hat{Q}_0$ be an $\epsilon$-approximation of $Q_0^*$, then

$$
\begin{aligned}
\left\| \hat{Q}_0|_{\tilde{\mathcal{S}} \times \mathcal{A}} - \tilde{Q}^* \right\|_\infty &\le \left\| \hat{Q}_0|_{\tilde{\mathcal{S}} \times \mathcal{A}} - Q_0^*|_{\tilde{\mathcal{S}} \times \mathcal{A}} \right\|_\infty + \left\| Q_0^*|_{\tilde{\mathcal{S}} \times \mathcal{A}} - \tilde{Q}^* \right\|_\infty \\
&\le \left\| \hat{Q}_0 - Q_0^* \right\|_\infty + \left\| Q_0^*|_{\tilde{\mathcal{S}} \times \mathcal{A}} - \tilde{Q}^* \right\|_\infty \\
&\le 2\epsilon.
\end{aligned}
\tag{136}
$$

Therefore, obtaining $2\epsilon$-approximation $\tilde{Q}^*$ for $\tilde{\mathcal{M}}$ with a quantum generative oracle reduced to obtaining $\epsilon$-approximation value $\hat{Q}_0^*$ for $\mathcal{M}$ with a quantum generative oracle. Then, it implies that the algorithm $\mathcal{K}$ inherits the lower bound for obtaining $2\epsilon$-approximation $\tilde{Q}^*$ for $\tilde{\mathcal{M}}$ with a quantum generative oracle. Note that $\mathcal{M}$ is a time-independent MDP. Then the quantum generative oracle $\mathcal{G}$ is the same as $\tilde{\mathcal{G}}$ defined in Theorem B.5. By Theorem B.5, we know that the lower bound for obtaining $2\epsilon$-approximation $\tilde{Q}^*$ for $\tilde{\mathcal{M}}$ with a quantum generative oracle is $\Omega(SA\Gamma^{1.5}/\epsilon)$. This implies the lower bound for obtaining $\epsilon$-optimal Q value function $\hat{Q}_0$ of finite horizon MDP $\mathcal{M}$ is $\Omega(SAH^{1.5}/(\epsilon \log^{1.5}(\epsilon^{-1})))$.

Note that the above content also implies that obtaining $2\epsilon$-approximation $\tilde{Q}^*$ for $\tilde{\mathcal{M}}$ with a classical generative oracle reduced to obtaining $\epsilon$-approximation value $\hat{Q}_0^*$ for $\mathcal{M}$ with a classical generative oracle. By Theorem B.5, we know that the lower bound for obtaining $2\epsilon$-approximation $\tilde{Q}^*$ for $\tilde{\mathcal{M}}$ with a classical generative oracle is $\Omega(SA\Gamma^3/\epsilon)$. Therefore, the classical lower bound for finite horizon MDP is $\Omega(SAH^3/(\epsilon^2 \log^3(\epsilon^{-1})))$. ∎

### B.3.3. Lower Bounds for Finite-horizon MDPs (Proof of Theorem 4.7)

**Proof.** Since time-independent and finite-horizon MDP is a special case of time-dependent and finite-horizon MDP, we know that the lower bound of obtaining an $\epsilon$-approximation $\hat{V}_0$ of $V_0^*$ for time-dependent and finite-horizon MDP $\mathcal{M}$ with a classical or quantum generative oracle inherits the corresponding lower bound in Lemma B.6. Besides, obtaining $\epsilon$-approximations $\hat{V}_0$ of $V_0^*$ is a sub-task of obtaining $\epsilon$-approximations $\hat{V}_h$ of $V_h^*$ for all $h \in [H]$. Therefore, the lower bound of obtaining $\epsilon$-approximations $\hat{V}_h$ of $V_h^*$ for all $h \in [H]$ for time-dependent and finite-horizon MDP $\mathcal{M}$ with access to a classical or quantum generative oracle inherits the lower bound of obtaining $\epsilon$-approximations $\hat{V}_0$ of $V_0^*$ with a classical or quantum generative oracle. Therefore, algorithm $\mathcal{K}$ has the desired classical and quantum lower bounds for obtaining $\epsilon$-optimal V value functions $\{\hat{V}_h\}_{h=0}^{H-1}$. With Lemma B.7, similar idea also applies to obtain the classical and quantum lower bound of obtaining $\epsilon$-optimal Q value functions $\{\hat{Q}_h\}_{h=0}^{H-1}$.

Suppose $\mathcal{K}$ can output an $\epsilon$-optimal policy $\hat{\pi}$ for a finite horizon and time-dependent MDP $\mathcal{M}$, then the corresponding V-values $\{\hat{V}_h\}_{h=0}^{H-1} := \{V_h^{\hat{\pi}}\}_{h=0}^{H-1}$ induced by $\hat{\pi}$ are $\epsilon$-optimal. Therefore, $\mathcal{K}$ has the desired classical and quantum lower bounds for obtaining the $\epsilon$-optimal policy $\hat{\pi}$ by inheriting the corresponding lower bound for obtaining $\epsilon$-optimal V-value functions $\{\hat{V}_h\}_{h=0}^{H-1}$. ∎

