# OpenReview forum: "Quantum Algorithms for Finite-horizon Markov Decision Processes"
_ICML.cc/2025/Conference — ICML 2025 poster_

### Official Review · Reviewer_FX1s · 2025-03-11

**Overall Recommendation:** 4

**Summary:**

This paper presents four quantum algorithms for time-dependent, finite-horizon Markov Decision processes (MDPs) in both the exact dynamics setting and the generative model setting:

1. In the exact dynamics setting, the algorithm QVI-1 achieves a quadratic speedup in the active space size (A) for computing the optimal policy and V-value. QVI-2 provides a sub-quadratic quantum speedup in the state space (S).

2. In the diffusion model setting, the algorithm QVI-3 and QVI-4 achieves speedups in terms of A (sub-quadratic) and estimation error $\epsilon$ (quadratic).

**Claims And Evidence:**

The submission claims the correctness and efficiency of the proposed quantum algorithms. Their correctness and complexity bounds are rigorously proved in the manuscript. Meanwhile, the authors also establish a quantum lower bound for finite-horizon MDPs, demonstrating that QVI-3 and QVI-4 are nearly minimax optimal.

**Essential References Not Discussed:**

A few references are highly relevant to quantum accelerations of optimal control/RL but are not discussed by the author.

- The authors claim that "However, the quantum algorithm and analysis there cannot be applied to general finite-horizon MDPs." This paper (https://arxiv.org/abs/2206.04741) proposes amplitude estimation to estimate the value function of a policy in a finite MDP with a finite horizon.

- (Cornelissen, 2018) is not the only work that improves policy gradient. This paper (https://arxiv.org/abs/2411.01391) provides a super-quadratic improvement in policy gradient estimation based on the quantum numerical linear algebra (ODE solvers + LCU) technique.

**Experimental Designs Or Analyses:**

No numerical experiment is presented in this submission.

**Methods And Evaluation Criteria:**

Overall, the proposed methods and evaluation in this submission appear to make sense:
- The proposed quantum algorithms, including the input (query) models and the input/output (in the form of pseudo code), are clearly stated. They incorporate standard quantum subroutines such as Quantum Mean Estimation and Quantum Maximum Searching to accelerate existing classical algorithms for time-dependent, finite-horizon MDPs.
- The correctness and computational complexity are established through rigorous mathematical proof. Technical overviews of the theoretical results are provided for every quantum algorithm, and they look plausible.

**Other Comments Or Suggestions:**

- Part 1 in Theorem 4.2: please use \left( and \right) in the big-O notation.
- While the paper claims the algorithms QVI-3 and QVI-4 achieve the "minimax optimal", this concept is never clearly defined in the main text. It would be nice to discuss the minimax optimality for self consistency.

**Other Strengths And Weaknesses:**

Strengths:
- The quantum lower bounds for finite-horizon MDPs are new results in the literature.
- The quantum mean estimation with binary oracles (QMEBO) subroutine is clearly stated and its efficiency is proved (Theorem 3.6). It can be of independent interest in future research.

Weaknesses:
- The efficient construction of the query input model (Definitions 3.2 and 4.1) appears highly nontrivial. Is it possible that these input models might be more expensive than the quantum algorithms themselves and thereby nullify the quantum speedups?

**Questions For Authors:**

1. Throughout this paper, I couldn’t find a clear explanation of the algorithm name ‘QVI.’ Could you clarify what ‘QVI’ stands for?
2. What are the possible applications/extensions of the techniques presented in the paper?

**Relation To Broader Scientific Literature:**

The pursuit of quantum speedups in stochastic control and reinforcement learning has been an active area of research, with finite-horizon MDPs serving as a standard problem in this domain.
- This submission investigates several quantum algorithms for time-dependent, finite-horizon MDPs and provides rigorous complexity analyses. These algorithms present evidence of quantum speedups for control and RL problems, which is promising for the entire community.
- In addition, the proof of quantum query lower bounds for finite-horizon MDPs advances our understanding of the computational limits of quantum computers in control and RL.

**Theoretical Claims:**

I read the technical overview of the quantum algorithms. The quantum accelerations are identified as follows:
1. In QVI-1, the optimal action is obtained by taking the maximum over the whole action space in the Bellman recursion, which is accelerated by Quantum Maximum Search (QMS, Theorem 3.3)
2. In QVI-2, the quantum speedup is achieved by an improved estimation of $P^T_{h|s,a} \hat{V}_{h+1}$ through the Quantum Mean Estimation with Binary Oracles (QMEBO, Theorem 3.6, proven by the authors)
3. In QVI-3, the Quantum Mean Estimation (QME, Theorem 4.2) is used to improve an $\epsilon$-approximation of $P^T_{h|s,a} \hat{V}_{h+1}$. This can be further accelerated by QMS as like QVI-1.
4. QVI-4 focuses on the computation of Q-values. It adapts the total variance technique (Wang et al., 2021) to the time-dependent, finite-horizon setting. Quantum speedups are achieved by a sequence of QME subroutines.

The technical summary is clear and easy to follow. The results appear to be technically solid, although I was not able to verify the proofs in the appendices line by line.

---

> ### Author Rebuttal · Authors · 2025-03-31
>
> We sincerely thank Reviewer FX1s for the thoughtful and constructive feedback. Below, we address the reviewer’s concerns regarding references, the typo in Theorem 4.2, the definition of "minimax optimal," the clarity of the algorithm name "QVI," the cost of constructing quantum oracles, and the potential applications and extensions of our techniques.
>
> The statement in Section 1 that "the quantum algorithm and analysis there cannot be applied to general finite-horizon MDPs," refers to (Naguleswaran et al., 2006), which only focuses on a specific class of MDPs—deterministic shortest path problems—and does not generalize to broader finite-horizon MDPs. We agree that (Wiedemann et al., 2022) addresses general finite-horizon MDPs, and we have properly cited this work in Section 1. However, their algorithm is inefficient, with a quantum sample complexity exponential in the state space for obtaining a near-optimal policy. Besides, we appreciate the reviewer for recommending [1] which provides a super-quadratic improvement in policy gradient estimation. However, [1] did not directly show how such a method can be applied to solve MDP problems. It would be an interesting direction to explore whether this powerful technique can be used to improve the results in (Cornelissen, 2018) for solving MDP problems.
>
> We apologize for the typo in Theorem 4.2 and we will correct this in the revised manuscript. Additionally, we acknowledge the lack of clarity in defining "minimax optimal." To clarify, we plan to replace "minimax optimal" with " (asymptotically) optimal" and formally define that an algorithm is (asymptotically) optimal if its query/sample complexity matches the corresponding lower bound up to constant factors [3]. Accordingly, we propose to revise the claim that "**QVI-3** and **QVI-4** are nearly minimax optimal" to "**QVI-3** and **QVI-4** are nearly (asymptotically) optimal (up to log terms) in computing near-optimal V/Q values and policies, provided the time horizon $H$ is a constant." For example, **QVI-3**'s quantum sample complexity in computing near-optimal V values and policies is $\tilde{O}\left( \frac{S \sqrt{ A } H^{3}}{\epsilon} \right)$ (Theorem 4.4) while the quantum lower bound is $\Omega\left( \frac{S\sqrt{ A }H^{1.5}}{\epsilon \log^{1.5}(\epsilon^{-1})} \right)$ (Theorem 4.7), differing by a factor of $H^{1.5}$ and a log term.
>
> We apologize for the lack of clarity in the algorithm name "QVI." We plan to revise the sentence "..., we propose a quantum value iteration algorithm QVI-1, ..." in the first paragraph of our summarized contributions in Section 1 to explicitly introduce the term "QVI". "QVI" stands for Quantum Value Iteration, where “VI” refers to the classical Value Iteration algorithm and “Q” indicates our quantum adaptation using subroutines like **QMS** and **QME**.
>
> We appreciate the concern about the cost of constructing the quantum oracles in Definitions 3.2 and 4.1. For the quantum generative model in Definition 4.1, as noted in our response to Reviewer Xd2s, **the classical generative model G and the quantum generative model $\mathcal{G}$ have similar costs at the elementary gate-level** if the classical circuit of the classical generative model is accessible. Furthermore, assuming that the classical generative model can be called in constant time and that we have access to quantum random access memory (QRAM) in [2], the time complexities of **QVI-3** and **QVI-4** are the same as the sample complexities of **QVI-3** and **QVI-4** up to log terms, so the reported speedups for **QVI-3** and **QVI-4** remains valid. Similarly, the time complexity of **QVI-1** and **QVI-2** is not degraded by the construction of quantum oracle $O_{\mathcal{QM}}$ (Definition 3.2) either. Specifically, suppose the classical oracle $O_{\mathcal{M}}$ in Definition 3.1 is a computer program, and we have its source code, we can also efficiently convert the classical circuit of $O_{\mathcal{M}}$ to a quantum circuit to implement the quantum oracle $O_{\mathcal{QM}}$. Unlike the quantum generative model, the output of the quantum oracle $O_{\mathcal{QM}}$ is not a superposition. Therefore, even without access to QRAM, the time complexities of **QVI-1** and **QVI-2** are the same as the quantum query complexities of **QVI-1** and **QVI-2** up to log terms as long as the classical oracle $O_{\mathcal{M}}$ can be called in constant time.
>
> We thank the reviewer for encouraging more discussion on the applications and extensions of our techniques. Our quantum algorithms have potential use in robotics (e.g., path planning) and operations research (e.g., inventory management). Our theoretical techniques could be extended to partially observable or multi-agent MDPs.
>
> [1] Clayton, Connor, et al. "Differentiable Quantum Computing for Large-scale Linear Control.".
>
> [2] Giovannetti, Vittorio, Seth Lloyd, and Lorenzo Maccone. "Quantum random access memory.".
>
> [3] Cormen, Thomas H., et al. "Introduction to algorithms".

---

### Official Review · Reviewer_LjjN · 2025-03-12

**Overall Recommendation:** 3

**Summary:**

In this work, the authors propose quantum algorithms for solving time-dependent, finite-horizon Markov Decision Processes (MDPs). The goal is to estimate the optimal policy that maximizes the expected reward over the finite time horizon, given a finite and discrete state and action space. Equivalently, this task can be viewed as maximizing the V-value function, which represents the sum of all future rewards for a given policy and initial state.
The authors present quantum algorithms for estimating the optimal policy and value function in two settings: the exact dynamics setting, where the agent has full knowledge of the transition probabilities for all state-action pairs at each time step, and the generative model setting, where the agent can only sample transition states for specific state-action pairs using a generative model. In both cases, the proposed quantum algorithms achieve a polynomial advantage over the best known classical algorithm in terms of query complexity to the oracle giving the transition probabilities.
In the exact dynamics setting, the authors leverage the quantum maximum searching algorithm to achieve a quadratic improvement in query complexity, reducing it from O(A) to O(\sqrt(A)), where A is the size of the action space. Additionally, they propose a second algorithm that provides a quadratic speedup with respect to the state space dimension when computing an epsilon-approximation of the optimal policy and value function. In the generative model setting, the proposed quantum algorithms achieve polynomial speedups in query complexity with respect to the action space, time horizon interval, and approximation error for the optimal policy, value function (V-value), and state-action value function (Q-value). These speedups are obtained using the quantum mean estimation algorithm.
Finally, the authors establish a lower bound for quantum algorithms in the generative model setting, indicating that their proposed algorithms are nearly optimal.

**Claims And Evidence:**

To the best of my knowledge, all the claims appear to be well-supported. In particular, the quantum speedups rely on well-established algorithms, such as Quantum Maximum Search and Quantum Mean Estimation, and it seems reasonable that these contribute to a computational advantage.

**Essential References Not Discussed:**

I am not sure about essential, but https://arxiv.org/pdf/2212.09328 contains results about comparing finite horizon approximations of infinite time settings which I think are important for this discussion.

**Experimental Designs Or Analyses:**

The paper does not contain any experiment.

**Methods And Evaluation Criteria:**

The paper does not include any numerical experiments to support its claims, as it is purely theoretical. However, benchmarking the classical and quantum algorithms based on query complexity to the transition probability function seems reasonable. Additionally, the implementation of the quantum oracle for this function appears well-founded, making the comparison fair.

**Other Comments Or Suggestions:**

no other comments

**Other Strengths And Weaknesses:**

Strength: I think the results in the paper are solid and it is an interesting combination of well-known quantum algorithms for MDPs.
Weaknesses:
- I am concerned about the novelty of the results. As the authors acknowledge, the use of quantum maximum search and quantum mean estimation for solving MDPs was already introduced in [3] for the infinite-horizon case. However, they do not clearly explain the challenges in adapting these techniques to the finite-horizon setting or what novel contributions they introduced to make this adjustment.
Also appendix B of https://arxiv.org/pdf/2212.09328 establishes a rather tight relationship between finite and infinite time horizons. Furthermore if I am not mistaken that paper and related works actually prove results for finite horizons and then show it also implies good performance for infinite horizons.
-Regarding the time-dependent case, for finite times, one can reduce this to time independent case, by expanding the state space to encode the timestamp. If one does this and applies previous results, do we get something different?
- The speedup achieved is only polynomial (quadratic) compared to classical algorithms using rather well known techniques. Additionally, in the exact dynamics setting, the improvement is only relative to the best-known classical algorithm. Establishing a lower bound for classical algorithms, as was done in the generative model setting, would strengthen the results.
- The presentation of the results lacks clarity. In particular, a high-level overview of the proposed algorithms and their workings would significantly improve the readability and understandability of the paper

**Questions For Authors:**

1)	What are the difficulties of extending the results in [3] to the finite horizon case considered in this paper?
2)	Could you establish lower bounds on the query complexity for classical methods in the exact dynamics setting?
3)	Could you provide a high-level explanation of your algorithms in the main text, in addition to the pseudocode?
4)    Regarding the time-dependent case, for finite times, one can reduce this to time independent case, by expanding the state space to
       encode the timestamp. If one does this and applies previous results, do we get something different?

**Relation To Broader Scientific Literature:**

The paper discusses its relation to previous work in the introduction, highlighting that quantum algorithms have already been proposed for infinite-horizon problems with time-invariant value functions [3]. In contrast, this work focuses on finite-horizon and time-dependent scenarios. Additionally, the authors explicitly state where the key query advantage arises from the application of previously known algorithms. (Quantum Maximum Search [1] and Quantum Mean Estimation [2]).

**Theoretical Claims:**

I have skimmed the proofs and all seem fine. The claims are consistent what one would expect based on previous literature.

---

> ### Author Rebuttal · Authors · 2025-03-31
>
> We appreciate the reviewer’s concern regarding the perceived lack of novelty in our quantum algorithms. While our algorithms leverage **QMS** and **QME** as in prior work (Wang et al., 2021), their analysis for infinite-horizon MDPs can not be readily applied to time-dependent and finite-horizon MDPs. A key contribution of our work lies in the design and analysis of **QVI-4**. Note that in our setting, the value operator in Definition 2.1 is time-dependent and lacks contraction property-unlike the infinite-horizon case, which allows iterative policy/value updates via fixed points. This lacks of contraction property in our setting poses significant challenges in designing **QVI-4**. In SolveMdp1 (Wang et al., 2021), the $\epsilon_{k}$ optimal V/Q-values and policy can be directly used to initialize the next epoch $k+1$. This is not feasible in our case. To address this, we propose another initialization strategy in **QVI-4**, setting $V_{k+1,h}^{(0)}= V_{k,h}$ for all $h\in[H]$ and initializing $V_{k,H}= V_{k,H}^{(0)}=\mathbf{0}$. We then use induction on $k$ to prove the correctness of **QVI-4** (Lemma B.4), showing non-trivial technical contributions in extending quantum algorithms to finite-horizon MDPs.
>
> We appreciate the reviewer’s question about the lower bound for classical methods in the exact dynamics setting. We can derive a classical lower bound of $\Omega(S^2 A)$ in the exact dynamics setting by adapting the method from [2] to finite-horizon MDPs. Consider two hard instances $M_{1}$​ and $M_{2}$ similar to those in [2]. ​In $M_{1}$, we can derive that the optimal value for $s\in \mathcal{S}\_{U}$ is $V^{\*}\_{h}(s)=\frac{H-1-h}{2}$, while in $M_{2}$, the optimal value for $s\in \mathcal{S}\_{U} \setminus \{ \overline{s} \}$ remains $V^{\*}\_{h}(s)=\frac{H-1-h}{2}$ but $V\_{h}^{\*}(\overline{s})=H-1-h$. To achieve $\frac{H-1}{4}$-optimal $V\_{0}$ with high probability, any algorithm must distinguish $M_{1}$ from $M_{2}$, requiring to search for two discrepancies in an array of size $\Omega(S^{2} A)$. Therefore, the classical lower bound for computing an $\epsilon$ optimal $V_{0}$ for the time-independent and finite-horizon MDP is $\Omega(S^{2} A)$ for $\epsilon \in (0,\frac{H-1}{4})$. This implies the classical lower bound for obtaining an $\epsilon$ optimal policy for the time-dependent and finite-horizon MDP in the exact dynamics setting is $\Omega(S^{2} A)$. This classical lower bound can also be derived using the reduction technique in Appendix B.3.
>
> We acknowledge the reviewer’s concern on the clarity of a high-level overview of the proposed algorithms. Sections 3 and 4 already include high-level overviews (also noted by Reviewer FX1s), and we plan to make them more prominent. Besides, **QMEBO** uses binary oracles to encode the $P_{h|s,a}$ and $\hat{V}\_{h}$ and transfer this information to amplitude via controlled rotation unitary operators, followed by amplitude estimation to compute an estimate of $P_{h|s,a}^{T} \hat{V}\_{h}$.
>
> We thank the reviewer for pointing out the missing reference [1]. While [1] studies quantum speedup for policy gradient methods in finite-horizon MDP and explores connections to infinite-horizon MDP, our work differs in three key differences. First, [1] only provides quantum speedups for estimating the gradients of V values. It does not produce a bound on the overall complexity for their algorithms to converge and obtain a near-optimal policy. In contrast, we directly quantify the complexity for computing a near-optimal policy. Second, regarding the approximation between finite-time and infinite-time, note that [1] uses the approximation to solve infinite-horizon MDP. Perhaps due to this reason, their result (Theorem 3.1 and 4.1) requires the finite-horizon MDP to also have a discount factor away from 1. In contrast, our paper is not interested in infinite-horizon MDP. Instead, we only use it to derive a lower bound. As a result, for our solution to finite-horizon MDP, we do not require a discount factor away from 1. Therefore, our results are new and more broadly applicable than [1].
>
> We agree that a time-dependent MDP can be converted to a time-independent MDP by expanding the state space to $\mathcal{S}' = \mathcal{S} \times [H]$. However, this transformation does not reduce the complexity. Even including [1] , the only known quantum algorithm for general finite-horizon MDP remains that of (Wiedemann et al., 2022), whose sample complexity to obtain an $\epsilon$-optimal policy is $O(A^{3SH/2}H/\epsilon)$ for time-dependent setting and $O(A^{3S/2}H/\epsilon)$ for time-independent setting. Thus, even after reduction, their complexity with the enlarged state space $S'$ remains $O(A^{3SH/2}H/\epsilon)$, which is inferior to our algorithms.
>
> [1] Jerbi, Sofiene, et al. "Quantum policy gradient algorithms.".
>
> [2] Chen, Yichen, and Mengdi Wang. "Lower bound on the computational complexity of discounted markov decision problems.".

---

> > ### Comment · Reviewer_LjjN · 2025-04-03
> >
> > I thank the authors for a strong and clear response, and I am inclined to raise the score.
> > I have an additional question. The authors state:
> >
> >  "We thank the reviewer for pointing out the missing reference [1]. While [1] studies quantum speedup for policy gradient methods in finite-horizon MDP and explores connections to infinite-horizon MDP, our work differs in three key differences. First, [1] only provides quantum speedups for estimating the gradients of V values. It does not produce a bound on the overall complexity for their algorithms to converge and obtain a near-optimal policy. In contrast, we directly quantify the complexity for computing a near-optimal policy. Second, regarding the approximation between finite-time and infinite-time, note that [1] uses the approximation to solve infinite-horizon MDP. Perhaps due to this reason, their result (Theorem 3.1 and 4.1) requires the finite-horizon MDP to also have a discount factor away from 1. In contrast, our paper is not interested in infinite-horizon MDP. Instead, we only use it to derive a lower bound. As a result, for our solution to finite-horizon MDP, we do not require a discount factor away from 1. Therefore, our results are new and more broadly applicable ..."
> >
> > Is the estimation of gradients not the dominating cost in the process?
> > I appreciate the categorical difference between unit discount and arbitrary close to unit. But is this an important difference beyond this?

---

> > > ### Author Response · Authors · 2025-04-07
> > >
> > > We sincerely thank Reviewer LjjN for the constructive feedback and for the positive recognition of our work throughout the review process. We are especially grateful for the reviewer’s decision to raise the score. Below, we address the additional questions regarding (i) the computational cost of estimating the gradient of the value function in policy gradient algorithms, and (ii) the distinction between unit discount and near-unit discount settings.
> > >
> > > While we agree that estimating the gradient of V value in the policy gradient methods is computationally expensive, we emphasize that their convergence properties are also important. Note that the underlying optimization problem is typically non-convex (see, for example, Lemma 11.5 in [2]). In that case, these algorithms may only converge to local optima or stationary points, which may be far from the global optimum and thus fail to obtain a near-optimal policy. In contrast, our QVI algorithms are designed to consistently obtain a near-optimal policy, offering a significant advantage in solution quality. Second, even reaching a stationary point with policy gradient methods remains computationally intensive. Specifically, we believe the overall sample complexity of their algorithms with respect to the error term $\epsilon$ is much worse than that of our **QVI** algorithms. As shown in Lemma 11.8 of [2], for a $\beta$-smooth value function $V_{\pi_\theta}$ with stochastic gradients (variance bounded by $\sigma^2$), the stochastic gradient ascent (SGA) algorithm converges to an $\epsilon$-approximation stationary point in expectation after $K\geq O\left( \frac{\sigma^{2}}{\epsilon^2} \right)$ iterations. As noted in [2], the variance of the stochastic gradients is normally huge in practice, which further increases the computational burden. When combined with the quantum techniques for estimating the gradient of $V_{\pi_{\theta}}$ in [1], the total quantum sample complexity of obtaining a stationary point for a finite-horizon MDP-with a reward function bounded by $\mid R\mid_{\max}=1$, time horizon $H$ and discount factor $\gamma\in(0,1)$-scales as $O\left( \sqrt{ d }  \frac{D H^{3}\sigma^{2}}{\epsilon^{3} (1-\gamma)} \right)$ (numerical gradient estimation) or $O\left( d ^{\xi(p)}  \frac{B_{p} H^{2} \sigma^{2}}{\epsilon^{3} (1-\gamma)} \right)$ (analytical gradient estimation). Note that both quantum sample complexities show an $O\left( \frac{1}{\epsilon^3} \right)$ dependence on the error term $\epsilon$. This highlights the significant computational cost of policy gradient methods, even for achieving a suboptimal stationary point. In contrast, **QVI-3** and **QVI-4** demonstrate a much more favorable $O\left( \frac{1}{\epsilon} \right)$ dependence on $\epsilon$, which implies that our algorithms require much fewer quantum samples to achieve a high-quality near-optimal policy, especially as the desired precision ($\epsilon \rightarrow 0$) increases.
> > >
> > > We thank the reviewer for the insightful follow-up question regarding the difference between the unit discount and arbitrary close-to-unit discount settings. Although we agree with the reviewer that in the finite-horizon MDP setting, there should not be a significant theoretical distinction between these two setting, the results in [1] show an $O\left( \frac{1}{1-\gamma} \right)$ dependence on $\gamma$ which will explode when $\gamma \to 1$. Specifically, in a finite-horizon MDP with a fixed horizon $H$, the cumulative reward is computed over a finite number of steps, and the value function is inherently bounded by the horizon $H$ and the reward function (bounded by $\mid R\mid_{\max} = 1$ in our case). As a result, the introduction of a discount factor $\gamma$ is not necessary to ensure the boundedness of the value function, unlike in the infinite-horizon MDP setting, where $\gamma < 1$ is typically introduced to make the infinite sum of discounted rewards remains finite. In fact, setting $\gamma=1$ is more common for finite-horizon MDPs in many applications, such as robotics, game playing, and resource management, where the goal is to maximize total reward over a fixed time period—e.g., completing a task within a set number of steps. Note that our **QVI** algorithms can easily extend to solve those finite-horizon MDP with a discount factor $\gamma<1$ without changing the quantum query/sample complexity. In contrast, as discussed above, the quantum sample complexities of reaching a stationary point for a finite-horizon MDP using the policy gradient method in [1] show an $O\left( \frac{1}{1-\gamma} \right)$ dependence on $\gamma$. This implies that their result requires the finite-horizon MDP to have a discount factor $\gamma\in(0,1)$ and incurs significantly higher sample complexity as $\gamma \to 1$. Therefore, our results are more broadly applicable than [1].
> > >
> > > [1] Jerbi, Sofiene, et al. "Quantum policy gradient algorithms.".
> > >
> > > [2] Alekh, Agarwal, et al. "Reinforcement Learning: Theory and Algorithms".

---

### Official Review · Reviewer_Xd2s · 2025-03-13

**Overall Recommendation:** 3

**Summary:**

In this submission, the authors presented quantum algorithms for finite-horizon Markov decision processes (MDPs). These quantum algorithms cover MDPS in the exact dynamics setting and in the generative model setting. Polynomial speedups are achieved. In addition, lower bounds on the query complexity for the generative model setting are also proved in this submission, although not tight. The quantum algorithms are based on well-known quantum techniques including quantum maximum finding and quantum mean estimation.

## update after rebuttal
I appreciate the authors' rebuttal comments to address my concerns. Most of my concerns have been addressed and I have increased my score.

**Claims And Evidence:**

Theoretical proofs are provided for the claims.

**Essential References Not Discussed:**

N/A

**Experimental Designs Or Analyses:**

N/A

**Methods And Evaluation Criteria:**

N/A

**Other Comments Or Suggestions:**

N/A

**Other Strengths And Weaknesses:**

I appreciate the authors' effort on studying quantum speedups for an important machine learning problem. In the generative model setting, the classical lower bounds presented in this submission show a clear separation between quantum and classical algorithms. The quantum lower bounds rule out the possibility of exponential speedups.

There are a few weaknesses:

1. Technically, the quantum algorithm is presented in the form that simply replaces the maximum finding and mean estimation of the classical algorithm to well-known quantum subroutines. In this sense, these quantum algorithms lack technical contribution in both the quantum and classical regimes.

2. Although it is nice to see the low bounds, they are still not tight. (But I think this is a minor weakness).

3. I am not fully convinced about the quantum generative model shown in Definition 4.1. In general, QSample is considered as a harder model than the classical sampling model. So it might not be fair to compare the quantum generative model with the classical model. It would be nice if some justification of this comparison can be provided.

**Questions For Authors:**

How to justify the comparison between the quantum and classical generative models, particularly for MDP?

**Relation To Broader Scientific Literature:**

The problems this submission studies may find applications in quantum machine learning.

**Theoretical Claims:**

I checked the proofs and they appear to be correct.

---

> ### Author Rebuttal · Authors · 2025-03-31
>
> We sincerely thank Reviewer Xd2s for the detailed feedback on our submission. Below, we address the reviewer’s concerns regarding the technical contributions and the justification of the comparison between quantum and classical generative models, while proposing feasible revisions to strengthen our submission.
>
> We acknowledge the reviewer’s concern that our quantum algorithms may appear to simply replace classical subroutines with well-known quantum ones, such as quantum maximum searching **QMS** (Durr & Hoyer, 1999) and quantum mean estimation **QME** (Montanaro, 2015). However, we would like to highlight the significant technical contributions of our work, which go beyond straightforward substitutions. First, a key technical contribution of our work is the development of the Quantum Mean Estimation with Binary Oracles (**QMEBO**, Algorithm 3) subroutine in the exact dynamics setting (Section 3.2). QMEBO adapts quantum mean estimation for binary oracles, achieving a speedup in the state space $S$ (from $O(S)$ to $O(\sqrt{ S } /\epsilon)$) for near-optimal policies. This is a non-trivial adaptation that enables the application of quantum techniques to MDPs. Second, while our algorithms do leverage well-known **QMS** and **QME**, it is nontrivial to infuse the two existing quantum subroutines into existing reinforcement learning frameworks. For example, as pointed in Section 4.2, when integrating the total variance technique and **QME**, we can not directly apply **QME2** as its classical counterpart, because **QME2** requires prior knowledge of $\sigma\_{k,h}^{s,a}$ and an upper bound on $\sigma\_{k,h}^{s,a}$ to estimate the $\mu\_{k,h}^{s,a}$ to an error of $\frac{\epsilon \sigma\_{k,h}^{s,a}}{(2H^{1.5})}$. To address this, we propose to use **QME1** to obtain an estimate $(\hat{\sigma}\_{k,h}^{s,a})^{2}$ of $(\sigma\_{k,h}^{s,a})^{2}$ with an error $4b$ and use **QME2** to estimate $\mu\_{k,h}^{s,a}$ with an error proportional to $\overline{\sigma}\_{k,h}^{s,a}=\sqrt{  (\hat{\sigma}\_{k,h}^{s,a})^{2}+4b}$. This integration demonstrates the technical depth required to adapt quantum subroutines to our setting, contributing to the field of quantum reinforcement learning.
>
> We thank the reviewer for raising the important question about the fairness of comparing quantum and classical generative models for MDPs. In fact, the comparison is fair because implementing quantum generation model $\mathcal{G}$ (Definition 4.1) has a comparable overhead as the classical generation model $G$ (Eq. 3, Section 4). In the classical generative model setting, it is assumed that we can access to a classical generative model/simulator $G$ that draws samples from the distribution $P_{h|s,a}$. The classical generative model makes particular sense when the environment is a computer program, and thus we can use the same computer program to simulate the drawing of samples. Specifically, if the simulator is a computer program and we have its source code, then for the classical generative model $G$ we can produce a Boolean circuit $\mathcal{C}$ that acts as the simulator, i.e., draws samples from the distribution $P_{h|s,a}$. For the quantum generative model, we use the fundamental result in quantum computation ([1], Nielsen & Chuang, 2010, Section 1.4.1) that any classical circuit $\mathcal{C}$ with $N$ logic gates can be efficiently converted into a quantum circuit $\mathcal{Q}$ with $O(N)$ logic gates, capable of computing on quantum superpositions of inputs. Moreover, the conversion is efficient and can be achieved by simple conversion rules at the logic gate level by using the Toffoli gate. The authors in (Wang et al., 2021, arXiv:2112.08451) confirmed this by explicitly constructing the quantum generative model for infinite-horizon MDPs from a circuit of the corresponding classical generative model in Appendix A. We believe such a construction method can be readily extended to the case of finite-horizon MDPs. Thus, the classical generative model $G$ and the quantum generative model $\mathcal{G}$ **have comparable costs at the elementary gate level**, making the comparison between the quantum sample complexity with the classical sample complexity fair. Furthermore, the time complexities of **QVI-3** and **QVI-4** are the same as the quantum sample complexities of **QVI-3** and **QVI-4** up to log terms under the assumptions that the classical generative model can be called in constant time and that we have access to quantum random access memory (QRAM) proposed in [2].  Roughly speaking, QRAM is a memory that stores the classical values and that allows them all to be read at once in quantum superposition. To address the reviewer's concern, we propose to add a paragraph in Section 4 to explicitly justify the comparison between the quantum and classical generative models.
>
> [1] Bennett, Charles H. "Logical reversibility of computation.".
>
> [2] Giovannetti, Vittorio, Seth Lloyd, and Lorenzo Maccone. "Quantum random access memory.".

---

### Official Review · Reviewer_tc3r · 2025-03-14

**Overall Recommendation:** 3

**Summary:**

This paper explores quantum algorithms designed to improve the efficiency of solving finite-horizon Markov Decision Processes (MDPs) in two settings: exact dynamics and generative models. The main contribution is the introduction of quantum value iteration (QVI) algorithms.

**Claims And Evidence:**

The claims made in the paper are generally supported by formal proofs, especially in the complexity and correctness of the proposed quantum algorithms. There are no immediately obvious problematic claims.

**Essential References Not Discussed:**

NA

**Experimental Designs Or Analyses:**

The manuscript does not include explicit experimental evaluations but focuses on the theoretical aspects of the algorithms. This is appropriate for the topic, as the aim is to establish the theoretical foundations of quantum speedups in MDPs. The only concern might be the lack of empirical validation for the proposed algorithms, though this is often the case for early theoretical work in quantum algorithms.

**Methods And Evaluation Criteria:**

The proposed methods and evaluation criteria are suited to the problem.

**Other Comments Or Suggestions:**

The paper could benefit from a clearer distinction between the classical methods it compares against. While it mentions improvements over classical algorithms, specific examples or comparisons to concrete classical algorithms could strengthen the case for quantum superiority.

**Other Strengths And Weaknesses:**

The paper is original in its focus on finite-horizon MDPs and provides a clear exploration of quantum speedups in this domain. The algorithms are innovative and mathematically rigorous. One weakness, however, is the lack of empirical validation of the proposed algorithms in real-world applications, which could have provided a clearer understanding of their practical impact.

**Questions For Authors:**

NA

**Relation To Broader Scientific Literature:**

The paper connects well with prior work in quantum reinforcement learning and quantum algorithms for MDPs.

**Theoretical Claims:**

The theoretical claims in the paper, particularly concerning the correctness and complexity of the quantum algorithms, appear solid. The proofs provided for the algorithms seem logically consistent.

---

> ### Author Rebuttal · Authors · 2025-03-31
>
> We sincerely thank Reviewer tc3r for the thorough and constructive feedback on our submission. Below, we address the reviewer’s concerns and provide clarifications to strengthen our submission.
>
> We acknowledge the reviewer’s concern regarding the lack of empirical validation in our current submission. As a primarily theoretical contribution, our work focuses on establishing the correctness and complexity of the proposed quantum value iteration (QVI) algorithms for finite-horizon MDPs in two distinct settings, as evidenced by the formal proofs provided in the appendix. We recognize the importance of empirical evaluation in demonstrating practical impact; however, the current state of quantum hardware poses significant challenges for implementing and testing quantum algorithms at scale. Specifically, the limited qubit counts, high error rates, and restricted access to large-scale quantum computers make it difficult to empirically validate the proposed speedups at this stage. We are committed to pursuing empirical validation in future work. Besides, to address the reviewer’s suggestion for making the quantum speedup more concrete, we propose to include an illustrative example of a toy MDP, such as the Inventory Management Problem, in Section 3 and Section 4 to demonstrate the difference in query complexity between classical value iteration and QVI algorithms.

---

### Decision · Program_Chairs · 2025-05-01

**Decision:**

Accept (poster)

**Comment:**

The reviewers agree that the theoretical contribution is solid and well-founded. The rebuttal was described by one reviewer as strong and clear.

Overall, no significant points of criticism remain, so I rate the paper as Accept.